# Consistency Is Not Always Correct: Towards Understanding the Role of Exploration in Post-Training Reasoning

## Abstract

Foundation models exhibit broad knowledge but limited task-specific reasoning, motivating post-training strategies such as RL with verifiable rewards (RLVR) and inference scaling with outcome or process reward models (ORM/PRM). While recent work highlights the role of *exploration* and *entropy stability* in improving pass@K, empirical evidence points to a paradox: RLVR and ORM/PRM typically reinforce existing tree-like reasoning paths rather than expanding the reasoning scope, raising the question of why exploration helps at all if no new patterns emerge. To reconcile this paradox, we adopt the perspective of Kim et al. (2025), viewing easy (e.g., simplifying a fraction) versus hard (e.g., discovering the some symmetry) reasoning steps as low versus high probability Markov transitions, and formalize post-training dynamics through Multi-task Tree-structured Markov Chains (TMC). In this tractable model, pretraining corresponds to tree-graph discovering, while post-training corresponds to CoT reweighting. We provably show that several phenomena recently observed in empirical studies arise naturally in this setting: (**1**) RLVR induces a *squeezing effect*, reducing CoT entropy and forgetting some correct paths; (**2**) population rewards of ORM/PRM encourage consistency rather than accuracy, thereby favoring common patterns; and (**3**) certain rare, high-uncertainty CoTs by base model are responsible for solving hard problem instances. Together, these explain why exploration—even when confined to the base model's tree scope—remains essential: it preserves access to rare but crucial CoTs needed for difficult cases, which are squeezed out by RLVR or unfavored by inference-scaling. Building on this, we further prove that exploration strategies such as rejecting easy instances and KL regularization help preserve rare CoTs. Empirical simulations corroborate our theoretical results.

## 1 Introduction

Foundation models provide broad knowledge and versatile capabilities across tasks, yet their task-specific reasoning remains constrained. For reasoning datasets with only $0/1$ verifiers, many studies explore post-training strategies, including Reinforcement Learning with Verifiable Reward (RLVR) finetuning (Xin et al., 2024; Shao et al., 2024; Guo et al., 2025a; Yu et al., 2025), as well as inference scaling with Outcome Reward Models (ORM) or Process Reward Models (PRM) (Lightman et al., 2023; Snell et al., 2024), both aiming to obtain task-specific experts.

Recently, a line of work has emphasized maintaining *exploration* and *entropy stability* to prevent entropy collapse, and observed that suitable entropy preservation during post-training yields *systematic performance gains*, such as improved pass@K on math benchmarks (Xiong et al., 2025; Li et al., 2025b; Ren & Sutherland, 2025; Wang et al., 2025; Cui et al., 2025; Zuo & Zhu, 2025).

However, seemingly **paradoxical** findings emerge: RLVR typically aligns models with target objectives by reinforcing existing reasoning paths, *rather than* expanding their tree-like reasoning scope (Snell et al., 2024; Yue et al., 2025; AI et al., 2025; Gandhi et al., 2025). Similarly, ORM/PRM-guided inference scaling biases models toward pre-existing Chain-of-Thought (CoT) patterns instead of incentivizing genuinely new branches. This raises a natural question:

*Why can exploration help, given post-training cannot explore beyond the base model's tree scope?*

Our work takes a first step toward reconciling this tension, motivated the key phenomena below.

**Phenomenon 1: Squeezing Effect of RLVR**. RLVR implicitly reduces CoT entropy (Li et al., 2025a; Wu et al., 2025; Deng et al., 2025), shrinking confidence over CoTs with high frequency to be correct, and sometime at the cost of forgetting certain correct CoTs within the base model's scope (Wu et al., 2025; Shao et al., 2024; Wen et al., 2025).

**Phenomenon 2: Neural Verifier Checks Consistency, Not Accuracy**. Inference-scaling with ORM/PRM may avoid the learner's squeezing effect, yet empirical evidence shows that neural scorers are prone to reward consistency rather than true accuracy (Xu et al., 2025; Guo et al., 2025b). As a result, they would favor CoTs that follow *common, high-frequency* reasoning patterns.

**Phenomenon 3: Merits of Rare CoTs.** A widely-utilized difficulty measure for reasoning dataset (e.g., GSM8K (Cobbe et al., 2021) and AQuA (Ling et al., 2017)) is the *pass rate* (i.e., the frequency with which a base model correctly solves an instance under parallel attempts) (Tong et al., 2024; Parashar et al., 2025). This implies that hard instances correspond to rare-but-correct CoTs with low model confidence, whereas common CoTs typically reflect frequent patterns to easier instances.

Taken together, these observations suggest a resolution:

*Exploration, **even when confined within the existing tree scope**, helps prevent the model from entirely forgetting rare CoTs that may be crucial for hard instances, and preserved broad-capability.*

**Our Contributions.** In this work, we rigorously formalize and prove these phenomena within a tractable theoretical framework. Motivated by the view that discrete graphs naturally abstract the sequential structure of complex reasoning (Xu et al., 2019; Sanford et al., 2024; Abbe et al., 2024; Besta et al., 2024), we model each reasoning step as a Markov *state transition* following Kim et al. (2025). Pretraining is framed as a *tree-graph* discovering process over child states across tasks, while post-training CoT generalization is modeled by a Multi-task Tree-structured Markov Chain (TMC). We prove that our toy model captures key **Phenomena 1–3** with population $0/1$ reward (expected accuracy), and then provide theoretical justification for exploration techniques such as rejecting easy questions (Yu et al., 2025; Xiong et al., 2025; Zhang et al., 2025a) and KL regularization. Our paper is organized as below.

- Sec. 2 introduces a multi-task Tree-structured Markov chain model to capture diverse CoT reasoning patterns across tasks, explicitly linking instance difficulty with pass rate.
- Sec. 3 analyzes a simple softmax model and shows that RLVRs inherit a *simplicity bias*, over-favoring easier CoTs due to the *advantage-driven squeezing effect*. We further provide theoretical justification for rejecting easy instances and applying KL regularization, which both promote valid hard CoT learning.
- Sec. 4 demonstrates that inference-scaling with ORM/PRM assigns credit to CoTs by their accuracy likelihood, leading to overemphasis on easier CoTs. We further show that PRMs with BoN can be interpreted as special cases of a more general *Doob h's Transformed-induced PRM* (DPRM). In principle, DPRM is equivalent to soft-BoN (Verdun et al., 2025) asymptotically, enabling adjustable preservation of base-model capabilities and better alignment with hard-to-reason and cross-task patterns.

Discussions of additional related work are in App. D. All proofs are deferred to the appendix.

**Humble Remark**. While we prove Phenomena 1–3 and the benefits of those existing techniques in our theory-friendly setting that captures partial but crucial rationales, we do not overclaim their *direct* applicability to GPT or large-scale models, given the many unmodeled complexities, as discussed in App. E and G.4.

## 2    TREE-STRUCTURED MULTI-TASK REASONING

### 2.1    MULTI-TASK COT AS TREE-STRUCTURED MARKOV CHAINS

We propose Tree-structured Markov Chain (TMC) framework to abstractly model the tree-like reasoning capability of base model, following Kim et al. (2025); Nichani et al. (2024).

**Definition 1** (Tree-structured Markov Chains (TMC).)**.** *A process $X = (X_t)_{t \geq 0}$ is defined on a finite state space $S = \bigcup_{l=1}^{L} S_l$, where $S_l \cap S_{l'} = \emptyset$ for $l \neq l'$, and transitions occur from $S_l$ to $S_{l+1}$ with probability kernel $\mathbb{P}(\cdot|o_l)$ for $o_l \in S_l$. Define $M_0 = |S_1|$ and $M = \max_{l, o_l \in S_l} |C_{o_l}|$, where $C_{o_l} \subset S_{l+1}$ is the high-probability transition subset. The TMC satisfies:*

- *Root states $o_1 \in S_1$ are sampled with $\mathbb{P}_{\text{TMC}}(o_1) = \Theta(1/M_0)$.*

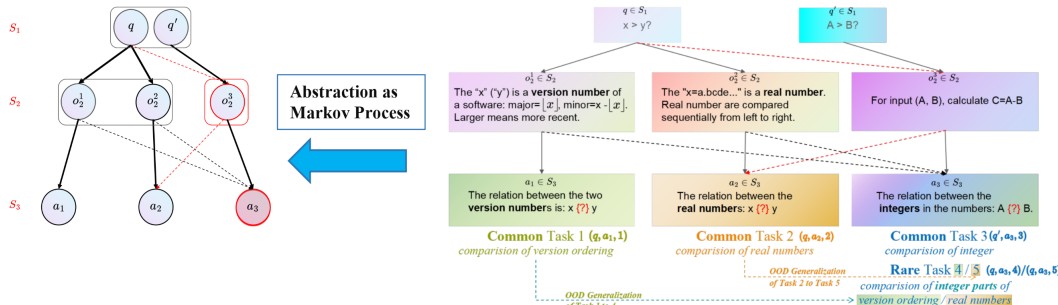

Figure 1: *Left*: abstraction of a 3-layer TMC. Nodes are states grouped into layers $S_1$–$S_3$; solid arrows denote high-prob (confident) transitions and dashed arrows denote low-prob (unsure) transitions. A task is a tuple $(\mathbf{q}, \mathbf{a}, \mathbf{k})$, where $\mathbf{q} \in \{q, q'\}$ is the question *state*, $\mathbf{a} \in \{a_1, a_2, a_3\}$ is the answer *state*, and $\mathbf{k} \in [5]$; *Right*: a concrete illustration of a 5-task, 3-layer Multi-task TMC. $x, y$ in $q$ represent real numbers (with decimals), whereas $A, B$ in $q'$ represent integers. We here use two instances, namely $3.9 > 3.11$? of $q$ and $3 > 3$? of $q'$ to describe the five tasks: (T1) decimal version ordering ($3.9 < 3.11$); (T2) real-number comparison ($3.9 > 3.11$); (T3) integer equality ($3 - 3 = 0$); (T4) integer-part version ordering (e.g., $3.9 = 3.11$); and (T5) integer-part real-number comparison ($3.9 = 3.11$). Tasks 1–3 are common and each admits $\geq 1$ easy-to-reason CoT, while Tasks 4–5 are rare and admit *only* hard-to-reason CoTs. For Task 2, there are two *valid* CoTs: $q \to o_2^2 \to a_2$ (where $o_2^2$ merely left-to-right compares number) and $q_2 \to o_2^3 \to a_2$ (where $o_2^3$ performs the arithmetic calculation). The instance $3.9 > 3.11$? admits both CoTs *correct*. However, for *hard* question instances such as $0.8 + 3.1 > 2.11 + 1.0$?, *only* the hard-to-reason CoT $q_2 \to o_2^3 \to a_2$ is *correct*, since it requires explicit calculation—left-to-right token comparison alone doesn't suffice.

- *For $o_l \in S_l$ and $o_{l+1} \in C_{o_l}$, we have $\mathbb{P}_{\text{TMC}}(o_{l+1} \mid o_l) = \Theta(1/M)$, while if $o'_{l+1} \notin C_{o_l}$, $\mathbb{P}_{\text{TMC}}(o'_{l+1} \mid o_l) = o(1/M^2)$ (feeble, $\geq c > 0$) or 0.*
- *The topology ensures that for each $q \in S_1$ there are $n_q = O(1) \geq 1$ high probability CoT traces $(q = o_1, \ldots, o_L)$, i.e. traces with $o_{l+1} \in C_{o_l}, \forall l \in [L-1]$.*

In our TMC (Def. 1), *states* represent *logical assertion* (e.g., a sentence or mathematical expression) rather than surface tokens (Kim et al., 2025). Especially, the CoTs with $\geq 1$ sparse edge (i.e., edge with feeble transition probability $o(1/M^2)$) are called *hard-to-reason* CoTs, otherwise *easy-to-reason* CoTs. The following definition formalizes the reason we called them "*easy*" or "*hard*" based on their uncertainty, modeling after the widely utilized difficulty measure–namely *pass rate*– for reasoning dataset (e.g., GSM8K (Cobbe et al., 2021) and AQuA (Ling et al., 2017)).

**Definition 2** (Multi-task Capability in TMC. (Informal)). *Let $X = (X_t)_{t \geq 0}$ be a TMC (Def. 1), and let $\mathcal{T}$ be a set of tasks. Each task $k \in \mathcal{T}$ is specified by a collection of state tuples $(q, a, k)$, where all tuples have distinct $q$ and $a$. Among these CoTs, a nonempty subset is* valid *for $(q, a, k)$, satisfying:*

- *(i) all easy-to-reason CoTs are* valid *for $(q, a, k)$ and* invalid *for any $k' \neq k$;*
- *(ii) every nonzero transition in TMC appears in $\geq 1$* valid *CoT across all tasks;*
- *(iii) each $(q, a, k)$ induces a QA distribution $\mathcal{D}_a^{q,k}$, and for any sampled instance $(\mathbf{Q}, \mathbf{A}) \sim \mathcal{D}_a^{q,k}$, only a determined subset of* valid *CoTs is* correct, *where the probability that any* valid *CoT is* correct *for the $(\mathbf{Q}, \mathbf{A})$ is **proportional** to its likelihood among* valid *CoTs.*

*A task is* common *if it admits $\geq 1$* valid *easy-to-reason CoTs, and* rare *otherwise.*

Fig. 1 illustrates Def. 1 and Def. 2; a more detailed version of Def. 2 (Def. 4) is given in Sec. H. Condition (i) avoids major task conflicts, (ii) removes redundancy so every edge contributes, and (iii) links model confidence to *pass rate* per **Phenomenon 3**, enabling error analysis. We distinguish two notions of CoT:

- *Validity*[1]: a **task-level** property, indicating whether it solves *any* $(\mathbf{Q}, \mathbf{A})$ under $(q, a, k)$.
- *Correctness*: an **instance-level** property, deterministically defined for a specific $(\mathbf{Q}, \mathbf{A})$.

**Why this definition?** First, **Phenomenon 1** directly motivates us to formally bridge uncertainty and *pass rate* across tasks. Yet, not all rare outputs are useful—some may be not *correct* for all instances of the current task, surfacing only due to *shared* reasoning states across tasks. This motivates our definition of *validity* to distinguish useful task-specific CoTs. Second, the key property inherent

---

[1]Our *validity* also speaks that no all hard-to-reason CoTs useful, see App. E a discussion.

from the pass rate is that, easy-to-reason CoTs cover most instances in $\mathcal{D}_a^{q,k}$, but some instances can still *only* be *correctly* solved by hard-to-reason CoTs. Our Multi-task TMC framework highlights the importance of such rare reasoning paths, consistent with large-scale evidence that many errors on GSM8K, AQuA, and MATH arise from *misapplied common patterns* (e.g., assuming overlapping events are independent) per observed in Sun et al. (2025); more relevant empirical examples appear in Rmk. 5. We can then define outcome signal to verify *correctness* as follow.

**Outcome Reward Signal.** Let $\mathbf{o}_l \in \mathbb{R}^{|S|}$ be the one-hot encoding of observation $o_l$, and $\mathbf{o} = (\mathbf{o}_1, \ldots, \mathbf{o}_L)^\top$ the full trajectory. In mathematical reasoning tasks, the *correctness* of a CoT trace is deterministic and verifiable Yue et al. (2025); Xiong et al. (2025); Setlur et al. (2025a; 2024; 2025b), typically via formal systems such as *Lean4* Yang et al. (2023). Hence, for any QA pair $(\mathbf{Q}, \mathbf{A}) \sim \mathcal{D}_{a_q}^{q,k}$ in task $k \in \mathcal{T}$, we define the $R_{(\mathbf{Q},\mathbf{A})}^k(\cdot) : \mathbb{R}^{L \times |S|} \to \{0,1\}$ as

$$R_{(\mathbf{Q},\mathbf{A})}^k(\mathbf{o}) = \mathbb{1}\big(\mathbf{o} \in \mathcal{G}_{\mathbf{Q},\mathbf{A}}^{(k)}\big), \tag{1}$$

where $\mathcal{G}_{\mathbf{Q},\mathbf{A}}^{(k)}$ is the deterministic set of *correct* CoTs for $\mathbf{Q}$ within the *valid* CoT collection for task tuple $(q, a, k)$–in practice, *Lean4* only certifies the overall correctness of a CoT trace—providing an outcome reward signal—without verifying individual process reasoning steps.

## 2.2 PRETRAINED BASE MODEL

**Base Model.** For simplicity, following Kim et al. (2025), we model the LLM base model using a straightforward linear softmax predictor:

$$\hat{p}_{\boldsymbol{\theta}}(\cdot|x) = \text{softmax}(\langle \boldsymbol{\theta}, x \rangle), \tag{2}$$

where $\boldsymbol{\theta} \in \mathbb{R}^{|S| \times |S|}$ and $x \in \{0,1\}^{|S|}$ is a one-hot vector. This formulation is theoretically tractable and plausible, as noted by Li et al. (2025b); Ren & Sutherland (2025); Chen et al. (2025), which highlight that the LLM's final layer employs logits $h_{\boldsymbol{\theta}}(\cdot, \boldsymbol{x})$, encoded in the last token, to generate a softmax distribution over the vocabulary as the predictive probability for the next token. Following Kim et al. (2025), we train the base model through entropy loss as below, akin to the next-token prediction process despite in the Markov chain setting.

**Theorem 1** (Informal Version of Thm. 8)**.** *Let $X_0 \sim \text{Unif}(S \setminus S_L)$ and $X_1 \sim \mathbb{P}(\cdot|X_0)$ be random samples from the TMC $X$ in Def. 1, the softmax predictor trained by cross-entropy $L_{CE} = \mathbb{E}_{X_0,X_1}[\log \hat{p}_{\boldsymbol{\theta}^{(t-1)}}(X_1|X_0)]$ via Alg. 1 achieves the following: (1) After $T_1 = \widetilde{O}(M^2)$ iterations, the uniform convergence error of the predictor is $\widetilde{O}(\sqrt{M/T})$. (2) After thresholding, the predictor converges linearly to the true probabilities with error decaying as $\widetilde{O}(e^{-\Omega(T)})$.*

Similar to the treatment in Kim et al. (2025), in the subsequent sections, we suggest that the pretrained $\boldsymbol{\theta}^\star$ achieve the exact transition probability as the TMC model $\hat{p}_{\boldsymbol{\theta}^\star} = \mathbb{P}$ after pretraining. This is plausible given the longer timescales of pretraining relative to finetuning and inference.

## 3 SIMPLICITY BIAS OF RLVR FINETUNING: CHALLENGE AND ANTIDOTE

In this section, we first analyze the inherent simplicity biases of the standard RLVR finetunings, and then provide theoretical justifications for certain strategies that can alleviate this issue. Throughout, the expectation $\mathbb{E}[\cdot]$ is operated on $\boldsymbol{o}_1^i \sim P^k(\mathcal{Q}^k), (\mathbf{Q}, \mathbf{A}) \sim \mathcal{D}_{a_q}^{q,k}, \{\boldsymbol{o}^i\}_{i=2}^G \sim \hat{p}_{\boldsymbol{\theta}^k}^k(O|\boldsymbol{o}_1^i)$.

**REINFORCE** & **RAFT**. In terms of mathematics dataset, the standard REINFORCE objective maximizes the correctness of the sampled CoTs Xiong et al. (2025); Setlur et al. (2025a) (i.e., $R_{(\mathbf{Q},\mathbf{A})}^k(\mathbf{o}) = 1$ in our case). Separately, RAFT (Rejection Sampling Finetuning) Dong et al. (2023); Touvron et al. (2023); Yuan et al. (2023) maximize cross-entropy on successful CoT sampled from current policy. Their objectives in our TMC case are

$$\mathcal{J}_{\text{REINFORCE}}(\boldsymbol{\theta}^k) = \mathbb{E}\left[R_{(\mathbf{Q},\mathbf{A})}^k(\boldsymbol{o})\right], \tag{3}$$

$$\mathcal{J}_{\text{RAFT}}(\boldsymbol{\theta}^k) = \mathbb{E}\left[\sum_{l=1}^{L-1} \log \hat{p}_{\boldsymbol{\theta}^k}(\boldsymbol{o}_{l+1}|\boldsymbol{o}_l) R_{(\mathbf{Q},\mathbf{A})}^k(\boldsymbol{o})\right]. \tag{4}$$

**PPO & GRPO**. Proximal Policy Optimization (PPO) Schulman et al. (2017); OpenAI (2018) and Group Relative Policy Optimization (GRPO) Shao et al. (2024) both optimize clipped surrogate objectives with temperature $\beta > 0$:

$$\mathcal{J}_{\text{PPO}}^k(\boldsymbol{\theta}^k) = \mathbb{E}\left[\frac{1}{L}\sum_{l=1}^{L-1}\min\left(\frac{\hat{p}_{\boldsymbol{\theta}^k}(\boldsymbol{o}_{l+1}^i|\boldsymbol{o}_l^i)}{\hat{p}_{\text{old}}^k(\boldsymbol{o}_{l+1}^i|\boldsymbol{o}_l^i)}A_{l+1}^{\hat{p}_{\boldsymbol{\theta}},k},\text{clip}\left(\frac{\hat{p}_{\boldsymbol{\theta}^k}(\boldsymbol{o}_{l+1}^i|\boldsymbol{o}_l^i)}{\hat{p}_{\text{old}}^k(\boldsymbol{o}_{l+1}^i|\boldsymbol{o}_l^i)},1-\epsilon,1+\epsilon\right)A_{l+1}^{\hat{p}_{\boldsymbol{\theta}},k}\right)\right], \quad (5)$$

$$\mathcal{J}_{\text{GRPO}}^k(\boldsymbol{\theta}^k) = \mathbb{E}\left[\frac{1}{GL}\sum_{i=1,l=1}^{G,L-1}\min\left(\frac{\hat{p}_{\boldsymbol{\theta}^k}(\boldsymbol{o}_{l+1}^i|\boldsymbol{o}_l^i)}{\hat{p}_{\text{old}}^k(\boldsymbol{o}_{l+1}^i|\boldsymbol{o}_l^i)}\hat{A}_{i,l+1}^k,\text{clip}\left(\frac{\hat{p}_{\boldsymbol{\theta}^k}(\boldsymbol{o}_{l+1}^i|\boldsymbol{o}_l^i)}{\hat{p}_{\text{old}}^k(\boldsymbol{o}_{l+1}^i|\boldsymbol{o}_l^i)},1-\epsilon,1+\epsilon\right)\hat{A}_{i,l+1}^k\right)\right] - \beta D_{\text{KL}}[\hat{p}_{\boldsymbol{\theta}^k}||\hat{p}_{\boldsymbol{\theta}^\star}]. \quad (6)$$

Here, the RL advantage Levine (2018) at reasoning step $l$ for task $k$ and predictor $\hat{p}_{\boldsymbol{\theta}}$ is $A_{l+1}^{\hat{p}_{\boldsymbol{\theta}},k}(\boldsymbol{o}_l,\boldsymbol{o}_{l+1}) = Q^{\hat{p}_{\boldsymbol{\theta}},k}(\boldsymbol{o}_l,\boldsymbol{o}_{l+1}) - V^{\hat{p}_{\boldsymbol{\theta}},k}(\boldsymbol{o}_l)$, where $V^{\hat{p}_{\boldsymbol{\theta}},k}(\boldsymbol{o}_l) := \mathbb{E}\left[R_{(\mathbf{Q},\mathbf{A})}^k(\boldsymbol{o})\big|\boldsymbol{o}_l\right]$ and $Q^{\hat{p}_{\boldsymbol{\theta}},k}(\boldsymbol{o}_l,\boldsymbol{o}_{l+1}) := \mathbb{E}\left[V^{\hat{p}_{\boldsymbol{\theta}},k}(\boldsymbol{o}_l)|\boldsymbol{o}_{l+1}\right]$. PPO (5) typically estimates $A_{l+1}^{\hat{p}_{\boldsymbol{\theta}},k}$ via GAE with an additional critic model, while GRPO (6) employs group-normalized advantages $\hat{A}_{i,l+1}^k = (R_{(\mathbf{Q},\mathbf{A})}^k(\boldsymbol{o}^i) - \mu)/\sigma$ computed across sampled CoTs, with $\mu,\sigma$ the group mean and std across $G$ sampled CoTs.

The following theorem shows that the above methods are inherently biased toward easy-to-reason CoTs per **Phenomenon 1** (Wu et al., 2025; Deng et al., 2025), resulting failure over hard instances.

**Theorem 2** (Squeezing Effect of RL-finetuning). *Consider a base model $\boldsymbol{\theta}^\star$ defined in Sec. 2.2 and a targeted task $k \in \mathcal{T}$ with $\Theta(M)$ valid hard-to-reason CoTs. Suppose we apply one of the following finetuning algorithms: REINFORCE, RAFT, PPO, or GRPO (without KL regularization) with access to the expected gradient oracle. For PPO/GRPO, assume the advantage is estimated accurately and the clipping threshold are functioning. Then, for any $\epsilon > 0$, there exists $t \geq \Omega\left(\eta^{-1}L^2M^L\log(ML/\epsilon)\right)$ such that for any valid hard to reason CoT $\boldsymbol{o}^{hard}$ for task $k$, we have*

$$\Pr(\boldsymbol{o}_{2:L}^{hard} \sim \hat{p}_{\boldsymbol{\theta}^{k,(t)}}^k(\cdot|\boldsymbol{o}_1^{hard})) \leq \epsilon.$$

*Therefore, for any $(\mathbf{Q},\mathbf{A})$ of task $k$, if all correct CoTs solving $(\mathbf{Q},\mathbf{A})$ are hard-to-reason, then the finetuned model $\hat{p}_{\boldsymbol{\theta}^{k,(t)}}$ satisfies*

$$\mathbb{E}_{\boldsymbol{o}_{2:L}\sim\hat{p}_{\boldsymbol{\theta}^{k,(t)}}^k(\cdot|\boldsymbol{o}_1)}\left[R_{(\mathbf{Q},\mathbf{A})}^k(\boldsymbol{o})\right] \leq \epsilon.$$

*Sketch of Proof.* The key observation is the following proposition, showing that along any easy-to-reason CoT for a task, the hard-to-learn CoT deviate from it would have smaller advantage.

**Proposition 1** (Advantage Gap between Easy and Hard CoT). *Let $X$ be a Multi-task TMC as in Def. 1 and 2, fix a common task state tuple $(q,a,k)$. Then, for the shared states $\boldsymbol{o}_l, l \in [L-1]$ of any valid easy-to-reason CoT $\boldsymbol{o}^{easy}$ and hard-to-learn CoT $\boldsymbol{o}^{hard}$, then there exists $c > 0$, such that $A_{l+1}^{\hat{p}_{\boldsymbol{\theta}^\star},k}(\boldsymbol{o}_l,\boldsymbol{o}_{l+1}^{easy}) \geq c > A_{l+1}^{\hat{p}_{\boldsymbol{\theta}^\star},k}(\boldsymbol{o}_l,\boldsymbol{o}_{l+1}^{hard}), \forall l \in [L-1].$*

We then denote PO as the algorithm of the PPO/GRPO in Thm. 2, through standard policy gradient derivation, it holds that

$$\nabla_{\boldsymbol{\theta}^k}\mathcal{J}_{\text{REINFORCE}}(\boldsymbol{\theta}^k) = \sum_{l=1}^{L-1}\mathbb{E}[\nabla_{\boldsymbol{\theta}^k}\log\hat{p}_{\boldsymbol{\theta}^k}(\boldsymbol{o}_{l+1}|\boldsymbol{o}_l)R_{(\mathbf{Q},\mathbf{A})}^k(\boldsymbol{o})], \quad (7)$$

$$\nabla_{\boldsymbol{\theta}^k}\mathcal{J}_{\text{RAFT}}(\boldsymbol{\theta}^k) = \sum_{l=1}^{L-1}\mathbb{E}[(1+\log\hat{p}_{\boldsymbol{\theta}^k}(\boldsymbol{o}_{l+1}|\boldsymbol{o}_l))\nabla_{\boldsymbol{\theta}^k}\log\hat{p}_{\boldsymbol{\theta}^k}(\boldsymbol{o}_{l+1}|\boldsymbol{o}_l)R_{(\mathbf{Q},\mathbf{A})}^k(\boldsymbol{o})], \quad (8)$$

$$\nabla_{\boldsymbol{\theta}^k}\mathcal{J}_{\text{PO}}(\boldsymbol{\theta}^k) = \sum_{l=1}^{L-1}\mathbb{E}\left[(1+(2\mathbb{1}(A_{l+1}^{\hat{p}_{\boldsymbol{\theta}^k},k}(\boldsymbol{o}_l,\boldsymbol{o}_{l+1})\geq 0)-1)\epsilon_{\text{clip}})A_{l+1}^{\hat{p}_{\boldsymbol{\theta}^k},k}(\boldsymbol{o}_l,\boldsymbol{o}_{l+1})\nabla_{\boldsymbol{\theta}^k}\log\hat{p}_{\boldsymbol{\theta}^k}(\boldsymbol{o}_{l+1}|\boldsymbol{o}_l)\right], \quad (9)$$

where $\nabla_{\boldsymbol{\theta}^k}\log\hat{p}_{\boldsymbol{\theta}^k}(\boldsymbol{o}_{l+1} \mid \boldsymbol{o}_l) = \boldsymbol{o}_{l+1} - \sum_{\boldsymbol{o}_{l+1}'\in S_{l+1}}\hat{p}_{\boldsymbol{\theta}^k}(\boldsymbol{o}_{l+1}' \mid \boldsymbol{o}_l)\boldsymbol{o}_{l+1}'$ holds in linear case. For valid hard CoTs $\boldsymbol{o}^{\text{hard},i}$ and easy CoTs $\boldsymbol{o}^{\text{easy},j}$ sharing the same $\boldsymbol{o}_l$ but a different $\boldsymbol{o}_{l+1}^{\text{hard},i}, \boldsymbol{o}_{l+1}^{\text{easy},j}$ at $l \in [L-1]$, where $i,j$ are index of hard and easy valid CoTs, the logits update difference under $\mathcal{J}_{\text{REINFORCE}}(\boldsymbol{\theta}^k)$ is

$$\Delta h_{\boldsymbol{\theta}^k}(\boldsymbol{o}_{l+1}^{\text{hard},i},\boldsymbol{o}_l) - \Delta h_{\boldsymbol{\theta}^k}(\boldsymbol{o}_{l+1}^{\text{easy},j},\boldsymbol{o}_l) = \eta[\nabla_{\boldsymbol{\theta}_{\boldsymbol{o}_{l+1}^{\text{hard},i},\boldsymbol{o}_l}^k}\mathcal{J}_{\text{REINFORCE}}(\boldsymbol{\theta}^k) - \nabla_{\boldsymbol{\theta}_{\boldsymbol{o}_{l+1}^{\text{easy},j},\boldsymbol{o}_l}^k}\mathcal{J}_{\text{REINFORCE}}(\boldsymbol{\theta}^k)]$$
$$= \eta[(A^{\hat{p}_{\boldsymbol{\theta}^k},k}(\boldsymbol{o}_l,\boldsymbol{o}_{l+1}^{\text{hard},i}) - A^{\hat{p}_{\boldsymbol{\theta}^k},k}(\boldsymbol{o}_l,\boldsymbol{o}_{l+1}^{\text{easy},j}))$$
$$+ V^{\hat{p}_{\boldsymbol{\theta}^k},k}(\boldsymbol{o}_l)(\hat{p}_{\boldsymbol{\theta}^k}(\boldsymbol{o}_{l+1}^{\text{hard},i}|\boldsymbol{o}_l) - \hat{p}_{\boldsymbol{\theta}^k}(\boldsymbol{o}_{l+1}^{\text{easy},j}|\boldsymbol{o}_l))] < 0.$$

Here, the inequality follows from Prop. 1 together with $\hat{p}_{\boldsymbol{\theta}^k}(\boldsymbol{o}_{l+1}^{\text{hard},i}|\boldsymbol{o}_l) \leq \hat{p}_{\boldsymbol{\theta}^k}(\boldsymbol{o}_{l+1}^{\text{easy},j}|\boldsymbol{o}_l)$. As a result, the ratio $\hat{p}_{\boldsymbol{\theta}^k}(\boldsymbol{o}_{l+1}^{\text{hard},i}|\boldsymbol{o}_l)/\hat{p}_{\boldsymbol{\theta}^k}(\boldsymbol{o}_{l+1}^{\text{easy},j}|\boldsymbol{o}_l)$ strictly decreases after each gradient update. From Eq.(8 and Eq.(9), RAFT's gradient further amplifies this gap through the $(1+\log(p))$ factor, while PPO's update similarly magnifies it via the term $(1+(2\mathbb{1}(A \geq 0)-1)\epsilon_{\text{clip}})A$. By induction, the disparity between easy and hard CoTs compounds over iterations, and the convergence proof then follows directly.

**Antidote 1: Rejection of Easy Questions.** Recent studies Yu et al. (2025); Xiong et al. (2025); Zhang et al. (2025a) show that rejecting instances, where *all* parallelly sampled CoTs are correct,

improves performance. In our setting, such instances correspond to those solvable by already well-learned easy CoTs. By discarding them and retaining only hard CoT correct-only instances, the model gradually shifts its focus toward harder reasoning paths. Formally, we define `RL-rej` as any algorithm in Thm. 2 augmented with rejection: whenever a sampled CoT has probability mass above $M^{-1}(1-\epsilon)$ by the current model, it is discarded. This ensures training emphasizes harder CoTs gradually in the small learning rate regime, prevents collapse into easy ones, and in the end secure all valid CoTs with probability at least $M^{-1}(1-\epsilon)$. We summarize this finding per below.

**Corollary 1** (`RL-rej` Enables Hard-CoT Learning). *Under the identical setting and assumptions of Thm. 2, consider applying* `RL-rej`*. Then, for any $\epsilon \geq 0$, there exists $t \geq \Omega(\eta^{-1} L^2 M^L \log(ML/\epsilon))$ such that for any valid hard to reason CoT $o^{hard}$ for task $k$, we have*

$$\Pr(o_{2:L}^{hard} \sim \hat{p}_{\boldsymbol{\theta}^k,(t)}^k(\cdot | o_1^{hard})) \geq \frac{1-\epsilon}{M}.$$

*Therefore, for any $(\mathbf{Q}, \mathbf{A})$ of task $k$ with $\geq 1$ correct CoTs, the finetuned model $\hat{p}_{\boldsymbol{\theta}^k,(t)}$ satisfies*

$$\mathbb{E}_{o_{2:L} \sim \hat{p}_{\boldsymbol{\theta}^k,(t)}^k(\cdot | o_1)} \left[ R_{(\mathbf{Q},\mathbf{A})}^k(o) \right] \geq \frac{1-\epsilon}{M}.$$

*That is, with $K \geq \frac{M}{1-\epsilon}(\log(\epsilon^{-1}))$, we have pass@K performance no worse than $1 - \epsilon$.*

Notably, after sufficient iterations, the algorithms in Thm. 2 and Cor. 1 concentrate probability mass on valid CoTs of the targeted task up to $\Theta(1 - \varepsilon)$ from its start state. Consequently, the generation probability of CoTs for other tasks sharing some state would be less than $o(\varepsilon)$, eroding cross-task capability. In what follows, we discuss an alternative exploration approach, which in design can preserve such meta-capabilities.

**Antidote 2: KL-regularization.** It is also worth noting that GRPO typically is equipped with a KL regularization term, as in Eq.(6. The formulation of KL-regularized Reinforcement Learning has been noticed as a distribution optimization (Fan et al., 2023; Black et al., 2024; Clark et al., 2024; Uehara et al., 2024; Marion et al., 2024; Kawata et al., 2025). In theory, the solution is a tilted (or Gibbs) distribution (Csiszár, 1975), as characterized below.

**Lemma 1** (Optimal Sampling of GRPO Variants). *For each task $k \in \mathcal{T}$, let $\boldsymbol{\theta}^\star$ denote the pretrained Foundation Model. Then the GRPO induces an optimal step-wise sampling distribution:*

$$\hat{p}_{\boldsymbol{\theta}^k}^{\mathrm{PO}}(o_{l+1} \mid o_l) \propto \hat{p}_{\boldsymbol{\theta}^\star}(o_{l+1} \mid o_l) \cdot \exp\left( \hat{r} \cdot \frac{A_l^k(o_{l+1})}{\beta} \right), \tag{10}$$

*where $\hat{r} \leq \Theta(M)$ and $A_l^k(o_{l+1})$ is the accurate RL advantage.*

Notably, the induced Gibbs distribution is governed by the KL-regularization temperature $\beta > 0$: a larger $\beta$ reduces the gap between CoTs with high and low advantage. The following corollary formalizes this intuition, showing that $\hat{p}_{\boldsymbol{\theta}^k}^{\mathrm{PO}}$ can, in principle, preserve the broad capability.

**Corollary 2** (KL-regularization Enables Hard-CoT learning and Maintain Cross-task Capability). *Consider a base model $\boldsymbol{\theta}^\star$ defined in Sec. 2.2, a targeted task $k \in \mathcal{T}$ and a different task $k' \neq k$, denote $\hat{p}_{\boldsymbol{\theta}^k}^{\mathrm{PO}}$ as the learner in Eq.(10). For any start state $o_1 = q$ of task $k$, suggest the number of CoTs starting from $o_1$ is $N_{o_1}$. Then for any $\epsilon'$ satisfying $1/N_{o_1} > \epsilon' \geq \epsilon > 0$, denote $\hat{p}_{\boldsymbol{\theta}^k,(t)}^k$ as the PPO/GRPO in Thm. 2 with $\epsilon$, then there exists $\beta = \Omega(ML/\log(\epsilon'^{-1}))$, such that*

1. ***Capable of Hard CoTs***: *For instance $(\mathcal{Q}, \mathbf{A})$ with only some hard-to-reason CoTs correct:*

$$\mathbb{E}_{o_{2:L} \sim \hat{p}_{\boldsymbol{\theta}^k}^{\mathrm{PO}}(\cdot | o_1)} \left[ R_{(\mathbf{Q},\mathbf{A})}^k(o) \geq \epsilon' \geq \epsilon \geq \mathbb{E}_{o_{2:L} \sim \hat{p}_{\boldsymbol{\theta}^k,(t)}^k(\cdot | o_1)} \left[ R_{(\mathbf{Q},\mathbf{A})}^k(o) \right].$$

2. ***Preserve Multi-task***: *For instance $(\mathcal{Q}, \mathbf{A})$ belonging to untargeted task $k' \neq k$:*

$$\mathbb{E}_{o_{2:L} \sim \hat{p}_{\boldsymbol{\theta}^k}^{\mathrm{PO}}(\cdot | o_1)} \left[ R_{(\mathbf{Q},\mathbf{A})}^{k'}(o) \right] \geq \epsilon' \geq \epsilon \geq \mathbb{E}_{o_{2:L} \sim \hat{p}_{\boldsymbol{\theta}^k,(t)}^k(\cdot | o_1)} \left[ R_{(\mathbf{Q},\mathbf{A})}^{k'}(o) \right].$$

*The pass@K performance of any task could be adjusted by temperature $\beta$ given $K$ and $\epsilon'$.*

## 4 SIMPLICITY BIAS OF POPULATION REWARD INFERENCE-SCALING: CHALLENGE AND ANTIDOTE

**ORM Mode**. At inference time, the *outcome reward model* (ORM) evaluates entire trajectories via an outcome-level reward $R_{\mathrm{out}}^k(o)$ (e.g., a neural scorer), guiding solution generation through **Best-of-$N$ (BoN) sampling** (Lightman et al., 2023). We define the natural ORM as $R_{\mathrm{out}}^k(o) = \mathbb{E}\left[ R^k(o) \right]$

(i.e., the expectation of instance-level rewards). Statistically, $R_{out}$ is the Bayes-optimal $L^2$ predictor (and the MLE under Gaussian noise). A neural scorer $R_\theta^k(\cdot)$ is then trained to approximate $R_{out}^k(\cdot)$ by $\arg\min_\theta \mathbb{E}\left[(R_\theta^k(\boldsymbol{o}) - R_{out}^k(\boldsymbol{o}))^2\right]$ which under standard conditions converges to $R_{out}^k$.

**PRM Mode**. Instead of outcome-level scoring, the *process reward model* (PRM) provides intermediate rewards along the reasoning trajectory: $R_{pro}^k(\boldsymbol{o}_l) = g(\boldsymbol{o}_1, \ldots, \boldsymbol{o}_l), \quad l \in \{1, \ldots, L\}$, where $g(\cdot)$ estimates step-wise utility Shao et al. (2024); Snell et al. (2024); Wang et al. (2024); Li et al. (2023). PRM can be integrated into structured decoding, e.g., **BoN** Lightman et al. (2023) (selecting top PRM-scoring step) or **Beam Search (BS)** Snell et al. (2024) (augmenting beam scores). Since process-level annotations are costly, most approaches design PRMs heuristically via **likelihood-based estimates**, which predict the expected final correctness given the current prefix:

$$R_{likelihood}^k(\boldsymbol{o}_l) = V^{\hat{p}_{\boldsymbol{\theta}^\star}}(\boldsymbol{o}_l) = \mathbb{E}\left[R^k(\boldsymbol{o}) \,\big|\, \boldsymbol{o}_l\right], \tag{11}$$

for all $\boldsymbol{o}_l \in S_l$. The expectation, which is operated on $\boldsymbol{o}_1 \sim P^k(\mathcal{Q}^k), (\mathbf{Q}, \mathbf{A}) \sim \mathcal{D}_{a_q}^{q,k}, \boldsymbol{o} \sim \hat{p}_{\boldsymbol{\theta}^\star}^k(O|\boldsymbol{o}_1)$, is typically approximated by Monte Carlo rollouts or by training a neural scorer $R_\theta$ with squared loss $\arg\min_\theta \mathbb{E}\left[(R_\theta(\boldsymbol{o}_l) - R_{likelihood}(\boldsymbol{o}_l))^2\right]$.

Indeed, our following theorem shows that the above two "population rewards" (i.e., expectation-based ORM/PRM) check *consistency* instead of *correctness*, per **Phenomenon 2** (Xu et al., 2025).

**Theorem 3** (Failure of Inference-Scaling with ORM/PRM). *Under the setting of Thm. 2, consider the ORM $R_{out}^k(\boldsymbol{o}) = \mathbb{E}[R^k(\boldsymbol{o})]$, the PRM $R_{likelihood}^k(\boldsymbol{o}_l)$ and inference methods: (i) ORM + BoN, (ii) PRM + BoN (step-wise), or (iii) PRM + BS with width $N$ and beam size $B \geq 1$. For any instance $(\mathbf{Q}, \mathbf{A})$ of task $(o_1, a, k)$, suppose all correct CoTs are hard-to-reason and their $\geq 1$ sparse edges diverge from shared states with some valid easy-to-reason CoT. Then, for any $\epsilon > 0$:*

- *If $N \geq \Omega(\log(\epsilon)/\log\left(\frac{M^L - M}{M^L}\right))$, method (i) fails with probability at least $1 - \epsilon$.*

- *If $N \geq \Omega(\log(\epsilon)/\log\left(\frac{M - 1}{M}\right))$, methods (ii) and (iii) fail with probability at least $1 - \epsilon$.*

***Sketch of Proof.*** Our key observation is the following Prop. 2, which reveals that population rewards systematically favor easy CoTs, assigning higher scores to $\boldsymbol{o}^{easy}$ than $\boldsymbol{o}^{hard}$.

**Proposition 2** (Population Rewards Favor Easy CoTs). *Under the same settings as Thm. 3, let $\boldsymbol{o}^{easy}$ be any valid easy-to-reason CoT and $\boldsymbol{o}^{hard}$ any valid hard-to-reason CoT under $(q, a, k)$. Then,*

$$R_{out}^k(\boldsymbol{o}^{easy}) > R_{out}^k(\boldsymbol{o}^{hard}), \quad R_{likelihood}^k(\boldsymbol{o}_l^{easy}) > R_{likelihood}^k(\boldsymbol{o}_l^{hard}), \quad \forall l \in [L], \, \boldsymbol{o}_l^{easy} \neq \boldsymbol{o}_l^{hard}, \, \boldsymbol{o}_{l-1}^{easy} = \boldsymbol{o}_{l-1}^{hard}.$$

***Proof.*** *The first inequality follows from Def. 2(iii): an easy-to-reason CoT has a larger probability of being correct over the distribution, whereas a hard-to-reason CoT, carries higher uncertainty and thus a smaller population-level chance of correctness. The second inequality follows from Prop. 1 by noting that $R_{likelihood}^k(\boldsymbol{o}_l) = A_{l+1}^{\hat{p}_{\boldsymbol{\theta}^\star}, k}(\boldsymbol{o}_{l-1}, \boldsymbol{o}_l) + V^{\hat{p}_{\boldsymbol{\theta}^\star}, k}(\boldsymbol{o}_{l-1}).$* $\square$

Given Prop. 2, the remaining proofs for Thm. 3 follow by choosing $N$ sufficiently large so that $\boldsymbol{o}^{easy}$ (or $\boldsymbol{o}_l^{easy}$) is sampled at least once across the $N$ parallel trials.

**Antidote: Gibbs Sampling.** Soft Best-of-N sampling (Soft-BoN) (Verdun et al., 2025) is designed to approximate the gibbs distribution $P_{Gibbs}^k(\boldsymbol{o})$ with $O(N^{-1})$ error, which is defined as

$$P_{Gibbs}^k(\boldsymbol{o}) \propto \left(\hat{p}_{\boldsymbol{\theta}^\star}(\boldsymbol{o}) \exp\left(\lambda R_{out}^k(\boldsymbol{o})\right)\right), \tag{12}$$

for $o_1 = q \sim P^k(\mathcal{Q}_k)$. Akin to Eq.(10), the distribution $P_{Gibbs}^k(\boldsymbol{o})$ also can control the trade-off between reward maximization and the divergence from the base model's predictive power.

**Corollary 3.** *(Csiszár, 1975) Consider a base model $\boldsymbol{\theta}^\star$ defined in Sec. 2.2, a targeted task $k \in \mathcal{T}$ and ORM $R_{out}^k(\boldsymbol{o}) = \mathbb{E}[R^k(\boldsymbol{o})]$. For $\lambda > 0$, the Eq.(12) is the solution of:*

$$\max_{P_{new}^k} \mathbb{E}_{P_{new}^k}[R_{out}^k(\boldsymbol{o})] - \frac{1}{\lambda} D_{KL}(P_{new}^k \| \hat{p}_{\boldsymbol{\theta}^\star}). \tag{13}$$

Indeed, through the statistical merit of Doob's h-transform techniques Uehara et al. (2024); Kawata et al. (2025); Rogers & Williams (2000); Chopin et al. (2023); Heng et al. (2024), we provably show that there is a principled framework to design process reward, which could mathematically generate the same CoT distribution as Eq.(12) with a temperature $\lambda > 0$.

**Definition 3.** *Doob's $h$-Transform-induced Process Reward Model (DPRM). Consider a base model $\boldsymbol{\theta}^\star$ defined in Sec. 2.2, a targeted task $k \in \mathcal{T}$ and ORM $R_{\text{out}}^k(\boldsymbol{o}) = \mathbb{E}[R^k(\boldsymbol{o})]$. The DPRM Adjuested Sampling (DPRM-AS) defines the process reward at step $l$ via harmonic function $h_k(\cdot)$, and sample according to step-wise distribution adjustment with $\lambda > 0$:*

$$R_{\text{DPRM}}^k(\boldsymbol{o}_l) = \frac{1}{\lambda} \log h_k(\boldsymbol{o}_l), \quad \text{where} \quad h_k(\boldsymbol{o}_l) = \mathbb{E}_{\boldsymbol{o}_{l+1:L} \sim \hat{p}_{\boldsymbol{\theta}^\star}} \left[ \exp\left(\lambda R_{\text{out}}^k(\boldsymbol{o})\right) \mid \boldsymbol{o}_l \right],$$

$$\hat{p}_{\boldsymbol{\theta}}^{new,k}(\boldsymbol{o}_{l+1} \mid \boldsymbol{o}_l) = \hat{p}_{\boldsymbol{\theta}^\star}(\boldsymbol{o}_{l+1} \mid \boldsymbol{o}_l) \cdot \frac{h_k(\boldsymbol{o}_{l+1})}{h_k(\boldsymbol{o}_l)} \propto \hat{p}_{\boldsymbol{\theta}}(\boldsymbol{o}_{l+1} \mid \boldsymbol{o}_l) \exp(\lambda R_{\text{DPRM}}^k(\boldsymbol{o}_{l+1})),$$

(14)

*where the first-step is initialized as $\hat{p}_{\boldsymbol{\theta}}^{new,k}(\boldsymbol{o}_1 \mid \boldsymbol{o}_0) := \mathbb{E}_{\boldsymbol{o}_1 \sim P^k(\mathcal{Q}_k)}[\exp(\lambda R_{\text{DPRM}}^k(\boldsymbol{o}_1))]$. Then, the induced distribution $P_{\text{DPRM}}^k(\boldsymbol{o}) := \prod_{l=0}^{L-1} \hat{p}_{\boldsymbol{\theta}}^{new,k}(o_{l+1}|o_l))$ satisfies $P_{\text{DPRM}}^k(\boldsymbol{o}) = P_{\text{Gibbs}}^k(\boldsymbol{o})$.*

**Remark 4.** *By $P_{\text{DPRM}}^k(\boldsymbol{o}) = P_{\text{Gibbs}}^k(\boldsymbol{o})$ as shown above, the Soft-BoN method is a realization of the induced distribution, with a convergence rate $O(N^{-1})$ (Verdun et al., 2025). Notably, $R_{\text{DPRM}}^k(\cdot)$ and heuristic $R_{\text{likelihood}}^k(\cdot)$ are both state-conditioned expectations with **equivalent estimation cost**.*

Inherently, we provably show the asymptotic equivalence between DPRM and heuristically-designed PRM $R_{\text{likelihood}}^k(\cdot)$ with certain sampling strategies, which we formalized as below.

**Corollary 4.** *Under the same settings as Def. 3, for $0 < \lambda < \infty$, it holds that **BoN/BS** with $R_{\text{DPRM}}^k(\boldsymbol{o}_l)$ is equivalent to **BoN/BS** with $R_{\text{likelihood}}^k(\boldsymbol{o}_l)$.*

***Sketch of Proof.*** The key observation is by the monotonicity of $\exp(\cdot)$ and $\log(\cdot)$, it holds that

$$\arg\max_{\boldsymbol{o}_l \in S_l^{\text{BoN}}} R_{\text{DPRM}}^k(\boldsymbol{o}_l) = \arg\max_{\boldsymbol{o}_l \in S_l^{\text{BoN}}} R_{\text{likelihood}}^k(\boldsymbol{o}_l),$$

where $S_l^{\text{BoN}} = \{\boldsymbol{o}_l^1, \dots, \boldsymbol{o}_l^N\}$ is the set of **BoN** candidates.

Through the similar techniques in Cor. 2, one can also show that $P_{\text{DPRM}}^k(\boldsymbol{o}) = P_{\text{Gibbs}}^k(\boldsymbol{o})$ preserved the broad capability in principle, as below.

**Corollary 5** (Gibbs Distribution Preserves Meta-Capability). *Under the same settings in Cor. 2, for any $\epsilon'$ satisfying $1/N_{o_1} > \epsilon' \geq \epsilon > 0$. Then there exists $\lambda = O(\log(\epsilon'^{-1})/ML)$, denote $\hat{p}_{\text{IS}}^k(\cdot)$ as any of the inference predictors (i)-(iii) in Cor. 2 with $N \geq \Omega(\log(\epsilon)/\log\left(\frac{M^L - M}{M^L}\right))$, it holds that*

1. ***Capable of Hard CoTs***: $\mathbb{E}_{\boldsymbol{o} \sim P_{\text{Gibbs}}^k(\boldsymbol{o})} \left[ R_{(\mathbf{Q}, \mathbf{A})}^k(\boldsymbol{o}) \right] \geq \epsilon' \geq \epsilon \geq \mathbb{E}_{\boldsymbol{o} \sim \hat{p}_{\text{IS}}^k(\boldsymbol{o})} \left[ R_{(\mathbf{Q}, \mathbf{A})}^k(\boldsymbol{o}) \right]$.

2. ***Preserve Multi-task***: $\mathbb{E}_{\boldsymbol{o} \sim P_{\text{Gibbs}}^k(\boldsymbol{o})} \left[ R_{(\mathbf{Q}, \mathbf{A})}^{k'}(\boldsymbol{o}) \right] \geq \epsilon' \geq \epsilon \geq \mathbb{E}_{\boldsymbol{o} \sim \hat{p}_{\text{IS}}^k(\boldsymbol{o})} \left[ R_{(\mathbf{Q}, \mathbf{A})}^k(\boldsymbol{o}) \right]$.

The pass@K of Soft-BoN ($o(\frac{1}{N})$ error to gibbs sampling (Verdun et al., 2025)) for any task could then be adjusted by temperature $\beta$ given $K$ and $\epsilon'$.

## 5 EMPIRICAL SIMULATIONS

To validate our theoretical findings, we run simulations on an abstract Tree-structured Markov Chain (TMC) with two tasks (**TASK 1** is the target), as shown in Tab. 1 below.

Table 1: Task definition and CoTs characteristics in our Multi-Task TMC simulation.

| Task | Path Index | State Transition | Type | Probability | Expected Correctness over $\mathcal{D}_a^{q,k}$ |
|---|---|---|---|---|---|
| TASK1 | 0 | $S_1[0] \to S_2[0] \to S_3[0] \to S_4[0]$ | **EASY**-To-REASON | 0.413223 | 0.727995 |
| | 1 | $S_1[0] \to S_2[0] \to S_3[1] \to S_4[0]$ | HARD-To-REASON | 0.075131 | 0.132363 |
| | 2 | $S_1[0] \to S_2[1] \to S_3[0] \to S_4[0]$ | HARD-To-REASON | 0.004132 | 0.007280 |
| | 3 | $S_1[0] \to S_2[1] \to S_3[1] \to S_4[0]$ | HARD-To-REASON | 0.075131 | 0.132363 |
| TASK2 | 0 | $S_1[1] \to S_2[0] \to S_3[0] \to S_4[1]$ | **EASY**-To-REASON | 0.413223 | 0.955691 |
| | 1 | $S_1[1] \to S_2[0] \to S_3[1] \to S_4[1]$ | HARD-To-REASON | 0.007513 | 0.017376 |
| | 2 | $S_1[1] \to S_2[1] \to S_3[0] \to S_4[1]$ | HARD-To-REASON | 0.004132 | 0.009557 |
| | 3 | $S_1[1] \to S_2[1] \to S_3[1] \to S_4[1]$ | HARD-To-REASON | 0.007513 | 0.017376 |

The TMC has $L = 4$ layers with two nodes each ($|S_l| = 2$). In layers 1–3, each state has one high-probability outgoing edge. Pretraining runs for $T_1 = 2000$ and $T_2 = 500$ steps (error $< 0.001$); fine-tuning for $T = 1000$ steps with learning rate $0.05$. Estimation of the Rewards/advantages use $1000/200$ Monte Carlo samples; temperature $\lambda = 0.5$; BoN uses $N = 15$. Training and testing each use 200 question instances sampled per Def. 2; BoN and Gibbs-style methods are fully enumerated. **TASK 1** requires reaching $S_4[0]$ from $S_1[0]$; **TASK 2** requires $S_4[1]$ from $S_1[1]$. A CoT is valid here

Table 2: CoT generation statistics for different strategies on TASK1 and TASK2. Values represent percentages of valid easy CoTs, valid hard CoTs, and invalid CoTs generated by each method.

| Strategy | TASK1 Valid Easy CoTs (%) | TASK1 Valid Hard CoTs (%) | TASK1 Invalid CoTs (%) | TASK2 Valid Easy CoTs (%) | TASK2 Valid Hard CoTs (%) | TASK2 Invalid CoTs (%) |
|---|---|---|---|---|---|---|
| **Base Model** | 21.67% | 8.07% | 70.27% | 20.03% | 1.10% | 78.87% |
| **Finetuned Methods** | | | | | | |
| REINFORCE | **94.33%** | 3.43% | 2.23% | **1.52%** | 0.87% | 97.62% |
| RAFT | **95.22%** | 2.33% | 2.45% | **2.30%** | 0.92% | 96.78% |
| PPO (Eq.11) | **91.82%** | 5.40% | 2.78% | **2.23%** | 1.03% | 96.73% |
| RL-rej (Sec.3.1) | 49.62% | **17.42%** | 32.97% | 30.63% | 2.27% | 67.10% |
| GRPO-KL (Eq.13) | 46.47% | **16.27%** | 37.27% | 54.18% | 1.68% | 44.13% |
| **Inference Scaling Methods** | | | | | | |
| Soft-BoN | 8.98% | **19.30%** | 71.72% | 7.00% | 17.27% | 75.73% |
| ORM-BoN w. $R_{\text{out}}^k(\cdot)$ | 21.00% | 7.30% | 71.70% | 20.23% | 0.97% | 78.80% |
| PRM-BoN w. $R_{\text{likelihood}}^k(\cdot)$ | 99.13% | 0.87% | 0.00% | 13.42% | 36.77% | 49.82% |
| DPRM-BoN | 99.52% | 0.48% | 0.00% | 13.40% | 37.02% | 49.58% |
| DPRM-AS (implemented by step-wise Soft-BoN) | 17.23% | **36.10%** | 46.67% | 12.02% | 38.02% | 49.97% |

if it connects the start and end states; among valid paths, only the one via $S_2[0] \to S_3[0]$ is *easy*, all others are *hard*. We report the proportions of easy, hard, and invalid CoTs from $S_1[0]$ (TASK 1) and $S_1[1]$ (TASK 2), as well as the expected correctness over the population per Def. 2.

**Findings in Tab. 2.** REINFORCE, RAFT, and PPO heavily favor easy-to-reason CoTs in TASK 1, suppressing hard-to-reason CoTs in TASK 1 and valid CoTs in TASK 2, showing clear *simplicity bias* and *forgetting*. In contrast, diversity-promoting methods (RL-rej, GRPO-KL, Soft-BoN, DPRM-AS) balance easy/hard-to-reason CoTs in TASK 1 and preserve TASK 2 CoT's generation capability, thanks to shared sparse edges in the TMC (two nodes per layer, see Table 1). ORM/PRM-BoN, relying on population rewards $R_{\text{out}}^k(\cdot)$, also overfavor easy-to-reason CoTs; PRM-BoN and DPRM-BoN behave similarly, as predicted. Further details are available in App. F.

Table 3: Theoretical comparison between RLVR and inference-scaling under our TMC setting. The first column indicates pass@K performance. The second and third columns assess whether a method assigns highest credit to easy-to-reason CoTs and whether it can also sample hard-to-reason CoTs with suitable temperature. The fourth column evaluates whether the method preserves the base model's multi-task capability. The results suggest that post-training methods tend to favor easy-to-reason CoTs, and that only methods capable of sampling hard-to-reason CoTs can achieve satisfactory pass@K.

| Methods | Succeed w. pass@K | Prefer Easy CoT | Capable of Hard CoT | Preserve Multi-task |
|---|---|---|---|---|
| REINFORCE (Eq.(3) / RAFT (Eq.(4) | ✗ | ✓ | ✗ | ✗ |
| PPO (Eq.(5) | ✗ | ✓ | ✗ | ✗ |
| RL-rej (Sec 3) | ✓ | ✓ | ✓ | ✗ |
| KL-regularized PO (Eq.(6) | ✓ | ✓ | ✓ | ✓ |
| ORM/PRM-BoN/BS (Sec 4) | ✗ | ✓ | ✗ | ✗ |
| Soft-BoN/DPRM-AS (Sec 4) | ✓ | ✓ | ✓ | ✓ |

# 6 CONCLUSION, LIMITATIONS, AND FUTURE WORK

We introduced a Tree-structured Markov toy framework to model foundation model's diverse multi-task reasoning patterns, and theoretically shows that both RLVR and inference-scaling exhibit a *simplicity bias*, favoring easier, common reasoning paths (consistency) rather than true correctness (Wang et al., 2022). Building on this, we demonstrated the benefit of exploration within the tree—mitigating this bias and preserving rare but crucial CoTs—as summarized in Tab. 3. Our analysis further highlights a sharp contrast with traditional RL (e.g., AlphaGo (Silver et al., 2016)): whereas RL advantages promote effective state-space exploration in standard RL, in post-training they instead push models to overemphasize easy (high-pass-rate) paths within the model's scope (Yue et al., 2025). This negative insight may also explain why Setlur et al. (2025a) employ independent models that reinterpret RL advantage differently for finetuning and PRM scoring.

Inside our proofs, the central clue lies in the expectation (population)-based reward estimators, namely $R_{\text{out}}^k(\boldsymbol{o}) = \mathbb{E}[R^k(\boldsymbol{o})]$ and $R_{\text{likelihood}}^k(\boldsymbol{o}_l) = V^{\hat{p}_{\boldsymbol{\theta}^\star}}(\boldsymbol{o}_l) = \mathbb{E}[R^k(\boldsymbol{o}) \mid \boldsymbol{o}_l]$. While these estimators are Bayes-optimal in the $L^2$ sense, they inherently favor frequent patterns, thereby down-weighting rare-but-valuable CoTs. This bias highlights the necessity of more reliable reward designs, as also discovered by Xu et al. (2025). Also, our current TMC framework is deliberately abstract and restrictive (see App. E and G.4), and could be generalized to more realistic reasoning models with variable depths. Another promising direction is to apply TMC analysis to reflective behavior and *aha* moments (Yu et al., 2025).

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

# Appendix

## TABLE OF CONTENTS

## A  ETHICS STATEMENT

This work adheres to the ICLR Code of Ethics. Our research involves the development and evaluation of inference scaling methods for large language models, which does not involve human subjects or sensitive data collection. All experiments are conducted in our toy settings with clear conditions and hyperparameters. Our methods aim to democratize access to advanced AI capabilities by reducing computational requirements, which we believe has positive societal implications. We have no conflicts of interest to declare, and this work was conducted independently without external sponsorship that could influence our findings.

## B  REPRODUCIBILITY STATEMENT

To ensure reproducibility of our work, we provide comprehensive details throughout the paper. All experimental configurations, hyperparameters, and implementation details are specified in the main text and appendix. Our code implementation will be made available as supplementary material, including the complete training scripts, evaluation protocols, and data processing pipelines. For the theoretical analysis, we provide complete proofs and derivations in the appendix. All experimental results can be reproduced using the provided code and configuration files. We have also included detailed computational resource requirements and training time estimates to facilitate reproduction of our results.

## C  LLM USAGE

We used a large language model (LLM) as a writing assistant to polish the English presentation of certain paragraphs. The LLM was not involved in research ideation, technical derivations, or experimental design. All scientific contributions and claims are solely by the authors.

## D  ADDITIONAL RELATED WORK

**LLMs as Markov Processes.** A growing body of work has drawn connections between large language models (LLMs) and Markovian dynamics. Zekri et al. (2024) established a theoretical equivalence between next-token prediction in LLMs and finite-state Markov chains, deriving scaling laws for in-context learning when prompted with such chains. Nichani et al. (2024) demonstrated that disentangled transformers are capable of learning Markov chains in context. Ildiz et al. (2024) studied how a single self-attention layer can simulate context-conditioned Markov chains, while Ding et al. (2025) showed that multi-layer transformers can approximate preconditioned gradient descent over Markovian distributions. Edelman et al. (2024) analyzed the distinct phases of training as transformers learn Markov chains, and Makkuva et al. (2024) investigated the function landscape of single-layer transformers on Markovian data, revealing challenges in learning higher-order chains. Rajaraman et al. (2024) proved that constant-depth transformers can learn $k$-order Markov processes when the next-token distribution depends on the previous $k$ tokens. Furthermore, Cao et al. (2025) showed that transformers can simulate the maximum likelihood estimation (MLE) algorithm for learning Bayesian networks, which subsume Markov chains as a special case. Despite these advances, most prior works focus on modeling sequential variable dependencies, without abstracting the structure to chain-of-thought (CoT) reasoning. The most relevant exception is the recent work of Kim et al. (2025), which investigates CoT processes under metastable Markov chain assumptions. They show the necessity of search, RL-based finetuning, and distillation to navigate sparse transition spaces, also under a softmax modeling assumption. Their proposed algorithm is tailored specifically for such metastable settings. Instead, motivated by real-world multi-task, tree-structured reasoning tasks with binary (0-1) rewards, our work aims to theoretically compare the intrinsic biases of RL-based finetuning and inference sampling, and to connect these with recent discussions on the squeezing effect, the benefits of reasoning diversity, and the inherent limitations of RL-based fine-tuning.

**Spectral Bias.** The study of spectral bias in deep learning is extensive, with many works showing that neural networks tend to learn low-frequency or simple patterns with high signal-to-noise ratio first (Arpit et al., 2017; Valle-Perez et al., 2018; Kalimeris et al., 2019; Chen et al., 2023a; Abbe

et al., 2023; Molina et al., 2024). Edelman et al. (2024) demonstrated that this simplicity bias during training can delay convergence to the correct solution in Markov chain learning. Chen et al. (2023b) observed that shallow layers in neural networks prioritize fitting lower-order functions, while Allen-Zhu & Li (2023) showed that this tendency in shallow networks can lead to drastically increased sample complexity due to their bias toward low-order polynomials. Tian (2024) examined simplicity bias from the perspective of algebraic structure learning. Other works have highlighted potential downsides: Shah et al. (2020); Yang et al. (2024) showed that such biases can be detrimental, causing models to overlook important features or be misled by spurious correlations. Recent work Ren & Sutherland (2025) identified the *squeezing effect* of Direct Preference Optimization: probability mass becomes increasingly concentrated on the outut that was most confident prior to the update. A following-up work Deng et al. (2025) identified similar phenomenon of GRPO. Separately, Li et al. (2025b) analyzes the nature of the cross-entropy loss, showing that it systematically shifts probability mass from non-target tokens to target tokens—regardless of the quality of the non-target options—ultimately leading to distribution collapse during finetuning. However, prior work has not systematically characterized how this squeezing effect influences fine-tuning dynamics. In our study, under the Tree-structured Markov Chain (TMC) framework and a linear softmax model, we show that binary outcome rewards can potentially amplify this effect, favoring simple reasoning paths during fine-tuning and contributing to the model's inductive bias.

**Process Reward Models (PRMs) & Reinforcement Learning with Verifiable Rewards (RLVR).** Process Reward Models (PRMs) and Reinforcement Learning with Verifiable Rewards (RLVR) both employ external verifiers to reward reasoning steps, with PRMs guiding inference Lightman et al. (2023); Li et al. (2023); Snell et al. (2024) and RLVR enhancing finetuning Wang et al. (2025); Foster et al. (2025). Setlur et al. (2025b) show that verifier-based scaling outperforms verifier-free approaches when the reward distributions have anti-concentration and heterogeneity properties. Foster et al. (2025) also analyzed on linear softmax model, for which they designed an algorithm that is computationally efficient, and showed the necessity of coverage within their framework. Yue et al. (2025) find RLVR's gains limited to small $k$, with base models matching or surpassing it at large $k$, suggesting RLVR reinforces existing reasoning rather than fostering new patterns—echoing our finding that RL finetuning overfits to simpler paths due to the squeezing effect. Schmied et al. (2025) highlight RLVR's "greediness", favoring easy actions akin to our findings, while Yu et al. (2025)'s DAPO and Xiong et al. (2025)'s minimalist approaches counter this by rejecting overly-correct samples, promoting diverse reasoning and keeping steady entropy, whose merits are also theoretically justified in our settings. Wang et al. (2025) also empirically showed the critical role of promoting exploration with diverse reasoning patterns. Setlur et al. (2025a) propose a separate prover policy to enhance exploration, noting the base model's advantage calculation limits diversity—supporting our observation of RLVR's bias toward simpler paths. Li et al. (2025b) add that cross-entropy finetuning reduces sampling diversity, reinforcing the need for varied inference strategies. These findings collectively underscore the value of diverse reasoning, motivating our comparison of RL and PRM under binary outcome rewards.

# E  LIMITATIONS AND BROADER IMPACT

**Unmodeled Complexity in Large-Scale**. While our theoretical analysis introduces new perspectives on finetuning and inference-scaling under binary (0–1) outcome supervision, several limitations remain. First, the latent reasoning model and neural formulation may require further refinement to better align with practical scenarios, including: handling *varying reasoning depths*; incorporating *structural priors* (e.g., multi-index models); modeling with nonlinear transformers instead of a linear softmax model (per discussed in App. G.4); and analyzing parameter-efficient tuning methods like LoRA Hu et al. (2021).

**Reward Hacking, and the benefit of consistency**. Even within the TMC framework, our formulation does not fully capture challenges such as robustness to noisy rewards, hallucinations, or reward hacking. For example, in Fig. 1, the trajectories $q \rightarrow o_2^1 \rightarrow a_3$ (valid for Task 4) and $q \rightarrow o_2^2 \rightarrow a_3$ (valid for Task 5) share the same endpoints but are invalid for each other's task, illustrating a form of reward misalignment or *hallucination*. This warrants deeper investigation. A concurrent study by Wen et al. (2025) raised a concern: rather than rewarding rare reasoning paths, they classified them as incorrect CoTs and treated common paths as *logically coherent*-which they assumed correct. They further advocated for stronger verifiers and new RLVR algorithms explicitly designed to

incentivize correct reasoning paths—a perspective we share. In our Multi-task TMC (Def. 2), our "validity" notion is to distinguish in-correct rare paths for a task with those correct ones. We left a more detailed discussions of the pros and cons of the simplicity bias an important future direction.

**Faster-vs-Better Trade-Off.** Moreover, although our results highlight the value of diversity—particularly when a non-negligible fraction of instances require hard-to-reason CoTs—our analysis does not quantify the additional computational cost such diversity induces. This reflects an inherent tradeoff: overfitting to simpler reasoning paths enables faster finetuning when the target is improving overall accuracy within certain iterations, while supporting diverse reasoning incurs greater complexity—a "no free lunch" scenario.

**Non-Markovianity of LLM Reasoning**. Markov-chain (MC) abstractions—where transition probabilities encode step difficulty—are well-established in prior theory (Xu et al., 2019; Sanford et al., 2024; Abbe et al., 2024; Besta et al., 2024; Kim et al., 2025). In particular, Kim et al. (2025) model LLM inference as a metastable MC and design algorithms showing benefits of search and distillation. Building on empirical evidence of tree-alike reasoning (Lightman et al., 2023; Snell et al., 2024; Yue et al., 2025; AI et al., 2025; Gandhi et al., 2025), and observed real-world hardness metrics (base-model *pass rates* Tong et al. (2024)), our Multi-task TMC is arguably more aligned with practice than prior work. We acknowledge that MC models cannot perfectly capture actual LLM inference, per Zhang et al. (2025b) on LLM non-Markovianity. Nonetheless, this does not diminish the value of MC-based theories: conclusions remain informative and can often be generalized to non-Markovian settings with suitable extensions.

While our findings are theoretical, they provide high-level justification for recent empirical efforts that promote reasoning diversity and reject overly easy instances, offering useful insights for future work on RL fine-tuning, PRM design, and inference strategies in LLMs. We do not anticipate any direct societal risks arising from this research.

# F ADDITIONAL EXPERIMENTS

## F.1 COMPREHENSIVE PERFORMANCE AND COVERAGE ANALYSIS

Building upon the empirical simulations presented in Section 5, we provide additional experimental results that further validate our theoretical findings. The following analysis examines both performance metrics (Pass@K rates) and coverage characteristics (valid CoT generation patterns) across different sampling strategies for both TASK1 and TASK2.

### F.1.1 PERFORMANCE ANALYSIS

Figure 2 and Figure 3 present the Pass@30 performance for TASK1 and TASK2, respectively, across all evaluated sampling strategies. The results demonstrate several key patterns that align with our theoretical predictions:

**TASK1 Performance:** The performance across different strategies shows relatively consistent results, with Pass@30 rates ranging from 0.65 to 0.73. Notably, DPRM achieves the highest performance (0.73), followed closely by Reinforce-rej (e.g. `RL-rej`) and GRPO-KL (both at 0.72). The base model performs moderately well (0.71), while PRM-BoN shows the lowest performance (0.65). This suggests that while most strategies can achieve reasonable performance on the primary task, there are meaningful differences in their effectiveness.

**TASK2 Performance:** The results reveal a stark contrast, with performance ranging from 0.35 to 0.95. The base model and diversity-promoting methods (Reinforce-rej (e.g. `RL-rej`), GRPO-KL) achieve the highest performance (0.95), demonstrating their ability to maintain capability on secondary tasks. In contrast, standard RL fine-tuning methods (REINFORCE, RAFT, PPO) show significantly degraded performance (0.35-0.48), confirming the *forgetting* phenomenon predicted by our theoretical analysis.

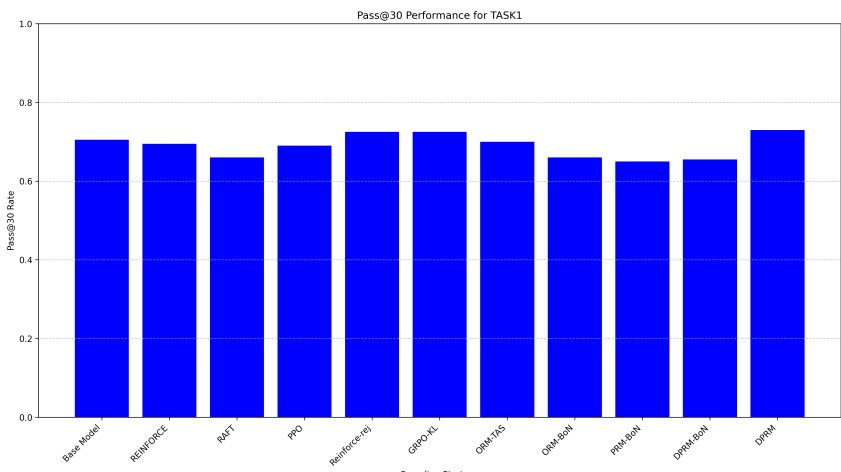

Figure 2: Pass@30 Performance for TASK1 across different sampling strategies. The results show relatively consistent performance across most methods, with DPRM achieving the highest rate of 0.73.

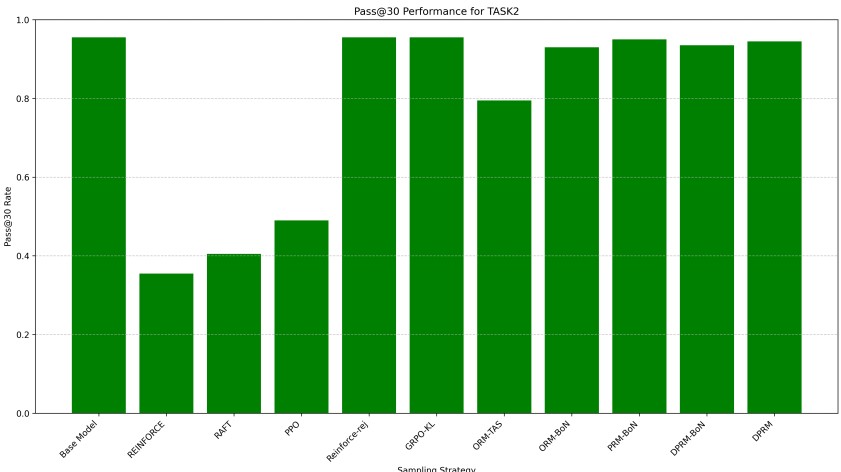

Figure 3: Pass@30 Performance for TASK2 across different sampling strategies. The results demonstrate significant performance degradation for standard RL methods (REINFORCE, RAFT, PPO) compared to diversity-promoting approaches, confirming the forgetting phenomenon.

### F.1.2 COVERAGE ANALYSIS

The coverage analysis, presented in Figure 4 and Figure 5, provides insights into the types of CoTs generated by each strategy. These stacked bar charts show the proportion of invalid, hard valid, and easy valid CoTs generated by each method.

**TASK1 Coverage:** The results reveal distinct patterns across different strategy categories. Standard RL fine-tuning methods (REINFORCE, RAFT, PPO) and PRM-based methods (PRM-BoN, DPRM-BoN) generate predominantly easy valid CoTs (90-98%) with minimal invalid CoTs, demonstrating strong *simplicity bias*. In contrast, diversity-promoting methods (Reinforce-rej (e.g. `RL-rej`), GRPO-KL) show a more balanced distribution, with substantial proportions of both easy and hard valid CoTs. The base model and ORM-based methods generate a high proportion of invalid CoTs (70-72%), indicating limited effectiveness in generating task-appropriate reasoning paths.

**TASK2 Coverage:** The coverage patterns for TASK2 are markedly different, reflecting the task's increased difficulty. Most strategies generate a high proportion of invalid CoTs, with standard RL methods showing particularly poor performance (97-98% invalid). However, diversity-promoting methods (GRPO-KL, PRM-BoN, DPRM-BoN, DPRM) achieve significantly better coverage, with

45-55% valid CoTs. This demonstrates the importance of diversity-promoting mechanisms for maintaining capability across multiple tasks.

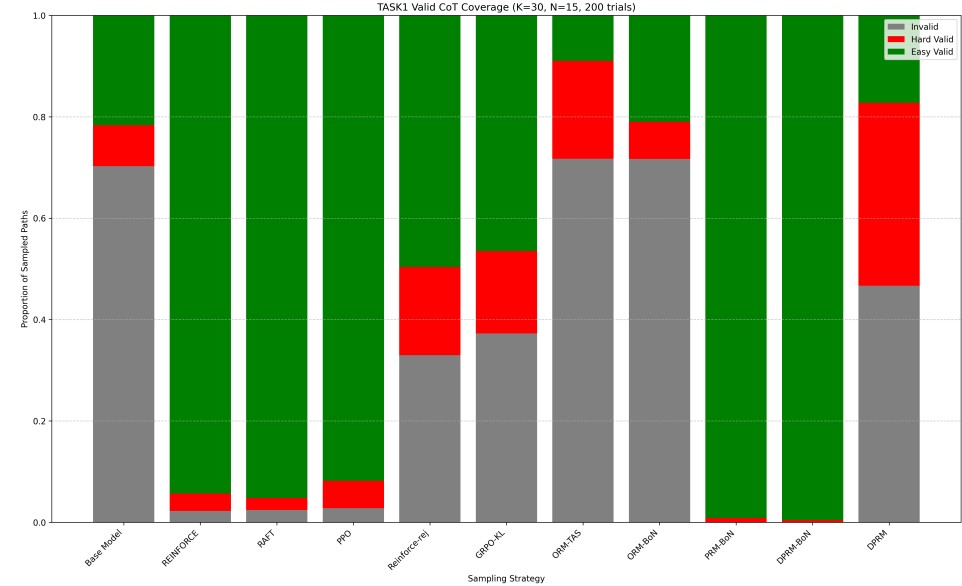

Figure 4: Valid CoT Coverage for TASK1 (K=30, N=15, 200 trials). The stacked bars show the proportion of invalid (gray), hard valid (red), and easy valid (green) CoTs generated by each strategy. Standard RL methods show strong simplicity bias with predominantly easy valid CoTs.

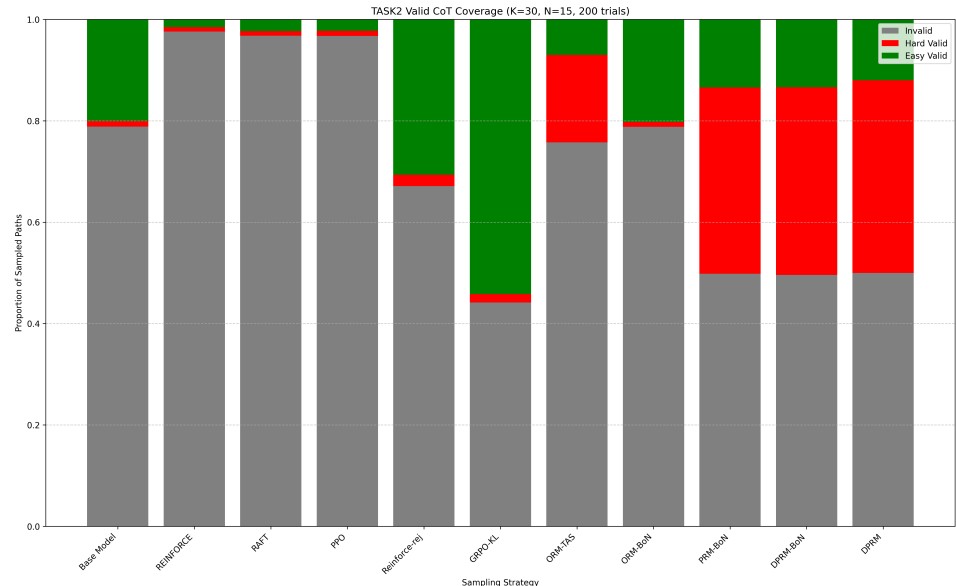

Figure 5: Valid CoT Coverage for TASK2 (K=30, N=15, 200 trials). The stacked bars show the proportion of invalid (gray), hard valid (red), and easy valid (green) CoTs generated by each strategy. Diversity-promoting methods achieve significantly better coverage compared to standard RL approaches.

### F.1.3 KEY INSIGHTS AND IMPLICATIONS

These comprehensive results provide several important insights that extend our theoretical analysis:

**Simplicity Bias Confirmation:** The coverage analysis clearly demonstrates the *simplicity bias* in standard RL fine-tuning methods, which overwhelmingly favor easy-to-reason CoTs while sup-

Table 4: Summary of Notations

| Notation | Description |
|---|---|
| $S_l, S$ | State space at layer $l$; $S = \bigcup_{l=1}^{L} S_l$ is the full state space. |
| $o, O, \boldsymbol{o}, o_l, \boldsymbol{o}_l$ | $O$ denotes a trajectory; $\boldsymbol{o}$ its one-hot form; $o_l, \boldsymbol{o}_l$ are step-$l$ token and embedding. |
| $\mathbb{P}(\cdot|\cdot)$ or $\mathbb{P}_{\mathrm{TMC}}(\cdot|\cdot)$, $\hat{p}_{\boldsymbol{\theta}}(\cdot|\cdot)$ | $\mathbb{P}(\cdot|\cdot)$ or $\mathbb{P}_{\mathrm{TMC}}(\cdot|\cdot)$: TMC kernel in Def. 1; $\hat{p}_{\boldsymbol{\theta}}(\cdot|\cdot)$: softmax predictor based on $\boldsymbol{\theta}$. |
| $C_{o_l}, D_{o_l}$ | High probability transition subset in $S_{l+1}$; Non-zero probability transition subset. |
| $\mathcal{Q}_k, (q, a_q, k)$ | $\mathcal{Q}_k \subseteq S_1$: question states in task $k$; $(q, a_q, k)$: task tuple with $q \mapsto a_q$. |
| $\mathcal{D}_{a_q}^{q,k}, \mathcal{G}_{\mathbf{Q},\mathbf{A}}^{(k)}$ | Instance Distribution over task tuple $(q, a_q, k)$; Correct CoTs for $(\mathbf{Q}, \mathbf{A}) \sim \mathcal{D}_{a_q}^{q,k}$. |
| $\mathcal{G}_{q,a_q}^{(k)}, \mathcal{G}_{q,a_q}^{(k),\mathrm{easy}}, \mathcal{G}_{q,a_q}^{(k),\mathrm{hard}}$ | Valid CoTs set for $(q, a_q, k)$; partitioned into easy and hard subsets. |
| $\mathcal{I}_{o_{l+1},o_l}^{(k)}, \mathcal{S}_{o_l}^{(k)}, \mathcal{S}_{o_l}^{(k),\mathrm{easy}}, \mathcal{S}_{o_l}^{(k),\mathrm{hard}}$ | Valid CoTs passing $(o_l, o_{l+1})$; subset of reachable $o_{l+1}$ from $o_l$ in valid CoTs; easy/hard-to-reason subsets. |
| $\boldsymbol{\theta}^\star, \boldsymbol{\theta}^k, \boldsymbol{\theta}^{k,(t)}$ | $\boldsymbol{\theta}^\star$: base model in Sec. 2.2; $\boldsymbol{\theta}^k$: task-$k$ model; superscript $(t)$: iteration. |
| $R_{\mathrm{out}}^k(\boldsymbol{o}), R_{\mathrm{out}}^{k,\hat{p}}(\boldsymbol{o})$ | $R_{\mathrm{out}}^k(\boldsymbol{o})$: Expected accuracy over $(\mathbf{Q}, \mathbf{A}) \sim \mathcal{D}_{a_q}^{q,k}$ for sampled CoT $\boldsymbol{o}$, by $\hat{p}_{\boldsymbol{\theta}^\star}$ or $\hat{p}$. |
| $R_{\mathrm{likelihood}}^k(\boldsymbol{o}_l), R_{\mathrm{DPRM}}^k(\boldsymbol{o}_l)$ | Expected accuracy of $\boldsymbol{o}_l$; DPRM reward in Eq.(14). |
| $A_{l+1}^{\hat{p}_{\boldsymbol{\theta}},k}(\boldsymbol{o}_l, \boldsymbol{o}_{l+1}), Q^{\hat{p}_{\boldsymbol{\theta}},k}(\boldsymbol{o}_l, \boldsymbol{o}_{l+1}), V^{\hat{p}_{\boldsymbol{\theta}},k}(\boldsymbol{o}_l)$ | RL's Advantage for task $k$; Expected accuracy of state $\boldsymbol{o}_{l+1}$ and $\boldsymbol{o}_l$. |
| $p_{\mathrm{acc}}^k(o)$ | Success probability of CoT $o$ for task $k$. |
| $\beta, \lambda$ | Temperature parameters of $\hat{p}_{\boldsymbol{\theta}^k}^{\mathrm{PO}}$ in Eq.(10) and $P_{\mathrm{Gibbs}}^k(\boldsymbol{o})$ in Eq.(12). |
| $\mathrm{Pass@K}_{q,k}^{\hat{p}}$ | Probability that $\hat{p}$ generates at least one correct CoT in $K$ samples for $(q, a_q, k)$. |
| $O(\cdot), \Omega(\cdot), \Theta(\cdot)$ | Standard asymptotic notation: upper, lower, and tight bounds, respectively. |

pressing hard-to-reason alternatives. This bias is particularly pronounced in TASK1, where RE-INFORCE, RAFT, and PPO generate 90-95% easy valid CoTs.

**Forgetting Phenomenon:** The dramatic performance degradation on TASK2 for standard RL methods (from 0.70-0.72 on TASK1 to 0.35-0.48 on TASK2) provides empirical evidence for the *forgetting* phenomenon predicted by our theoretical analysis. This confirms that overfitting to the primary task can severely compromise performance on secondary tasks.

**Diversity-Promoting Benefits:** Methods that promote diversity (Reinforce-rej (e.g. `RL-rej`), GRPO-KL, DPRM variants) demonstrate superior performance on TASK2 while maintaining reasonable performance on TASK1. This validates our theoretical prediction that diversity-promoting mechanisms are crucial for multi-task scenarios.

**Inference Scaling Effectiveness:** The PRM-based and DPRM-based inference methods show particularly interesting behavior, achieving high performance on TASK1 while maintaining reasonable coverage on TASK2. This suggests that process reward models can effectively guide reasoning without the computational overhead of fine-tuning.

These results collectively support our theoretical findings and provide practical guidance for designing effective multi-task reasoning systems in large language models.

# G  DETAILS OF REWARD MODELS AND METHODS

## G.1  SUMMARY OF NOTATIONS

We remark that in our setting, for all $l \in [L]$, $\boldsymbol{o}_l = e_{\boldsymbol{o}_l} \in \mathbb{R}^{|S|}$ denotes the one-hot encoding of token $o_l$ from the vocabulary. In practice, language models typically apply a softmax over the entire vocabulary to produce next-token probabilities. Hence, for simplicity, we do not distinguish between $\boldsymbol{o}_l$ and $o_l$ in notation, and treat them interchangeably throughout the paper. We summarize our notation in Table 4.

Let $\mathcal{Q}_k \subseteq S_1$ be the set of *question states* for task $k \in \mathcal{T}$. Suggest $P^k$ is a distribution over the *question states* $\mathcal{Q}_k$ associated with task $k$, denote $R_{\mathrm{out}}^k(\boldsymbol{o}) = \mathbb{E}_{(\mathbf{Q},\mathbf{A})\sim\mathcal{D}_{a_q}^{q,k}}[R_{(\mathbf{Q},\mathbf{A})}^k(\boldsymbol{o})]$ Setlur et al. (2025a; 2024) as the population reward over $\mathcal{D}_{a_q}^{q,k}$ of the task tuple $(q, a_q, k)$.

## G.2  RLVR FINETUNING

**REINFORCE.** The classical REINFORCE algorithm Williams (1992) maximizes the expected reward from sampled trajectories. For mathematical reasoning, a standard approach is using $0-1$ correctness of reasoning answer as the reward Xiong et al. (2025); Setlur et al. (2025a). In our TMC setting, for task $k$ and given prompt $q$, the REINFORCE objective is

$$\mathcal{J}_{\mathrm{REINFORCE}}(\boldsymbol{\theta}^k) = \mathbb{E}_{o_1=q\sim P^k(\mathcal{Q}_k),(\mathbf{Q},\mathbf{A})\sim\mathcal{D}_{a_q}^{q,k},\{\boldsymbol{o}^i\}_{i=2}^G\sim\hat{p}_{\boldsymbol{\theta}^k}(O|\boldsymbol{o}_1^i)} \left[\mathbb{1}\big(\boldsymbol{o} \in \mathcal{G}_{\mathbf{Q},\mathbf{A}}^{(k)}\big)\right], \quad (15)$$

where $\boldsymbol{o}_{1:L}$ denotes the trajectory sampled from the policy, and $\mathbb{1}(\boldsymbol{o} \in \mathcal{G}_{\mathbf{Q},\mathbf{A}}^{(k)}) \in \{0,1\}$ indicates whether the final output yields the correct answer. In our scenario, the objective would become

$$\mathcal{J}_{\text{REINFORCE}}(\boldsymbol{\theta}^k) = \mathbb{E}_{o_1=q\sim P^k(\mathcal{Q}_k), o_{2:L}\sim\hat{p}_{\boldsymbol{\theta}^k}}\left[R_{\text{out}}^k(\boldsymbol{o})\right].$$

**RAFT** (Rejection Sampling Fine-tuning) optimizes LLMs by sampling multiple responses from a policy, using a reward signal to select the best one, and then fine-tuning the policy using supervised learning on the selected best responses Xiong et al. (2025); Dong et al. (2023). The objective is to maximize the likelihood of these high-reward outputs:

$$\mathcal{J}_{\text{RAFT}}(\boldsymbol{\theta}) = \mathbb{E}_{[(q,o^*)\sim\mathcal{D}_{\text{RAFT}}]}[\log \pi_{\boldsymbol{\theta}}(o^*|q)], \tag{16}$$

where $\mathcal{D}_{\text{RAFT}}$ is a dataset constructed from queries $q$ and their corresponding best sampled responses $o^*$, as determined by a reward function. Xiong et al. (2025) found that a minimal RL approach to finetune the base model is to reject both the entirely correct and incorrect responses. In our TMC case, we have

$$\mathcal{J}_{\text{RAFT}}(\boldsymbol{\theta}^k) = \mathbb{E}_{\boldsymbol{o}_1\sim P^k(\mathcal{Q}^k), (\mathbf{Q},\mathbf{A})\sim\mathcal{D}_{a_{o_1}}^{o_1,k}, \boldsymbol{o}_{2:L}\sim\hat{p}_{\boldsymbol{\theta}^k}^k(O|\boldsymbol{o}_1)}\left[\sum_{l=1}^{L-1}\log \hat{p}_{\boldsymbol{\theta}^k}(\boldsymbol{o}_{l+1}|\boldsymbol{o}_l)R_{(\mathbf{Q},\mathbf{A})}^k(\boldsymbol{o})\right]$$

**Direct Preference Optimization (DPO)** optimizes the policy directly using a dataset of human preferences, provided as pairs of preferred ($o_w$) and dispreferred ($o_l$) responses for a given prompt $q$ Rafailov et al. (2024). It avoids explicit reward model training or reinforcement learning, instead optimizing a loss based on the policy's probability ratio relative to a reference policy $\pi_{ref}$:

$$\mathcal{J}_{\text{DPO}}(\boldsymbol{\theta}) = \mathbb{E}_{[(q,o_w,o_l)\sim\mathcal{D}_{\text{DPO}}]}[\log \sigma\left(\beta \log \frac{\pi_{\boldsymbol{\theta}}(o_w|q)}{\pi_{ref}(o_w|q)} - \beta \log \frac{\pi_{\boldsymbol{\theta}}(o_l|q)}{\pi_{ref}(o_l|q)}\right)], \tag{17}$$

where $\mathcal{D}_{\text{DPO}}$ is the preference dataset, $\sigma$ is the logistic sigmoid function, and $\beta$ is a temperature hyperparameter that scales the difference in log-probabilities.

In our TMC setting, for task $k$, suppose for each prompt $q$, the reference model (base model $\hat{p}_{\boldsymbol{\theta}^\star}$ or current model $\hat{p}_{\text{old}}^k$) produces two candidate trajectories: a preferred one $\boldsymbol{o}_{1:L}^+$, and a dispreferred one $\boldsymbol{o}_{1:L}^-$, where $R_{(\mathbf{Q},\mathbf{A})}^k(\boldsymbol{o}_L^+) = 1 > R_{(\mathbf{Q},\mathbf{A})}^k(\boldsymbol{o}_L^-) = 0$. The DPO objective for the current policy $\hat{p}_{\boldsymbol{\theta}^k}$ is:

$$\mathcal{J}_{\text{DPO}}(\boldsymbol{\theta}^k) := \sum_{(q,\boldsymbol{o}^+,\boldsymbol{o}^-)\in\mathcal{D}^k} \log \sigma\left(\beta \cdot \left[\log \frac{\hat{p}_{\boldsymbol{\theta}^k}(\boldsymbol{o}_{2:L}^+|\boldsymbol{o}_1^+)}{\hat{p}_{\text{old}}^k(\boldsymbol{o}_{2:L}^+|\boldsymbol{o}_1^+)} - \log \frac{\hat{p}_{\boldsymbol{\theta}^k}(\boldsymbol{o}_{2:L}^-|\boldsymbol{o}_1^-)}{\hat{p}_{\text{old}}^k(\boldsymbol{o}_{2:L}^+|\boldsymbol{o}_1^+)}\right]\right), \tag{18}$$

where $\sigma(\cdot)$ is the sigmoid function and $\beta > 0$ is a temperature hyperparameter controlling preference sharpness. This objective promotes the likelihood ratio of preferred over dispreferred CoTs as measured under $\hat{p}_{\boldsymbol{\theta}^k}$, relative to the fixed reference $\hat{p}_{\boldsymbol{\theta}^\star}$ used for sampling.

**Proximal Policy Optimization (PPO)** Schulman et al. (2017) optimizes LLMs by maximizing the following surrogate objective OpenAI (2018):

$$\mathcal{J}_{\text{PPO}}(\boldsymbol{\theta}) = \mathbb{E}_{[q\sim P(Q), o\sim\pi_{\boldsymbol{\theta}_{old}}(O|q)]}\frac{1}{|o|}$$

$$\sum_{t=1}^{|o|}\min[\frac{\pi_{\boldsymbol{\theta}}(o_t|q, o_{<t})}{\pi_{\boldsymbol{\theta}_{old}}(o_t|q, o_{<t})}A_t, \text{clip}(\frac{\pi_{\boldsymbol{\theta}}(o_t|q, o_{<t})}{\pi_{\boldsymbol{\theta}_{old}}(o_t|q, o_{<t})}, 1-\epsilon, 1+\epsilon)A_t], \tag{19}$$

where $A_t$ is the advantage computed via Generalized Advantage Estimation (GAE), requiring an *additional critic model*. $\epsilon$ is a clipping-related hyperparameter.

In our TMC setting, we have the advantage function as

$$A_{l+1}^{\hat{p}_{\boldsymbol{\theta}},k}(\boldsymbol{o}_l, \boldsymbol{o}_{l+1}) := Q^{\hat{p}_{\boldsymbol{\theta}},k}(\boldsymbol{o}_l, \boldsymbol{o}_{l+1}) - V^{\hat{p}_{\boldsymbol{\theta}},k}(\boldsymbol{o}_l). \tag{20}$$

Here the transition-value and state-value functions are

$$Q^{\hat{p}_{\boldsymbol{\theta}},k}(\boldsymbol{o}_l, \boldsymbol{o}_{l+1}) := \mathbb{E}_{o_1=q\sim P^k(\mathcal{Q}_k), \boldsymbol{o}_{l+2:L}\sim\hat{p}_{\boldsymbol{\theta}}}\left[R_{\text{out}}^k(\boldsymbol{o})\big|\boldsymbol{o}_l, \boldsymbol{o}_{l+1}\right], \tag{21}$$

$$V^{\hat{p}_{\boldsymbol{\theta}},k}(\boldsymbol{o}_l) := \mathbb{E}_{o_1=q\sim P^k(\mathcal{Q}_k), \boldsymbol{o}_{l+1}\sim\hat{p}_{\boldsymbol{\theta}}(\cdot|\boldsymbol{o}_l)}\left[Q^{\hat{p}_{\boldsymbol{\theta}},k}(\boldsymbol{o}_l, \boldsymbol{o}_{l+1})\right]. \tag{22}$$

The PPO objective OpenAI (2018) in our scenario is

$$
\begin{aligned}
\mathcal{J}_{\text{PPO}}(\boldsymbol{\theta}^k) =& \mathbb{E}_{q \sim P^k(\mathcal{Q}_k), (\mathbf{Q},\mathbf{A}) \sim \mathcal{D}_{a_q}^{q,k}, \{\boldsymbol{o}^i\}_{i=2}^G \sim \hat{p}_{\boldsymbol{\theta}^k}(O|\boldsymbol{o}_1^i)} \left[ \frac{1}{L} \sum_{l=1}^{L-1} \min\left[ \frac{\hat{p}_{\boldsymbol{\theta}^k}(\boldsymbol{o}_{l+1}^i|\boldsymbol{o}_l^i)}{\hat{p}_{\text{old}}^k(\boldsymbol{o}_{l+1}^i|\boldsymbol{o}_l^i)} A_{l+1}^{\hat{p}_{\boldsymbol{\theta}},k}, \right. \right. \\
& \left. \left. \text{clip}\left( \frac{\hat{p}_{\boldsymbol{\theta}^k}(\boldsymbol{o}_{l+1}^i|\boldsymbol{o}_l^i)}{\hat{p}_{\text{old}}^k(\boldsymbol{o}_{l+1}^i|\boldsymbol{o}_l^i)}, 1-\epsilon, 1+\epsilon \right) A_{l+1}^{\hat{p}_{\boldsymbol{\theta}},k} \right] \right].
\end{aligned}
\tag{23}
$$

In our modeling setup, the advantage estimate $A_{l+1}^{\hat{p}_{\boldsymbol{\theta}},k}$ aims to approximate Eq.(20), the gap between the value of making a particular transition at step $l$, versus the expected value of acting from state $\boldsymbol{o}_l$ without knowledge of $\boldsymbol{o}_{l+1}$.

**GRPO** Shao et al. (2024), in contrast, samples a group of output trajectories $\{o^i\}_{i=1}^G$ from $\pi_{\boldsymbol{\theta}_{old}}$ and optimizes:

$$
\begin{aligned}
\mathcal{J}_{\text{GRPO}}^k(\boldsymbol{\theta}) =& \mathbb{E}_{[q \sim P(Q), \{o^i\}_{i=1}^G \sim \pi_{\boldsymbol{\theta}_{old}}(O|q)]} \\
& \frac{1}{G} \sum_{i=1}^G \frac{1}{|o_{\cdot}^i|} \sum_{t=1}^{|o_{\cdot}^i|} \{ \min\left[ \frac{\pi_{\boldsymbol{\theta}}(o_t^i|q, o_{<t}^i)}{\pi_{\boldsymbol{\theta}_{old}}(o_t^i|q, o_{<t}^i)} \hat{A}_{i,t}, \text{clip}\left( \frac{\pi_{\boldsymbol{\theta}}(o_t^i|q, o_{<t}^i)}{\pi_{\boldsymbol{\theta}_{old}}(o_t^i|q, o_{<t}^i)}, 1-\epsilon, 1+\epsilon \right) \hat{A}_{i,t} \right] \\
& - \beta \mathbb{D}_{KL}[\pi_{\boldsymbol{\theta}}||\pi_{ref}] \},
\end{aligned}
\tag{24}
$$

where $\hat{A}_{i,t}$ is computed based on relative rewards within the sampled group, and $\beta$ controls KL regularization.

In our scenario, the formulation of GRPO Shao et al. (2024) equates

$$
\mathcal{J}_{\text{GRPO}}^k(\boldsymbol{\theta}^k) = \mathbb{E}_{q \sim P^k(\mathcal{Q}_k), (\mathbf{Q},\mathbf{A}) \sim \mathcal{D}_{a_q}^{q,k}, \{\boldsymbol{o}^i\}_{i=2}^G \sim \hat{p}_{\boldsymbol{\theta}^k}(O|\boldsymbol{o}_1^i)} \left[ \frac{1}{G} \sum_{i=1}^G \frac{1}{L} \sum_{l=1}^{L-1} \left\{ \min\left[ \frac{\hat{p}_{\boldsymbol{\theta}^k}(\boldsymbol{o}_{l+1}^i|\boldsymbol{o}_l^i)}{\hat{p}_{\text{old}}^k(\boldsymbol{o}_{l+1}^i|\boldsymbol{o}_l^i)} \hat{A}_{i,l+1}^k, \right. \right. \right.
$$

$$
\left. \left. \left. \text{clip}\left( \frac{\hat{p}_{\boldsymbol{\theta}^k}(\boldsymbol{o}_{l+1}^i|\boldsymbol{o}_l^i)}{\hat{p}_{\text{old}}^k(\boldsymbol{o}_{l+1}^i|\boldsymbol{o}_l^i)}, 1-\epsilon, 1+\epsilon \right) \hat{A}_{i,l+1}^k \right] \right\} - \beta D_{\text{KL}}[\hat{p}_{\boldsymbol{\theta}^k}||\hat{p}_{\boldsymbol{\theta}^\star}] \right],
\tag{25}
$$

**Outcome Supervision RL with GRPO**. A outcome reward model assigns scores $\mathbf{r} = \{r_1^k, ..., r_G^k\}$ to sampled outputs, which are then normalized: $\widetilde{r}_i = \frac{r_{i,index(l)}^k - \text{mean}(\mathbf{r}^k)}{\text{std}(\mathbf{r}^k)}$ within the group. The advantage is set as $\hat{A}_{i,l+1}^k = \widetilde{r}_i^k, \forall l \in [L-1]$, aiming to approximate Eq. (20). Here, for the task $k \in \mathcal{T}$, if we consider an *offline* scenario, our outcome reward model is $r_{i,index(l)}^k = R_{\text{out}}^k(\cdot)$ defined in Sec. 4.

**Process Supervision RL with GRPO**. Instead of a single reward per output, a process reward model assigns step-wise rewards $\mathbf{R} = \{\{r_{1,index(1)}^k, ..., r_{1,index(L)}^k\}, ...\}$, where $index(l)$ denotes the $l$-th step's end token index. Rewards are normalized: $\widetilde{r}_{i,index(l)}^k = \frac{r_{i,index(l)}^k - \text{mean}(\mathbf{R})}{\text{std}(\mathbf{R})}$. The advantage is computed as:

$$
\hat{A}_{i,l}^k = \sum_{index(j) \geq l} \widetilde{r}_{i,index(j)}^k,
\tag{26}
$$

and the policy is optimized via Eq. (6). Specifically, we could adopt $r_{i,index(l)}^k = R_{\text{pro}}^k(\boldsymbol{o}_l^i), \forall i \in [G], l \in \{1, \cdots, L\}$ in Sec. 4. However, this approach is unnatural - since $R_{\text{DPRM}}^k(\cdot)$ is designed for temperature-controlled adjusted sampling. Instead, a more common approach is to choose the $R_{\text{likelihood}}^k(\boldsymbol{o}_l)$ in Eq.(11) and $R_{\text{potential}}^k(\boldsymbol{o}_l)$.

In this work, following Xiong et al. (2025); Yu et al. (2025), we only studied the properties of GRPO with outcome reward. However, our theorem can include the GRPO with process reward by assuming that the advantage calculated in Eq.(26) is approximating Eq.(20) accurately.

### G.3 REWARD-BASED SAMPLING

**ORM Mode**. Given an input $x$, the model generates an CoT trajectory $o_1, \cdots, o_L$. Define $\boldsymbol{o}_l \in \mathbb{R}^{|S|}$ as the one-hot vector representing $o_1$, $\boldsymbol{o} = (\boldsymbol{o}_1, \cdots, \boldsymbol{o}_L)^\top \in \mathbb{R}^{L \times |S|}$ as the trajectory vector. An outcome reward model (ORM) $R_{\text{out}}^k(\cdot)$ assigns a scalar score based on the entire output:

$$R_{\text{out}}^k(\boldsymbol{o}) = f(\boldsymbol{o}), \tag{27}$$

where $f(\cdot)$ usually evaluates correctness, coherence, or other task-specific criteria Shao et al. (2024); Wang et al. (2024); Li et al. (2023); Snell et al. (2024).

**PRM Mode**. Instead of rewarding only the final output, a *process reward model* (PRM) assigns intermediate rewards along the reasoning trajectory:

$$R_{\text{pro}}^k(\boldsymbol{o}_l) = g(\boldsymbol{o}_1, ..., \boldsymbol{o}_l), \quad l \in \{1, ..., L\}, \tag{28}$$

where $g(\cdot)$ estimates step-wise utility using heuristics, verification signals, or learned evaluation metrics Shao et al. (2024); Snell et al. (2024); Wang et al. (2024); Li et al. (2023). Designing process rewards from outcome rewards is essential due to the high cost of human annotation. However, existing approaches are largely heuristic—either based on (i) the expected correctness of the final answer from the current state, typically via Monte Carlo rollouts Setlur et al. (2025a; 2024); Wang et al. (2024):

$$R_{\text{likelihood}}^k(\boldsymbol{o}_l) = \mathbb{E}_{\boldsymbol{o}_{l+1:L} \sim \hat{p}_{\boldsymbol{\theta}^\star}} \left[ R_{\text{out}}^k(\boldsymbol{o}) \mid \boldsymbol{o}_l \right], \tag{29}$$

and (ii) using binary signals to indicate whether the current state can still reach a correct solution Snell et al. (2024); Setlur et al. (2025b):

$$R_{\text{potential}}^k(\boldsymbol{o}_l) = \sup_{\boldsymbol{o}' : \boldsymbol{o}_l' = \boldsymbol{o}_l, \, \boldsymbol{o}' \in \mathcal{T}_{\text{all}}} R_{\text{out}}^k(\boldsymbol{o}') = \mathbf{1} \left\{ \exists \boldsymbol{o}' \in \mathcal{T}_{\text{all}} : R_{\text{out}}^k(\boldsymbol{o}', a) \right\}, \tag{30}$$

for all $\boldsymbol{o}_l \in S_l$, $l \in \{1, \ldots, L\}$. Here, $\mathcal{T}_{\text{all}}$ is typically approximated typically by Monte Carlo rollouts.

**Temperature-controlled Adjusted Sampling**. Here, we consider refinubf the sampling distribution using the reward model $R_{\text{out}}^k$. Define the original sampling probability of a trajectory $\boldsymbol{o}$ under $\boldsymbol{\theta}^\star$ as:

$$\hat{p}_{\boldsymbol{\theta}^\star}(\boldsymbol{o}) = \mathbb{P}_\rho^{\text{test}}(o_1) \prod_{l=1}^{L-1} \hat{p}_{\boldsymbol{\theta}^\star}(o_{l+1}|o_l),$$

where $\mathbb{P}_\rho^{\text{test}}(o_1) = \Theta(1/M_0)$ is the initial distribution over $S_1$. The adjusted sampling distribution, guided by $R_{\text{out}}^k$, is defined as:

$$P_{\text{Gibbs}}^k(\boldsymbol{o}) = \frac{\hat{p}_{\boldsymbol{\theta}^\star}(\boldsymbol{o}) \exp\left(\lambda R_{\text{out}}^k(\boldsymbol{o})\right)}{\sum_{\boldsymbol{o}' \in \mathcal{T}_{\text{all}}} \hat{p}_{\boldsymbol{\theta}^\star}(\boldsymbol{o}') \exp\left(\lambda R_{\text{out}}^k(\boldsymbol{o}')\right)} \propto \hat{p}_{\boldsymbol{\theta}^\star}(\boldsymbol{o}) \exp\left(\lambda R_{\text{out}}^k(\boldsymbol{o})\right). \tag{31}$$

for a temperature parameter $\lambda > 0$, with normalization over $\mathcal{T}_{\text{all}}$. The estimation of the $\mathcal{T}_{\text{all}}$ is typically through *Monte Carlo Rollout*. This discrete distribution reweights the pretrained model's probabilities to favor trajectories with higher estimated rewards, consistent with traditional sampling literature where the exponential form amplifies the influence of the reward signal. The form $P_{\text{Gibbs}}^k(\boldsymbol{o}) \propto \hat{p}_{\boldsymbol{\theta}^\star}(\boldsymbol{o}) \exp(\lambda R_{\text{out}}^k(\boldsymbol{o}))$ mirrors soft policy sampling in RL and NLP literature (e.g., REINFORCE or importance sampling). $\lambda$ controls the trade-off: large $\lambda$ heavily biases toward high-reward trajectories; small $\lambda$ preserves the original distribution.

### G.4 DISCUSSION ON BROADER FINETUNING SETTINGS

**Nonlinear Logits.** When the model's logits deviate from the linear form in Eq.(2 and instead follow the general parameterization of Eq.(90), i.e., $\hat{p}_{\boldsymbol{\theta}}(\cdot \mid \boldsymbol{x}) = \text{softmax}(h_{\boldsymbol{\theta}}(\cdot, \boldsymbol{x}))$ for $\boldsymbol{x} \in \{0, 1\}^{|S|}$, the fine-tuning dynamics become considerably more complex.

As noted in Remark 14, Lemma 7 depends on a set of extended conditions, notably the Parameter Isolation condition (Eq.(109), which typically fails to hold in practice. In large language models (LLMs), token representations are entangled via shared parameters across layers and positions, making it impossible to isolate updates per token. This design is aligned with in-context learning Nichani et al. (2024), where sequential dependencies are a fundamental modeling assumption.

**Why, then, is the linear model still meaningful?** Our formulation models reasoning as a discrete Markov chain—an abstraction used in several recent studies Xu et al. (2019); Sanford et al. (2024); Abbe et al. (2024); Besta et al. (2024); Kim et al. (2025)—where the current state encodes all information for current reasoning step. Thus, global token dependencies are captured in state transitions, eliminating the need for positional entanglement. Prior work Nichani et al. (2024); Edelman et al. (2024) has shown that transformers can successfully learn Markovian dynamics, and in our setting, the linear softmax model is already overparameterized enough to capture the TMC structure.

To understand the impact of nonlinearity more concretely, we adopt a first-order approximation of the logit update at transition $o_l \to o_{l+1}$ following Proposition 1 in Ren & Sutherland (2025):

$$\Delta \log \hat{p}_{\boldsymbol{\theta}}(\cdot | \boldsymbol{o}_l) = \eta [\nabla_{h(\cdot, \boldsymbol{o}_l)} \log \hat{p}_{\boldsymbol{\theta}}(\cdot | \boldsymbol{o}_l)] \mathbb{E}\{\mathcal{K}_{\boldsymbol{\theta}}(\boldsymbol{o}_l, \boldsymbol{o}_l^{\text{train}})[\nabla_{h(\cdot, \boldsymbol{o}_l)} \mathcal{J}^{\text{train}}(\boldsymbol{o}_{l+1}^{\text{train}}, \boldsymbol{o}_l^{\text{train}})]\} + O(\eta^2 \|\nabla_{\boldsymbol{\theta}} h_{\boldsymbol{\theta}}(\cdot, \boldsymbol{o}_l)\|_{\text{op}})$$

$$= \eta(\mathbb{I} - \mathbf{1}\hat{p}_{\boldsymbol{\theta}}(\cdot | \boldsymbol{o}_l)^\top) \mathbb{E}\{\mathcal{K}_{\boldsymbol{\theta}}(\boldsymbol{o}_l, \boldsymbol{o}_l^{\text{train}})[\nabla_{h(\cdot, \boldsymbol{o}_l)} \mathcal{J}^{\text{train}}(\boldsymbol{o}_{l+1}^{\text{train}}, \boldsymbol{o}_l^{\text{train}})]\} + O(\eta^2 \|\nabla_{\boldsymbol{\theta}} h_{\boldsymbol{\theta}}(\cdot, \boldsymbol{o}_l)\|_{\text{op}})$$

where $\mathcal{J}^{\text{train}}$ is the state-wise loss function (e.g. entropy loss or expected accuracy $Q(\boldsymbol{o}_{l+1}^{\text{train}}, \boldsymbol{o}_l^{\text{train}})$), $\mathcal{K}_{\boldsymbol{\theta}}$ is the empirical NTK (eNTK) defined as $\mathcal{K}_{\boldsymbol{\theta}} = (\nabla_{\boldsymbol{\theta}} h_{\boldsymbol{\theta}}(\cdot, \boldsymbol{o}_l) \nabla_{\boldsymbol{\theta}} h_{\boldsymbol{\theta}}(\cdot, \boldsymbol{o}_l)^\top)$, and the expectation is taken over question states $q = o_1 \sim P^k(\mathcal{Q}^k)$, training instances $(\mathbf{Q}, \mathbf{A}) \sim \mathcal{D}_{a_q}^{q,k}$, and sampled CoTs $\boldsymbol{o}^{\text{train}} \sim \hat{p}_{\boldsymbol{\theta}^k}(\cdot | q)$. In contrast to the linear case where $\mathcal{K}_{\boldsymbol{\theta}} = \boldsymbol{o}_l \boldsymbol{o}_l^\top$, the nonlinear update depends on the learned geometry of the representation space.

The squeezing effect occurs if

$$\frac{\hat{p}_{\boldsymbol{\theta}^k}(\boldsymbol{o}_{l+1}' | \boldsymbol{o}_l)}{\hat{p}_{\boldsymbol{\theta}^k}(\boldsymbol{o}_{l+1} | \boldsymbol{o}_l)} \leq 1 \quad \Longleftrightarrow \quad \Delta \log \hat{p}_{\boldsymbol{\theta}}(\boldsymbol{o}_{l+1}') \geq \Delta \log \hat{p}_{\boldsymbol{\theta}}(\boldsymbol{o}_{l+1}),$$

for $o_{l+1}' \in C_{o_l}$ and $o_{l+1} \in D_{o_l} \setminus C_{o_l}$. The update difference satisfies:

$$\Delta \log \hat{p}_{\boldsymbol{\theta}}(\boldsymbol{o}_{l+1}' | \boldsymbol{o}_l) - \Delta \log \hat{p}_{\boldsymbol{\theta}}(\boldsymbol{o}_{l+1} | \boldsymbol{o}_l) = \eta [\boldsymbol{o}_{l+1}' - \boldsymbol{o}_{l+1}]^\top \Theta((\mathbb{I} - \mathbf{1}\hat{p}_{\boldsymbol{\theta}}(\cdot | \boldsymbol{o}_l)^\top) \mathbb{E}\{\mathcal{K}_{\boldsymbol{\theta}}(\boldsymbol{o}_l, \boldsymbol{o}_l^{\text{train}})[\nabla_{h(\cdot, \boldsymbol{o}_l)} \mathcal{J}^{\text{train}}(\boldsymbol{o}_{l+1}^{\text{train}}, \boldsymbol{o}_l^{\text{train}})]\}.$$

This shows that the relative update magnitudes—and thus the squeezing effect—depend on the eNTK structure and how different CoT representations interact. If the non-linear representations of hard and easy CoTs are highly correlated, their learning dynamics may reinforce or suppress each other, analogous to the phenomenon in Ren & Sutherland (2025), where learning digit 4 accelerates digit 9 but impedes unrelated classes. In our setting, this implies that whether the squeezing effect persists under nonlinearity hinges on structural coupling between CoTs in the representation space. In real-world, different reasoning patterns do have co-relations, and we left the broader investigations with certain assumptions as an important future direction.

**DPO.** Recall from Eq.(18 that the DPO objective is defined as:

$$\mathcal{J}_{\text{DPO}}(\boldsymbol{\theta}^k) := \sum_{(q, \boldsymbol{o}^+, \boldsymbol{o}^-) \in \mathcal{D}^k} \log \sigma \left( \beta \cdot \left[ \log \frac{\hat{p}_{\boldsymbol{\theta}^k}(\boldsymbol{o}_{2:L}^+ | \boldsymbol{o}_1^+)}{\hat{p}_{\text{old}}^k(\boldsymbol{o}_{2:L}^+ | \boldsymbol{o}_1^+)} - \log \frac{\hat{p}_{\boldsymbol{\theta}^k}(\boldsymbol{o}_{2:L}^- | \boldsymbol{o}_1^-)}{\hat{p}_{\text{old}}^k(\boldsymbol{o}_{2:L}^- | \boldsymbol{o}_1^-)} \right] \right),$$

where $R_{(\mathbf{Q}, \mathbf{A})}^k(\boldsymbol{o}_L^+) = 1 > R_{(\mathbf{Q}, \mathbf{A})}^k(\boldsymbol{o}_L^-) = 0$. As discussed in Ren & Sutherland (2025), DPO can exhibit a squeezing effect, and such dynamics might also apply under our TMC reasoning framework. However, DPO is not a natural fit for our setting: we are concerned with correctness rather than relative preferences over reasoning paths. As such, the data required to support DPO—pairs $(q, \boldsymbol{o}^+, \boldsymbol{o}^-)$ indicating relative preference—is not directly meaningful in our binary (0–1) reward formulation. For this reason, while the objective form is stated for reference, we do not pursue further theoretical development of DPO in this work. Nonetheless, it may serve as a promising direction for future study of RLHF under the TMC framework with additional assumptions on preference structure.

## H  DETAILS AND PROOFS OF TMDP

**Remark 5.** *For the reader's high-level understanding, we here list some scenarios where the common valid reasoning patterns do not suffice for specific instance.*

- ***Problem type: Algebra (quadratic equations)***
  ***Common Valid CoT:*** *applying factorization method to solve quadratic equations.*
  ***Scenario it is not Correct:*** *when the quadratic polynomial is irreducible over integers (e.g., $x^2 + x + 1 = 0$), factorization fails.*

- *Problem type: Geometry (triangle side relations)*
  *Common Valid CoT: applying the Pythagorean theorem to relate side lengths of triangles.*
  *Scenario it is not Correct: when the triangle is not right-angled, Pythagoras' theorem does not hold.*

- *Problem type: Probability (complex event calculation)*
  *Common Valid CoT: applying the law of total probability to compute probabilities.*
  *Scenario it is not Correct: when the partition of events is not mutually exclusive, leading to double counting.*

- *Problem type: Number theory (modular arithmetic)*
  *Common Valid CoT: reasoning with modular addition to check congruences.*
  *Scenario it is not Correct: when an incorrect modulus is used (e.g., reducing modulo $6$ instead of $7$).*

- *Problem type: Combinatorics (counting problems)*
  *Common Valid CoT: applying permutation and combination formulas.*
  *Scenario it is not Correct: when order vs. unordered distinction is misapplied, such as using combinations when permutations are required.*

*This view is supported by recent large-scale error analyses on real math datasets. Sun et al. (2025) construct* MWPES-300K *(304,865 erroneous solutions across 15 LLMs and 4 datasets: SVAMP, GSM8K, AQuA, MATH) and discover that (i) error patterns diversify with dataset difficulty (e.g., MATH consistently elicits more diverse error types than GSM8K/SVAMP), indicating that simple "valid" patterns cease to be* correct *on harder instances; (ii) many failures arise from mis-applied* common patterns, *such as* Assumed independence of overlapping events *(AIO)*, Misapplication of probability formulas for independent events *(MPI)*, Incorrect combinatorial principles *(ICP)*, Unit/-Conversion errors *(UNE/FAC), or algebraic manipulation mistakes (MAM), showing that widely used CoT routes are not instance-wise reliable; and (iii)* Error-Aware Prompting *(EAP) selectively diverts models from their default CoT routes and yields sizable per-category gains on hard cases (e.g., AIO $+6.1pp$, MPI $+6.5pp$, UNE $+6.5pp$, FAC $+13.5pp$), evidencing the value of rarer, problem-specific reasoning paths over frequent but brittle patterns.*

*This aligns with recent findings Xiong et al. (2025); Li et al. (2025b); Ren & Sutherland (2025); Wang et al. (2025) highlighting the role of* reasoning diversity *and* entropy stability *in post-training, albeit evidence shows that post-training and inference-scaling do not explore beyond base model's tree-search knowledge Yue et al. (2025); AI et al. (2025); Gandhi et al. (2025).*

**Definition 4** (Formal Version of Def. 2)**.** *Let $X = (X_t)_{t \geq 0}$ be a Tree-structured Markov Chain (TMC) as defined in Def. 1, and let $\mathcal{T}$ be a collection of tasks. Each task $k \in \mathcal{T}$ specifies a set of different question state $\mathcal{Q}_k \subset S_1$, where each $q \in \mathcal{Q}_k$ has a corresponding unique correct answer $a_q^k$ under task $k$. For $(q, a_q^k) \neq (q', a_{q'}^k) \in \mathcal{Q}_k$ we have $q \neq q'$, $a_q^k \neq a_{q'}^k$.*

*A state tuple $(q, a_q = a_q^k, k)$ is called* common *if there exists at least one easy-to-reason chain of thought (CoT) $(o_1, \ldots, o_L)$ from $q$ to $a_q$, and* rare *otherwise. Each such state tuple is associated with a set $\mathcal{G}_{q,a_q}^{(k)} \subset S_1 \times \cdots \times S_L$ of valid CoTs.*

1. *All easy-to-reason CoTs from $q$ to $a_q$ belong to $\mathcal{G}_{q,a_q}^{(k)}$;*

2. *These CoTs are* not valid *for any task $k' \neq k$;*

3. *Hard-to-reason CoTs may or may not belong to $\mathcal{G}_{q,a_q}^{(k)}$;*

4. *Every edge $(o_l \rightarrow o_{l+1})$ with non-zero transition probability appears in some valid CoT for some task;*

5. *Each task state tuple $(q, a_q, k)$ induces a QA distribution $\mathcal{D}_a^{q,k}$, and the probability that a valid CoT $o_{1:L} \in \mathcal{G}_{q,a_q}^{(k)}$ is* correct *for a concrete instance $(\mathbf{Q}, \mathbf{A}) \sim \mathcal{D}_a^{q,k}$ is given by $p_{acc}^k(o)$:*

$$p_{acc}^k(o) = \frac{\prod_{l=1}^{L-1} \mathbb{P}_{\text{TMC}}(o_{l+1} \mid o_l)}{\sum_{o'_{1:L} \in \mathcal{G}_{q,a_q}^{(k)}} \prod_{l=1}^{L-1} \mathbb{P}_{\text{TMC}}(o'_{l+1} \mid o'_l)}. \tag{32}$$

*Further, we assume that the correctness of any different $o \neq o' \in \mathcal{G}_{q,a_q}^{(k)}$ for any instance $(\mathbf{Q}, \mathbf{A}) \sim \mathcal{D}_a^{q,k}$ is independent.*

*A task $k$ is denoted rare if there is no valid easy-to-reason CoT in its $\mathcal{G}_{q,a_q}^{(k)}$ for any question-answer state pair $(a, a_q) \in \mathcal{Q}_k$, and common other wise.*

**Remark 6.** *Here, the fifth condition is to provide merits for the probability distribution of the original TMC $X$ ($\mathbb{P}_{\text{TMC}}$) that models after real-world LLM. Typically, the predictive distribution obtained from pretraining would match the "frequency" of whether a CoT be valid for certain task. That is, through the 5-th condition, we justify why the original TMC $X$ ($\mathbb{P}_{\text{TMC}}$) would be equipped its distribution–driven by inherent chance to become a valid CoT for some reasoning task.*

*The independence assumption in the 5-th condition is for technical convenience. This definition would also induces instance $(\mathbf{Q}, \mathbf{A})$ under task $k$ that has no correct CoT, with probability $\prod_{o \in \mathcal{G}_{o_1,a_{o_1}^k}^{(k)}} (1 - p_{acc}^k(o))$. In real-world, the situation is far more complex, and we left the consideration of theory that assumes the interaction and co-relationship of the correctness of CoTs with different difficulty level for future work.*

*Some examples of Valid-not-Correct:*

*For some task tuple $(o_1, a_{o_1}, k)$, denote $\mathcal{G}_{o_1,a_{o_1}^k}^{(k),easy} \subseteq \mathcal{G}_{o_1,a_{o_1}^k}^{(k)}$ as the subset of valid easy-to-reason CoTs inside the valid CoTs for task $k$, and $\mathcal{G}_{o_1,a_{o_1}^k}^{(k),hard} := \mathcal{G}_{o_1,a_{o_1}^k}^{(k)} \setminus \mathcal{G}_{o_1,a_{o_1}^k}^{(k),easy}$ the subset of valid hard-to-reason CoTs. For any sampled instance $(\mathbf{Q}, \mathbf{A}) \sim \mathcal{D}_a^{q,k}$, it has the following two scenarios:*

- *With probability $\prod_{o \in \mathcal{G}_{o_1,a_{o_1}^k}^{(k),easy}} (1 - p_{acc}^k(o))$, the $(\mathbf{Q}, \mathbf{A})$ can only be correctly solved by some valid hard-to-reason CoTs in $\mathcal{G}_{o_1,a_{o_1}^k}^{(k),hard}$.*

- *With probability $1 - \prod_{o \in \mathcal{G}_{o_1,a_{o_1}^k}^{(k),easy}} (1 - p_{acc}^k(o))$, the $(\mathbf{Q}, \mathbf{A})$ can be correctly solved by some easy-to-reason CoTs in $\mathcal{G}_{o_1,a_{o_1}^k}^{(k),easy}$.*

*This division of probability space would equip bounding the pass@K performance when the model is only capable of all valid easy-to-reason CoTs. After the finetuned model is also capable of the hard-to-reason CoTs in $\widetilde{\mathcal{G}_{o_1,a_{o_1}^k}^{(k),hard}}$, we turn to be interested in the following division of probability space to discuss the pass@K performance:*

- *With probability $\prod_{o \in \mathcal{G}_{o_1,a_{o_1}^k}^{(k),easy} \cup \widetilde{\mathcal{G}_{o_1,a_{o_1}^k}^{(k),hard}}} (1 - p_{acc}^k(o))$, the $(\mathbf{Q}, \mathbf{A})$ can only be correctly solved by some **unlearned** valid hard-to-reason CoTs in $\mathcal{G}_{o_1,a_{o_1}^k}^{(k),hard} \setminus \widetilde{\mathcal{G}_{o_1,a_{o_1}^k}^{(k),hard}}$.*

- *With probability $1 - \prod_{o \in \mathcal{G}_{o_1,a_{o_1}^k}^{(k),easy} \cup \widetilde{\mathcal{G}_{o_1,a_{o_1}^k}^{(k),hard}}} (1 - p_{acc}^k(o))$, the $(\mathbf{Q}, \mathbf{A})$ can be correctly solved by some easy-to-reason CoTs or learned valid hard-to-reason CoTs in $\mathcal{G}_{o_1,a_{o_1}^k}^{(k),easy} \cup \widetilde{\mathcal{G}_{o_1,a_{o_1}^k}^{(k),hard}}$.*

We can characterize the breadth of tasks encoded in the topology of TMC as follows.

**Corollary 6** (Cardinality of Multi-task TMC). *Let $M_0 = |S_1|$, $M_L = |S_L|$, and for each $q \in S_1$ define*

$$A(q) = \big\{ a \in S_L : \exists \text{ easy-to-reason CoT } q \rightsquigarrow a \big\}, \qquad R(q) = \big\{ a \in S_L : \exists \text{ CoT } q \rightsquigarrow a \big\}.$$

*By Def. 1, we have $|A(q)| = n_q = O(1)$, and $n_q \geq 2$ for all $q$. Define the set of all tasks as*

$$\mathcal{T} = \{ k : S_1 \to S_L \}, \qquad \mathcal{T}_{\text{common}} = \{ k : k(q) \in A(q) \text{ for all } q \}, \qquad \mathcal{T}_{\text{rare}} = \mathcal{T} \setminus \mathcal{T}_{\text{common}}.$$

*Then*

$$|\mathcal{T}_{\text{common}}| = \prod_{q \in S_1} |A(q)| = \Theta(c^{M_0}), \qquad 2 \leq c \leq \max_q n_q = O(1),$$

*and*

$$|\mathcal{T}| \leq \prod_{q \in S_1} |R(q)| \leq M_L^{M_0}, \qquad |\mathcal{T}_{\text{rare}}| = |\mathcal{T}| - \Theta(c^{M_0}).$$

*In particular, although the total number of tasks grows exponentially in $M_0$, the number of* common *tasks is exponentially smaller whenever $M_L \gg c$.*

**Proof.***Proof of Lemma 6. Each task $k \in \mathcal{T}$ is a function $k : S_1 \to S_L$, so $|\mathcal{T}| \leq \prod_{q \in S_1} |R(q)|$, where $R(q)$ contains all reachable answers (via any CoT) from $q$. The set $\mathcal{T}_{\text{common}}$ consists of tasks for which $k(q) \in A(q)$ for all $q$, so*

$$|\mathcal{T}_{\text{common}}| = \prod_{q \in S_1} |A(q)| = \prod_{q \in S_1} n_q = \Theta(c^{M_0}),$$

*with $c \in [2, \max_q n_q] = O(1)$. The rest follows directly by subtraction.* $\square$

**Lemma 2.** *Consider a TMC $X = (X_t)_{t \geq 0}$ defined in Def. 1, and a specific task $k \in \mathcal{T}$ defined in Def. 2. Then, for any fixed $q = o_1 \in S_1$ and corresponding correct answer $a = o_L \in S_L$, with non-trivial probability, there exists at least one hard-to-reason CoT trajectory (i.e., a path containing at least one sparse edge) from $q$ to $a$. Specifically, the probability of having at least one such hard-to-reason trajectory, denoted $\mathbb{P}_{deg}(o_L = a|o_1 = q)$, is lower bounded as:*

$$\mathbb{P}_{deg}(o_L = a|o_1 = q) \geq \Theta(\epsilon \cdot M^{L-3}) \geq c \geq \Theta(M^{-2}) > 0,$$

*for some constant $c$, where $\epsilon = O(1/M^{L-2})$ is the transition probability of a sparse edge.*

**Proof of Lemma 2.** *Fix $q = o_1$ and $a = o_L$. Let $\Pi$ be the set of all length-$L$ trajectories $\tau = (o_1, \ldots, o_L)$ with $o_L = a$. We split $\Pi = \Pi_{\text{nd}} \cup \Pi_{\text{deg}}$ according to whether $\tau$ has zero or at least one sparse edge.*

*By Def. 1, there exist $O(1)$ "easy-to-reason" trajectories from $q \in S_1$ to $a \in S_L$, each consisting entirely of high-probability transitions $C_{o_l}$. Each transition $o_l \to o_{l+1}$ along these paths has probability $\Theta(1/M)$. Therefore, for a trajectory of $L - 1$ steps, the total probability of such a path is:*

$$\mathbb{P}_{high} = O(1) \cdot \left( \Theta\left(\frac{1}{M}\right) \right)^{L-1} = O\left(M^{-(L-1)}\right).$$

*Similarly, as the number of hard-to-reason CoT is below $\Theta(M)$, given that $\mathbb{P}_{sparse} \leq \Theta(1/M^{L-2})$, we conclude the total probability by union bound $\mathbb{P}_{\text{TMC}}(o_L = a \mid o_1 = q) = \Theta(M^{-(L-1)})$.*

$\square$

**Theorem 7** (Intrinsic Properties of Multi-task TMC). *Let $X = (X_t)_{t \geq 0}$ be a Tree-structured Markov Chain (TMC) and $\mathcal{T}$ a set of tasks, per defined in Def. 1 and 2.*

1. *(**Task Interference**) Let tasks $k, k' \in \mathcal{T}$ share at least one question state $q \in S_1$ or answer state $a \in S_L$, with distinct valid QA pairs $(q, a_q) \in \mathcal{Q}_k$ and $(q', a_{q'}) \in \mathcal{Q}_{k'}$. Suppose the transition probabilities along edges in $\mathcal{G}_{q,a_q}^{(k)}$ are amplified such that the TMC reaches $a_q$ with probability $1 - \delta$ (where $\delta = o(M^{-L}) << 1$) via valid CoTs in $\mathcal{G}_{q,a_q}^{(k)}$. Then for all shared $q$ or $a$, every originally easy-to-reason CoT in $\mathcal{G}_{q',a_{q'}}^{(k')}$ must satisfy:*

$$\mathbb{P}_{\text{TMC}}(o_{l+1}|o_l) = o(1/M^2) \quad \exists (o_l \to o_{l+1}) \in \tau \text{ and } \tau \in \mathcal{G}_{q',a_{q'}}^{(k')},$$

*i.e., all such CoTs degenerate into hard-to-reason paths. Similarly, for any task $\hat{k} \neq k \in \mathcal{T}$ whose valid CoT set $\mathcal{G}_{\hat{q},a_{\hat{q}}}^{(\hat{k})}$ has at least one easy-to-reason CoT $\hat{o}$ sharing some transitions $\hat{o}_l \to \hat{o}_{l+1}$ with the CoTs in $\mathcal{G}_{q,a_q}^{(k)}$. Then $\hat{o}$ becomes hard-to-reason.*

2. *(**Correctness Bottleneck**) Suppose the probability mass of valid hard-to-reason CoTs traveling from $q$ to $a_q$ for task $k$ in the original TMC $X$ ($\mathbb{P}_{\text{TMC}}$) is $\Delta$. Then suppose a model $\hat{p}$ satisfies:*
   - *The total probability mass from $q$ to $a$ is $1 - C$.*
   - *The fraction of easy-to-reason CoTs among CoTs traveling from $q$ to $a_q$ is $1 - \epsilon$.*

*Then the expected correctness over the QA distribution $\mathcal{D}_{a_q}^{q,k}$ is upper bounded by:*

$$R_{\text{out}}^{k}{}^{\hat{p}}(\boldsymbol{o}) \leq \Theta((1-C)[(1-\epsilon)\frac{1}{1+\Delta M^{L-1}} + \epsilon\frac{\Delta M^{L-1}}{1+\Delta M^{L-1}}])$$

*Besides, we denote the pass@K performance of model $\hat{p}$ for task tuple $(q, a_q, k)$ (the probability that at least succeed once among $K$ trials) as $\text{Pass@K}_{q,k}^{\hat{p}}$:*

$$\text{Pass@K}_{q,k}^{\hat{p}} := \Pr_{\substack{\{\boldsymbol{o}^i\}_{i\in[K]}\sim\hat{p}(O|q) \\ (\mathbf{Q},\mathbf{A})\sim\mathcal{D}_{a_q}^{q,k}}}[\bigcup_{i=1}^{K}\mathbb{1}(\boldsymbol{o}^i \in \mathcal{G}_{\mathbf{Q},\mathbf{A}}^{(k)})]. \tag{33}$$

*When $C = 0$, $\text{Pass@K}_{q,k}^{\hat{p}}$ is upper bounded by*

$$\text{Pass@K}_{q,k}^{\hat{p}} \leq \underbrace{\Theta([(\frac{\Delta M^{L-1}}{1+\Delta M^{L-1}})^{\text{n}_q}(1-(1-\varepsilon)^K)])}_{\text{upper bound of pass@K of instance that cannot be solved by easy CoTs}} + \underbrace{\Theta([[(1-(\frac{\Delta M^{L-1}}{1+\Delta M^{L-1}})^{\text{n}_q}(1-\varepsilon^K)])}_{\text{upper bound of pass@K of instance that can be solved by some easy CoT}}).$$

*If*

$$\varepsilon = o(\sqrt[K]{1 - C_{\text{Err}}/(\frac{\Delta M^{L-1}}{1+\Delta M^{L-1}})^{\text{n}_q}})$$

*for some $C_{\text{Err}} \in (0, (\frac{\Delta M^{L-1}}{1+\Delta M^{L-1}})^{\text{n}_q})$, then we have the pass@K performance upper bounded by*

$$1 - \Omega(C_{\text{Err}}) = o(1),$$

*with **constant error** $\Omega(C_{\text{Err}}) = \Theta(1)$.*
*When $C = \epsilon = 0$, we have*

$$R_{\text{out}}^{k}{}^{\hat{p}}(\boldsymbol{o}) \leq \Theta(\frac{1}{1+\Delta M^{L-1}})$$

*And the pass@K performance is upper bounded by*

$$\text{Pass@K}_{q,k}^{\hat{p}} \leq \Theta(1 - (\frac{\Delta M^{L-1}}{1+\Delta M^{L-1}})^{\text{n}_q}).$$

**Proof.** *Proof of Thm. 7.*

***1. non-negligible decay of transitions for other tasks when overfit a target task.***

*Fix a shared question state $q \in S_1$. By Def. 2(ii), the easy-to-reason CoTs in $\mathcal{G}_{q,a_q}^{(k)}$ and $\mathcal{G}_{q,a_{q'}}^{(k')}$ are disjoint. The amplification condition implies:*

$$\sum_{\tau\in\mathcal{G}_{q,a_q}^{(k)}} \mathbb{P}(\tau|q) \geq 1 - \delta = 1 - o(M^{-L}).$$

*Since $\sum_{\tau\in \text{all CoTs from } q} \mathbb{P}(\tau|q) = 1$, the remaining CoTs (including those in $\mathcal{G}_{q,a_{q'}}^{(k')}$) must satisfy:*

$$\sum_{\tau\in\mathcal{G}_{q,a_{q'}}^{(k')}} \mathbb{P}(\tau|q) \leq \delta = o(M^{-L}) << 1.$$

*For any easy-to-reason CoT $\tau = (q, o_2, \ldots, o_L = a_{q'}) \in \mathcal{G}_{q,a_{q'}}^{(k')}$, the original transition probabilities satisfy $\mathbb{P}_{\text{TMC}}(o_{l+1}|o_l) = \Theta(1/M)$ for all edges. However, since the total probability mass for $\tau$ is now $o(M^{-L})$, we have:*

$$\prod_{l=1}^{L-1} \mathbb{P}_{\text{TMC}}(o_{l+1}|o_l) = o(M^{-L}).$$

*Give $\mathbb{P}_{\text{TMC}}(o_{l+1}|o_l) \leq \Theta(1/M)$, this forces at least one transition term to decay to $o(1/M^2)$. Otherwise, if any edge retained $\mathbb{P}_{\text{TMC}}(o_{l+1}|o_l) = \Theta(1/M)$, the product would be $\Theta(M^{-(L-1)})$, contradicting $\mathbb{P}(\tau|q) = o(M^{-L})$.*

*For a shared answer state $a \in S_L$, as well as $\hat{o}$ in othe task sharing some transition with CoTs in $\mathcal{G}_{q,a_q}^{(k)}$, the same logic applies.*

### 2. non-negligible error when only favoring easy-to-learn CoTs.

*We have the total mass of the valid CoTs for task $k$ in the original $X$ as*

$$Z \geq \underbrace{\Theta(\frac{1}{M^{L-1}})}_{Z_{easy}} + \underbrace{\Delta}_{Z_{hard}}$$

*By Def. 2, where the expected correctness over a QA sample is proportion to the CoT's likelihood in the original $X$, we can combine the components to upper bound the expected correctness:*

$$R_{\text{out}}^{k}{}^{\hat{p}}(\boldsymbol{o}) = \mathbb{E}_{\substack{(\mathbf{Q},\mathbf{A})\sim\mathcal{D}_{a_q}^{q,k} \\ \boldsymbol{o}\sim\hat{p}(\cdot|q)}}[\mathbb{1}(\boldsymbol{o}\in\mathcal{G}_{\mathbf{Q},\mathbf{A}}^{(k)})] \leq \Theta((1-C)[(1-\epsilon)\frac{1}{1+\Delta M^{L-1}}+\epsilon\frac{\Delta M^{L-1}}{1+\Delta M^{L-1}}])$$

*Especially, by the first discussion of division of probability space in Remark 6, it is direct to deduce that the probability that one specific $(\mathbf{Q},\mathbf{A}) \sim \mathcal{D}_{a_q}^{q,k}$ cannot be solved by every easy-to-reason CoT is*

$$\prod_{o\in\mathcal{G}_{o_1,a_{o_1}^k}^{(k),easy}} (1-p_{acc}^k(o)) = \Theta((1-\frac{1}{1+\Delta M^{L-1}})^{\mathbf{n}_q}) = \Theta((\frac{\Delta M^{L-1}}{1+\Delta M^{L-1}})^{\mathbf{n}_q})$$

*When facing these instances, we have the probability of success to be **at most** $\varepsilon$ when $C = 0$.*

*Besides, the probability that $(\mathbf{Q},\mathbf{A}) \sim \mathcal{D}_{a_q}^{q,k}$ can be solved by some easy-to-reason CoT is*

$$1 - \Theta((\frac{\Delta M^{L-1}}{1+\Delta M^{L-1}})^{\mathbf{n}_q})$$

*When facing these instances, we have the probability of success to be **at most** $1 - \varepsilon$ when $C = 0$.*

*Therefore, collaborating with $\mathbf{n}_q = O(1), \forall q \in S_1$, the pass@K performance (the probability that at least succeed once among $K$ trials) is upper bounded by*

$$\underbrace{\Theta([(\frac{\Delta M^{L-1}}{1+\Delta M^{L-1}})^{\mathbf{n}_q}(1-(1-\varepsilon)^K)])}_{\text{upper bound of pass@K of instance that cannot be solved by easy CoTs}} + \underbrace{\Theta([(1-(\frac{\Delta M^{L-1}}{1+\Delta M^{L-1}})^{\mathbf{n}_q})(1-\varepsilon^K)])}_{\text{upper bound of pass@K of instance that can be solved by some easy CoT}} ).$$

*If*

$$\varepsilon = o(\sqrt[K]{1 - C_{\text{Err}}/(\frac{\Delta M^{L-1}}{1+\Delta M^{L-1}})^{\mathbf{n}_q}}),$$

*for some $C_{\text{Err}} \in (0, (\frac{\Delta M^{L-1}}{1+\Delta M^{L-1}})^{\mathbf{n}_q})$, then we have the pass@K performance upper bounded by $1 - \Omega(C_{\text{Err}})$ with constant error $\Theta(C_{\text{Err}}) = \Theta(1)$.*

*When $C = \epsilon = 0$, we have $R_{\text{out}}^k(\boldsymbol{o}) \leq \Theta(\frac{1}{1+\Delta M^{L-1}})$. The pass@K performance is upper bounded by*

$$\Theta(1 - (\frac{\Delta M^{L-1}}{1+\Delta M^{L-1}})^{\mathbf{n}_q}).$$

$\square$

**Lemma 3** (Formal Version of Prop. 1). *Let $\boldsymbol{\theta}^\star$ be the base model in Eq.(2 that exact predicts the distribution of a Multi-task TMC as in Def. 1 and 2, fix a common task state tuple $(q, a, k)$. For any valid easy-to-reason CoT $o^{easy}$ and hard-to-learn CoT $o^{hard}$ that share the states $o_l$, and deviate at the $l+1$ layer (i.e., $o_l^{easy} = o_l^{hard} = o_l$, $o_{l+1}^{easy} \neq o_{l+1}^{hard}$), if the total number of valid hard-to-reason CoTs is bounded by $\Theta(M)$, we have $A_{l+1}^{\hat{p}_{\boldsymbol{\theta}^\star},k}(\boldsymbol{o}_l,\boldsymbol{o}_{l+1}^{easy}) \geq c_1 > 0$, $A_{l+1}^{\hat{p}_{\boldsymbol{\theta}^\star},k}(\boldsymbol{o}_l,\boldsymbol{o}_{l+1}^{hard}) \leq -c_2 < 0, \forall l \in [L-1]$ for some constants $c_1, c_2 > 0$.*

**Proof of Lemma 3.**

*First, it is direct to see that given any valid easy-to-learn CoT $o^{easy} \in \mathcal{G}_{q,a_q}^{(k)}$ and hard-to-learn CoT $o^{hard} \in \mathcal{G}_{q,a_q}^{(k)}$ for task tuple $(q, a, k)$, we have*

$$\frac{\prod_{l=1}^{L-1} \mathbb{P}_{\text{TMC}}(o_{l+1}^{easy} \mid o_l)}{\sum_{o_{1:L}' \in \mathcal{G}_{q,a_q}^{(k)}} \prod_{l=1}^{L-1} \mathbb{P}_{\text{TMC}}(o_{l+1}' \mid o_l')} \Big/ \frac{\prod_{l=1}^{L-1} \mathbb{P}_{\text{TMC}}(o_{l+1}^{hard} \mid o_l)}{\sum_{o_{1:L}' \in \mathcal{G}_{q,a_q}^{(k)}} \prod_{l=1}^{L-1} \mathbb{P}_{\text{TMC}}(o_{l+1}' \mid o_l')} \geq \frac{\Theta(M^{-1})}{o(M^{-1})} > \Omega(M), \tag{34}$$

*by Eq.(32).*

*That is, the expected accuracy of $o^{easy}$ on any instance from $\mathcal{D}_{a_q}^{q,k}$, denoted as $p_{acc}^k(o^{easy})$ is larger than the $o^{hard}$, denoted as $p_{acc}^k(o^{hard})$, with a ratio no less than $\Theta(M)$.*

*Fix $l$ and $o_l$. Write*

$$Q^{\hat{p}_{\boldsymbol{\theta}^\star},k}(\boldsymbol{o}_l, \boldsymbol{o}_{l+1}^{easy}) = \mathbb{E}_{\substack{q=o_1 \sim P^k(\mathcal{Q}_k) \\ \boldsymbol{o} \sim \hat{p}_{\boldsymbol{\theta}^\star}(\cdot | \boldsymbol{o}_1)}}[\mathbb{1}(o_L = a_q)p_{acc}^k(o) \mid {}_{o_{l+1}=o_{l+1}^{easy}}^{o_l=o_l}],$$

*and*

$$V^{\hat{p}_{\boldsymbol{\theta}^\star},k}(\boldsymbol{o}_l) = \sum_{\boldsymbol{o}_{l+1} \in S_{l+1}} \hat{p}_{\boldsymbol{\theta}^\star}(\boldsymbol{o}_{l+1} \mid \boldsymbol{o}_l) Q^{\hat{p}_{\boldsymbol{\theta}^\star},k}(\boldsymbol{o}_l, \boldsymbol{o}_{l+1}).$$

*By definition and Lemma 2 there are $\Theta(1)$ length-$(L-l)$ continuations each with probability $\Theta(M^{-(L-l)})$. Hence*

$$Q^{\hat{p}_{\boldsymbol{\theta}^\star},k}(\boldsymbol{o}_l, \boldsymbol{o}_{l+1}^{easy}) = \Theta\big(M^{-(L-l)}\mathbb{E}[p_{acc}^k(o) \mid {}_{o_{l+1}=o_{l+1}^{easy}}^{o_l=o_l}]\big),$$

*Also we see $o_{l+1}^{easy} \in C_{o_l}$. Then $\Pr[o_{l+1}^{easy} \mid o_l] = \Theta(1/M)$, thus*

$$V^{\hat{p}_{\boldsymbol{\theta}^\star},k}(\boldsymbol{o}_l) \geq \Theta\big(M^{-(L-l+1)}\mathbb{E}[p_{acc}^k(o) \mid {}_{o_{l+1}=o_{l+1}^{easy}}^{o_l=o_l}]\big).$$

*Therefore, we have*

$$A_{l+1}^{\hat{p}_{\boldsymbol{\theta}},k}(\boldsymbol{o}_l, \boldsymbol{o}_{l+1}^{easy}) = Q^{\hat{p}_{\boldsymbol{\theta}^\star},k}(\boldsymbol{o}_l, \boldsymbol{o}_{l+1}^{easy}) - V^{\hat{p}_{\boldsymbol{\theta}^\star},k}(\boldsymbol{o}_l) \geq \Theta\big(M^{-(L-l+1)}\mathbb{E}[p_{acc}^k(o) \mid {}_{o_{l+1}=o_{l+1}^{easy}}^{o_l=o_l}]\big) > 0$$

*$o_{l+1}^{hard} \notin C_{o_l}$. Then $\Pr[o_{l+1}^{hard} \mid o_l] = o(M^{-2})$, and the best possible continuations contribute at most $\Theta(M^{-(L-l-1)})$ each. Thus*

$$Q^{\hat{p}_{\boldsymbol{\theta}^\star},k}(\boldsymbol{o}_l, \boldsymbol{o}_{l+1}^{hard}) \leq o(M^{-2}) \cdot O\big(M^{-(L-l-1)}\big)\mathbb{E}[p_{acc}^k(o) \mid {}_{o_{l+1}=o_{l+1}^{hard}}^{o_l=o_l}]$$
$$= o\big(M^{-(L-l+1)}\mathbb{E}[p_{acc}^k(o) \mid {}_{o_{l+1}=o_{l+1}^{hard}}^{o_l=o_l}]\big).$$

*Therefore*

$$A_{l+1}^{\hat{p}_{\boldsymbol{\theta}},k}(\boldsymbol{o}_l, \boldsymbol{o}_{l+1}^{hard}) = Q_{l+1}^{\hat{p}_{\boldsymbol{\theta}},k}(\boldsymbol{o}_l, \boldsymbol{o}_{l+1}^{hard}) - V^{\hat{p}_{\boldsymbol{\theta}^\star},k}(\boldsymbol{o}_l)$$
$$= o\big(M^{-(L-l+1)}\mathbb{E}[p_{acc}^k(o) \mid {}_{o_{l+1}=o_{l+1}^{hard}}^{o_l=o_l}]\big) - \Theta\big(M^{-(L-l+1)}\mathbb{E}[p_{acc}^k(o) \mid {}_{o_{l+1}=o_{l+1}^{easy}}^{o_l=o_l}]\big)$$
$$< 0.$$

*Given that for a chosen TMC $\mathbb{P}_{\text{TMC}}(\cdot \mid \cdot) = \hat{p}_{\boldsymbol{\theta}^\star}(\cdot | \cdot)$, the $\mathbb{E}[p_{acc}^k(o) \mid {}_{o_{l+1}=o_{l+1}^{easy}}^{o_l=o_l}], \mathbb{E}[p_{acc}^k(o) \mid {}_{o_{l+1}=o_{l+1}^{hard}}^{o_l=o_l}]$ are constants. Therefore, by choosing some positive constants $c_1 = \Theta(M^{-(L+1-l)}), c_2 = \Theta(M^{-(L+1-l)})$ to bound the advantages, we complete the proof.* $\qquad \square$

## I  DETAILS AND PROOFS OF PRETRAINING

Following Kim et al. (2025), we could have the following theorem.

**Theorem 8.** *Let $X_0 \sim \text{Unif}(S \setminus S_L)$ and $X_1 \sim \mathbb{P}(\cdot | X_0)$ be random samples from the TMC $X$ in Def. 1. Let $X_0 \sim \text{Unif}(S \setminus S_L)$ and $X_1 \sim \mathbb{P}(\cdot | X_0)$ be random samples from the TMC $X$ defined in Def. 1. For $i \in S_l$, define:*

$$D_{o_l} = \{o_{l+1} : \mathbb{P}_{\text{TMC}}(o_{l+1}|o_l) > 0\}, \quad c = \min_{o_{l+1} \in D_{o_l}} \mathbb{P}_{\text{TMC}}(o_{l+1}|o_l) > 0,$$

*then the softmax predictor trained via Algorithm 1 satisfies:*

---

**Algorithm 1** Pretraining of Foundation Model Kim et al. (2025)

---

1: set $\boldsymbol{\theta}^{(0)} = \boldsymbol{0}$, $\eta = O(M)$,
2: $T_1 = \widetilde{O}(M^2 c^{-2})$, $T_2 = \widetilde{O}(Mc^{-2})$
3: **for** $t = 1, \cdots, T_1$ **do**
4: $\quad \boldsymbol{\theta}^{(t)} = \boldsymbol{\theta}^{(t-1)} + \eta \nabla \mathbb{E}_{X_0, X_1}[\log \hat{p}_{\boldsymbol{\theta}^{(t-1)}}(X_1 | X_0)]$
5: **end for**
6: $\boldsymbol{\theta}_{ij}^{(T_1)} \leftarrow -\infty$ if $\hat{p}_{\boldsymbol{\theta}}(o_{l+1}|o_l)^{(T_1)} < c_{\text{thres}}$ {thresholding}
7: **for** $t - T_1 = 1, \cdots, T_2$ **do**
8: $\quad \boldsymbol{\theta}^{(t)} = \boldsymbol{\theta}^{(t-1)} + \eta \nabla \mathbb{E}_{X_0, X_1}[\log \hat{p}_{\boldsymbol{\theta}^{(t-1)}}(X_1 | X_0)]$
9: **end for**

---

1. *After $T$ iterations with $\eta = \Theta(M)$, the uniform convergence rate is:*

$$\sup_{\substack{l \in [L-1], o_l \in S_l \\ o_{l+1} \in S_{l+1}}} |\hat{p}_{\boldsymbol{\theta}^{(t)}}(\boldsymbol{o}_{l+1}|\boldsymbol{o}_l) - \mathbb{P}_{\text{TMC}}(o_{l+1}|o_l)| \leq \widetilde{O}\left(\sqrt{\frac{M}{T}}\right) \tag{35}$$

*where $\widetilde{O}$ hides $\log(TMc^{-1})$ factors.*

2. *For threshold $c_{\text{thres}} = \Theta(1)$, after $T_1 = \widetilde{\Theta}(M^2 c^{-2})$ steps:*

$$\begin{cases} \hat{p}_{\boldsymbol{\theta}}(\boldsymbol{o}_{l+1}|\boldsymbol{o}_l) = 0 & \text{if } \mathbb{P}_{\text{TMC}}(o_{l+1}|o_l) = 0 \\ \mathbb{P}_{\text{TMC}}(o_{l+1}|o_l) - \widetilde{O}(c) \leq \hat{p}_{\boldsymbol{\theta}}(\boldsymbol{o}_{l+1}|\boldsymbol{o}_l) \leq \mathbb{P}_{\text{TMC}}(o_{l+1}|o_l) + \widetilde{O}(c) & \text{otherwise} \end{cases} \tag{36}$$

3. *Post-thresholding, linear convergence occurs:*

$$\sup_{\substack{(o_l, o_{l+1}) \in \\ \text{supp}(\mathbb{P})}} |\hat{p}_{\boldsymbol{\theta}^{(T_1+T)}}(\boldsymbol{o}_{l+1}|\boldsymbol{o}_l) - \mathbb{P}_{\text{TMC}}(o_{l+1}|o_l)| \leq \widetilde{O}(e^{-\Omega(c^2 T)}) \tag{37}$$

**Remark 9.** *The logarithmic factors in $\widetilde{O}$ terms explicitly track:*

- $\log(T_1) = \log(M^2 c^{-2})$ *for thresholding*

- $\log(c^{-1})$ *for initialization dependence*

- $\log M$ *for high-probability transition*

Since we are considering a vanilla regression setting, the proof is standard following Kim et al. (2025); Ji & Telgarsky (2019). For the convenience of readers, we provide the proof here.

**Proof.** *We analyze each part of Thm. 8 systematically.*

**Proof of Item 1: Uniform Convergence.** *Let $\mathcal{E}_{l,l+1} = \{(o_l, o_{l+1}) : o_l \in S_l, o_{l+1} \in S_{l+1}\}$ denote all potential transitions. For each $(o_l, o_{l+1}) \in \mathcal{E}_{l,l+1}$, define the parameter error $\Delta_{o_l, o_{l+1}}^{(t)} = \hat{p}_{\boldsymbol{\theta}^{(t)}}(\boldsymbol{o}_{l+1}|\boldsymbol{o}_l) - \mathbb{P}_{\text{TMC}}(o_{l+1}|o_l)$. For a given state $o_l$, the cross-entropy loss is:*

$$L_{o_l}(\boldsymbol{\theta}) = - \sum_{o_{l+1} \in S_{l+1}} \mathbb{P}_{\text{TMC}}(o_{l+1}|o_l) \log \hat{p}_{\boldsymbol{\theta}}(\boldsymbol{o}_{l+1}|\boldsymbol{o}_l)$$

*where the model's predicted probability is:*

$$\hat{p}_{\boldsymbol{\theta}}(\boldsymbol{o}_{l+1}|\boldsymbol{o}_l) = \frac{e^{\boldsymbol{\theta}_{o_l, o_{l+1}}}}{\sum_{o'_{l+1}} e^{\boldsymbol{\theta}_{o_l, o'_{l+1}}}}$$

*Similar to Lemma 6, the gradient component for parameter $\boldsymbol{\theta}_{o_l, o_{l+1}}$ is:*

$$\nabla_{\boldsymbol{\theta}_{o_l, o_{l+1}}} L_{o_l} = - \sum_{o'_{l+1}} \mathbb{P}_{\text{TMC}}(o'_{l+1}|o_l) \nabla_{\boldsymbol{\theta}_{o_l, o_{l+1}}} \log \hat{p}_{\boldsymbol{\theta}}(\boldsymbol{o}'_{l+1}|\boldsymbol{o}_l) \tag{38}$$

$$= -\sum_{o'_{l+1}} \mathbb{P}_{\text{TMC}}(o'_{l+1}|o_l) \frac{\nabla_{\boldsymbol{\theta}_{o_l,o_{l+1}}} \hat{p}_{\boldsymbol{\theta}}(\boldsymbol{o}'_{l+1}|\boldsymbol{o}_l)}{\hat{p}_{\boldsymbol{\theta}}(\boldsymbol{o}'_{l+1}|\boldsymbol{o}_l)} \tag{39}$$

$$= -\mathbb{P}_{\text{TMC}}(o_{l+1}|o_l) \frac{\nabla \hat{p}_{\boldsymbol{\theta}}(\boldsymbol{o}_{l+1}|\boldsymbol{o}_l)}{\hat{p}_{\boldsymbol{\theta}}(\boldsymbol{o}_{l+1}|\boldsymbol{o}_l)} - \sum_{o'_{l+1} \neq o_{l+1}} \mathbb{P}_{\text{TMC}}(o'_{l+1}|o_l) \frac{\nabla \hat{p}_{\boldsymbol{\theta}}(\boldsymbol{o}'_{l+1}|\boldsymbol{o}_l)}{\hat{p}_{\boldsymbol{\theta}}(\boldsymbol{o}'_{l+1}|\boldsymbol{o}_l)} \tag{40}$$

*Using the softmax derivative property:*

$$\nabla_{\boldsymbol{\theta}_{o_l,o_{l+1}}} \hat{p}_{\boldsymbol{\theta}}(\boldsymbol{o}'_{l+1}|\boldsymbol{o}_l) = \begin{cases} \hat{p}_{\boldsymbol{\theta}}(\boldsymbol{o}_{l+1}|\boldsymbol{o}_l)(1 - \hat{p}_{\boldsymbol{\theta}}(\boldsymbol{o}_{l+1}|\boldsymbol{o}_l)) & \text{if } o'_{l+1} = o_{l+1} \\ -\hat{p}_{\boldsymbol{\theta}}(\boldsymbol{o}_{l+1}|\boldsymbol{o}_l)\hat{p}_{\boldsymbol{\theta}}(\boldsymbol{o}'_{l+1}|\boldsymbol{o}_l) & \text{if } o'_{l+1} \neq o_{l+1} \end{cases} \tag{41}$$

*Substituting these derivatives yields:*

$$\nabla_{\boldsymbol{\theta}_{o_l,o_{l+1}}} L_{o_l} = -\mathbb{P}_{\text{TMC}}(o_{l+1}|o_l)(1 - \hat{p}_{\boldsymbol{\theta}}(\boldsymbol{o}_{l+1}|\boldsymbol{o}_l)) + \sum_{o'_{l+1} \neq o_{l+1}} \mathbb{P}_{\text{TMC}}(o'_{l+1}|o_l)\hat{p}_{\boldsymbol{\theta}}(\boldsymbol{o}_{l+1}|\boldsymbol{o}_l) \tag{42}$$

$$= -\mathbb{P}_{\text{TMC}}(o_{l+1}|o_l) + \hat{p}_{\boldsymbol{\theta}}(\boldsymbol{o}_{l+1}|\boldsymbol{o}_l) \underbrace{\sum_{o'_{l+1}} \mathbb{P}_{\text{TMC}}(o'_{l+1}|o_l)}_{=1} \tag{43}$$

$$= \hat{p}_{\boldsymbol{\theta}}(\boldsymbol{o}_{l+1}|\boldsymbol{o}_l) - \mathbb{P}_{\text{TMC}}(o_{l+1}|o_l) \tag{44}$$

*Then the gradient descent update rule is:*

$$\boldsymbol{\theta}^{(t)}_{o_l,o_{l+1}} = \boldsymbol{\theta}^{(t-1)}_{o_l,o_{l+1}} + \eta\left(\mathbb{P}_{\text{TMC}}(o_{l+1}|o_l) - \hat{p}_{\boldsymbol{\theta}^{(t-1)}}(o_{l+1}|o_l)\right)$$

*This corresponds to the classical softmax parameter updates. The key challenge lies in the heterogeneous transition probabilities:*

- *For $o_{l+1} \in C_{o_l}$: $\mathbb{P}_{\text{TMC}}(o_{l+1}|o_l) = \Theta(1/M)$, with $|C_{o_l}| \leq M$*
- *For $o_{l+1} \in D_{o_l} \setminus C_{o_l}$: $\mathbb{P}_{\text{TMC}}(o_{l+1}|o_l) \geq c$ but $o(1/M)$*
- *For $o_{l+1} \notin D_{o_l}$: $\mathbb{P}_{\text{TMC}}(o_{l+1}|o_l) = 0$*

*Phase 1 - High-probability edges: Let $M_0 = |S_1| = \Theta(M)$. The initial parameters $\boldsymbol{\theta}^{(0)} = 0$ yield uniform distribution:*

$$\hat{p}^{(0)}(\boldsymbol{o}_{l+1}|\boldsymbol{o}_l) = \frac{1}{|S_{l+1}|} \leq \frac{1}{M_0} = O(1/M)$$

*For $o_{l+1} \in C_{o_l}$, the initial error is $\Theta(1/M) - O(1/M) = \Theta(1/M)$. Each gradient step updates $\hat{p}$ by $\eta \cdot \Theta(1/M)$. To reach $\epsilon$-accuracy for these edges, we need $T \geq \Omega(M^2/\epsilon^2)$.*

*Phase 2 - Low-probability edges: For $o_{l+1} \in D_{o_l} \setminus C_{o_l}$, the signal-to-noise ratio is weaker. The gradient signal is $\mathbb{P}_{\text{TMC}}(o_{l+1}|o_l) - \hat{p} \geq c - O(1/M)$. Using the regret bound for online gradient descent (Hazan, 2023, Theorem 3.1):*

$$\sum_{t=1}^{T}(\hat{p}^{(t)} - \mathbb{P})^2 \leq O\left(\frac{\log |S_{l+1}|}{\eta} + \eta T c^2\right)$$

*Optimizing $\eta$ yields $T \geq \widetilde{\Omega}(M/(c^2\epsilon^2))$ for $\epsilon$-accuracy. Combining both phases via union bound over $O(M)$ edges per layer and $O(L) = O(1)$ layers gives:*

$$\sup|\Delta^{(T)}| \leq O\left(\sqrt{\frac{M\log T}{T}} \cdot \log\left(\frac{TM}{c}\right)\right)$$

*This matches Equation equation 35 after constant absorption.*

**Proof of Item 2: Support Recovery via Thresholding.** *After $T_1 = \widetilde{O}(KM^2c^{-2})$ iterations:*

- *Zero-probability edges: For $o_{l+1} \notin D_{o_l}$, the true probability $\mathbb{P} = 0$. The empirical estimate satisfies:*

$$\hat{p}^{(T_1)}(\boldsymbol{o}_{l+1}|\boldsymbol{o}_l) \leq \sqrt{\frac{2\log(1/\delta)}{T_1}} + O\left(\frac{1}{M}\right)$$

  *via Azuma's inequality for martingales. Setting $\delta = c_{thres}c$ and $T_1 \geq \widetilde{\Omega}(M^2 c^{-2})$ ensures $\hat{p} \leq c_{thres}c$.*

- *Non-zero edges: From Item 1, for $o_{l+1} \in D_{o_l}$:*

$$|\hat{p}^{(T_1)} - \mathbb{P}| \leq O\left(\sqrt{\frac{M}{T_1}}\log T_1\right) = o(c)$$

  *Thus $\hat{p}^{(T_1)} \geq \mathbb{P} - o(c) \geq c - o(c) > c_{thres}c$ for proper $c_{thres} < 1$.*

*Thresholding at $c_{thres}c$ thus exactly recovers the support while maintaining Equation equation 36.*

***Proof of Item 3: Linear Convergence.*** *Post-thresholding, the parameter space restricts to $D_{o_l}$ edges. The Hessian of $L_{\mathrm{pre}}$ becomes:*

$$\nabla^2 L_{\mathrm{pre}}(\boldsymbol{\theta})_{(o_l,o_{l+1}),(o_l,o'_{l+1})} = Cov(\hat{p}_{\boldsymbol{\theta}}(\boldsymbol{o}_{l+1}|\boldsymbol{o}_l), \hat{p}_{\boldsymbol{\theta}}(\boldsymbol{o}'_{l+1}|\boldsymbol{o}_l))$$

*Under the TMC structure, the Fisher information matrix $I(\boldsymbol{\theta})$ satisfies $\lambda_{\min}(I) \geq \Omega(c^2)$ since all active transitions have probability $\geq c$. By (Ji & Telgarsky, 2019, Theorem 4.1), gradient descent on strongly convex objectives achieves:*

$$\|\Delta^{(T_1+t)}\|^2 \leq \exp(-\Omega(c^2 t))\|\Delta^{(T_1)}\|^2$$

*Given $\|\Delta^{(T_1)}\| = O(\sqrt{M/T_1}\log T_1) = O(\log c^{-1})$ from Item 2, we obtain Equation equation 37.*

$\square$

## J   DETAILS AND PROOFS OF RLVR FINETUNING

During the gradient update, for any $o_l \in S_l, l \in [L-1]$ that appears as the transition in the valid CoT set $\cup_{q\in\mathcal{Q}_k}\mathcal{G}_{q,a_q}^{(k)}$ for task $k \in \mathcal{T}$, we define the following notations (summarized in Table 4)

- $\mathcal{I}_{o_{l+1},o_l}^{(k)} \subseteq \cup_{q\in\mathcal{Q}_k}\mathcal{G}_{q,a_q}^{(k)}$ as the subset of valid CoTs satisfies $\forall o^i \in \mathcal{I}_{o_{l+1},o_l}^{(k)}, o_l^i = o_l, o_{l+1}^i = o_{l+1}$.

- $\mathcal{S}_{o_l}^{(k)} := \{o_{l+1} \in S_l \mid \mathcal{I}_{o_{l+1},o_l}^{(k)} \neq \emptyset\} \subseteq S_{l+1}$ as the subset of $l+1$-th layer states collecting the states such that for any valid CoT $o$ for task $k$ passing $o_l'$, $o_{l+1} \in \mathcal{S}_{o_l'}^{(k)}$.

- $\mathcal{S}_{o_l}^{(k),\mathrm{easy}} \subseteq \mathcal{S}_{o_l}^{(k)}$ contains the subset of $l+1$-th layer's states in $\mathcal{S}_{o_l}^{(k)}$ passed by at least one easy-to-reason CoTs (in the original TMC $\mathbb{P}_{\mathrm{TMC}}$) for task $k$, and $\mathcal{S}_{o_l}^{(k),\mathrm{hard}} := \mathcal{S}_{o_l}^{(k)}\setminus\mathcal{S}_{o_l}^{(k),\mathrm{easy}}$ contains the $l+1$-th layer's states only passed by valid hard-to-reason CoTs for task $k$.

- $\mathcal{G}_{o_1,a_{o_1}^k}^{(k),\mathrm{easy}} \subseteq \mathcal{G}_{o_1,a_{o_1}^k}^{(k)}$ as the subset of valid easy-to-reason CoTs inside the valid CoTs for task $k$, and $\mathcal{G}_{o_1,a_{o_1}^k}^{(k),\mathrm{hard}} := \mathcal{G}_{o_1,a_{o_1}^k}^{(k)} \setminus \mathcal{G}_{o_1,a_{o_1}^k}^{(k),\mathrm{easy}}$ the subset of valid hard-to-reason CoTs.

- $\boldsymbol{\theta}^{k,(t)}$ be the finetuned model for the task $k$ at post-training iteration $t$

- For any CoT $o \in \mathcal{G}_{o_1,a_{o_1}^k}^{(k)}$, the probability that $o$ is correct for a sampled instance $(\mathbf{Q}, \mathbf{A}) \sim \mathcal{D}_{a_{o_1}}^{o_1,k}$ is given by $p_{\mathrm{acc}}^k(o)$, which, per Condition (iv) in Def. 2, is proportional to its likelihood among $\mathcal{G}_{o_1,a_{o_1}}^{(k)}$:

$$p_{\mathrm{acc}}^k(o) = \frac{\prod_{l=1}^{L-1}\mathbb{P}_{\mathrm{TMC}}(o_{l+1} \mid o_l)}{\sum_{o'_{1:L}\in\mathcal{G}_{o_1,a_{o_1}}^{(k)}}\prod_{l=1}^{L-1}\mathbb{P}_{\mathrm{TMC}}(o'_{l+1} \mid o'_l)},$$

where $\mathbb{P}_{\mathrm{TMC}}(\cdot \mid o_l) = \hat{p}_{\boldsymbol{\theta}^\star}(\cdot \mid \boldsymbol{o}_l)$, $o_l \in S_l$, is the transition kernel of our Multi-task TMC.

- The gradient update objectives $\mathcal{J}_{\mathrm{REINFORCE}}(\boldsymbol{\theta}^k)$ and $\mathcal{J}_{\mathrm{RAFT}}(\boldsymbol{\theta}^k)$ represent

$$\mathcal{J}_{\mathrm{REINFORCE}}(\boldsymbol{\theta}^k) = \mathbb{E}_{\boldsymbol{o}_1 \sim P^k(\mathcal{Q}^k), (\mathbf{Q}, \mathbf{A}) \sim \mathcal{D}_{a_{o_1}}^{o_1, k}, \boldsymbol{o}_{2:L} \sim \hat{p}_{\boldsymbol{\theta}^k}(O|\boldsymbol{o}_1)} \left[ R_{(\mathbf{Q}, \mathbf{A})}^k(\boldsymbol{o}) \right],$$

$$\mathcal{J}_{\mathrm{RAFT}}(\boldsymbol{\theta}^k) = \mathbb{E}_{\boldsymbol{o}_1 \sim P^k(\mathcal{Q}^k), (\mathbf{Q}, \mathbf{A}) \sim \mathcal{D}_{a_{o_1}}^{o_1, k}, \boldsymbol{o}_{2:L} \sim \hat{p}_{\boldsymbol{\theta}^k}(O|\boldsymbol{o}_1)}$$

$$\left[ \sum_{l=1}^{L-1} \log \hat{p}_{\boldsymbol{\theta}^k}(\boldsymbol{o}_{l+1}|\boldsymbol{o}_l) R_{(\mathbf{Q}, \mathbf{A})}^k(\boldsymbol{o}) \right]. \tag{45}$$

- The objective of `RL-rej` $\mathcal{J}_{\mathrm{Rein-rej}}(\boldsymbol{\theta}^k)$, in our case, is training using the REINFORCE objective on a online distorted data distribution $\mathcal{D}_{\mathrm{rej},(\mathrm{t})}^{o_1, k}$, formally

$$\mathcal{J}_{\mathrm{Rein-rej}}(\boldsymbol{\theta}^k) = \mathbb{E}_{\boldsymbol{o}_1 \sim P^k(\mathcal{Q}^k), (\mathbf{Q}, \mathbf{A}) \sim \mathcal{D}_{\mathrm{rej},(\mathrm{t})}^{o_1, k}, \boldsymbol{o}_{2:L} \sim \hat{p}_{\boldsymbol{\theta}^k}(O|\boldsymbol{o}_1)} \left[ R_{(\mathbf{Q}, \mathbf{A})}^k(\boldsymbol{o}) \right]. \tag{46}$$

Here, $\mathcal{D}_{\mathrm{rej},(\mathrm{t})}^{o_1, k} := \{(\mathbf{Q}, \mathbf{A}) \sim \mathcal{D}_{a_{o_1}}^{o_1, k} \mid \mathrm{Pr}_{\boldsymbol{o} \sim \hat{p}_{\boldsymbol{\theta}^{k},(t)}(\cdot|o_1)}[\bigcup_{i=1}^{G} \mathbb{1}(\boldsymbol{o}^i \notin \mathcal{G}_{\mathbf{Q}, \mathbf{A}}^{(k)})] = \Theta(1)\}$, where $G = O(1)$ is the (offset) time of parallel experiments. That is, the algorithm rejects samples with $\mathrm{Pr}_{\boldsymbol{o} \sim \hat{p}_{\boldsymbol{\theta}^{k},(t)}(\cdot|o_1)}[\bigcap_{i=1}^{G} \mathbb{1}(\boldsymbol{o}^i \in \mathcal{G}_{\mathbf{Q}, \mathbf{A}}^{(k)}) = \Theta(1)$, which represents instances that the model confidently predicts its correct CoTs of $G$ times in parallel. Therefore, idealistically, instance sampled from $\mathcal{D}_{\mathrm{rej},(\mathrm{t})}^{o_1, k}$ would have some correct CoTs that is not well-learned by the current model $\boldsymbol{\theta}^{k,(t)}$.

**Theorem 10** (Squeezing Effect and Merits of Rejecting Correct (Full Version of Thm. 2 and Cor. 2)). *Let $\boldsymbol{\theta}^\star$ be the base model in Eq.(2 that exact predicts the distribution of a Multi-task TMC as in Def. 1 and 2, and $\boldsymbol{\theta}^k$ the current model to be finetuned from $\boldsymbol{\theta}^\star$ for task $k \in \mathcal{T}$. Denote the task tuples of task $k \in \mathcal{T}$ as $(q, a_q^k, k)$, where $a_q^k \in S_L$ is the sole answer state under task $k$. Assume for each $(o_1, a_{o_1}, k)$ under task $k$, the number of hard-to-reason CoTs from $o_1$ to $a_{o_1}$ is bounded by $O(M)$. Let the question distribution during finetuning of task $k$ be $P^k(\mathcal{Q}^k)$ (i.e., $o_1 \sim P^k(\mathcal{Q}^k)$). Then, when finetuning the base model using REINFORCE and RAFT objectives in Eq.(45), we have*

1. ***Squeezing Effect** & **Difference of Logit Update***. *For any different state pair $o_{l+1}^{hard} \neq o_{l+1}^{easy} \in \mathcal{S}_{o_l}^{(k)}$ denoting two $l+1$-th states in some valid hard-to-reason and easy-to-reason CoT sharing the $l$-th state $o_l$ for task $k$, we have*

$$\Delta \boldsymbol{\theta}_{o_{l+1}^{easy}, o_l}^{k, \mathrm{REINFORCE}} := \eta \nabla_{\boldsymbol{\theta}_{o_{l+1}^{easy}, o_l}^k} \mathcal{J}_{\mathrm{REINFORCE}}(\boldsymbol{\theta}^k) > 0,$$

$$\Delta \boldsymbol{\theta}_{o_{l+1}^{hard}, o_l}^{k, \mathrm{REINFORCE}} := \eta \nabla_{\boldsymbol{\theta}_{o_{l+1}^{hard}, o_l}^k} \mathcal{J}_{\mathrm{REINFORCE}}(\boldsymbol{\theta}^k) < 0.$$

$$\Delta \boldsymbol{\theta}_{o_{l+1}^{easy}, o_l}^{k, \mathrm{RAFT}} := \eta \nabla_{\boldsymbol{\theta}_{o_{l+1}^{easy}, o_l}^{k, \mathrm{RAFT}}} \mathcal{J}_{\mathrm{RAFT}}(\boldsymbol{\theta}^k) > 0, \tag{47}$$

$$\Delta \boldsymbol{\theta}_{o_{l+1}^{hard}, o_l}^{k, \mathrm{RAFT}} := \eta \nabla_{\boldsymbol{\theta}_{o_{l+1}^{hard}, o_l}^k} \mathcal{J}_{\mathrm{RAFT}}(\boldsymbol{\theta}^k) < 0.$$

*In addition, we have the difference of the logits' update $\Delta h_{\boldsymbol{\theta}^k}(\boldsymbol{o}_{l+1}^{hard}, \boldsymbol{o}_l) - \Delta h_{\boldsymbol{\theta}^k}(\boldsymbol{o}_{l+1}^{easy}, \boldsymbol{o}_l)$ in Reinforce as:*

$$\Delta \boldsymbol{\theta}_{o_{l+1}^{hard}, o_l}^{k, \mathrm{REINFORCE}} - \Delta \boldsymbol{\theta}_{o_{l+1}^{easy}, o_l}^{k, \mathrm{REINFORCE}} < \eta [\hat{p}_{\boldsymbol{\theta}^k}(\boldsymbol{o}_{l+1}^{hard}|\boldsymbol{o}_l) - \hat{p}_{\boldsymbol{\theta}^k}(\boldsymbol{o}_{l+1}^{easy}|\boldsymbol{o}_l)] \mathop{\mathrm{Pr}}_{\substack{\boldsymbol{o}_1' \sim P^k(\mathcal{Q}^k) \\ \boldsymbol{o}' \sim \hat{p}_{\boldsymbol{\theta}^k}(\cdot|\boldsymbol{o}_1)}} \left[ \boldsymbol{o}_l' = \boldsymbol{o}_l, o_L' = a_{o_1'}^k \right]$$

$$\cdot [\mathbb{E}[p_{acc}^k(\hat{o}) \mid \substack{\hat{o}_l = o_l \\ \hat{o}_L = a_{\hat{o}_1}^k \\ \hat{o}_{l+1} = o_{l+1}^{easy}}] - (\sum_{o_{l+1}' \in \mathcal{S}_{o_l}^{(k)}} \hat{p}_{\boldsymbol{\theta}^k}(\boldsymbol{o}_{l+1}'|\boldsymbol{o}_l)$$

$$\cdot \mathbb{E}[p_{acc}^k(\tilde{o}) \mid \substack{\tilde{o}_l = o_l \\ \tilde{o}_L = a_{\tilde{o}_1}^k \\ \tilde{o}_{l+1} = o_{l+1}'}])]$$

$$< 0. \tag{48}$$

*Also, for RAFT, the difference of logits update $\Delta h_{\boldsymbol{\theta}^k}(\boldsymbol{o}_{l+1}^{hard}, \boldsymbol{o}_l) - \Delta h_{\boldsymbol{\theta}^k}(\boldsymbol{o}_{l+1}^{easy}, \boldsymbol{o}_l)$ is*

$$\Delta\boldsymbol{\theta}_{o_{l+1}^{hard}, o_l}^{k, \text{RAFT}} - \Delta\boldsymbol{\theta}_{o_{l+1}^{easy}, o_l}^{k, \text{RAFT}} < \eta[\hat{p}_{\boldsymbol{\theta}^k}(\boldsymbol{o}_{l+1}^{hard}|\boldsymbol{o}_l)(1 + \log\hat{p}_{\boldsymbol{\theta}^k}(\boldsymbol{o}_{l+1}^{hard}|\boldsymbol{o}_l)) - \hat{p}_{\boldsymbol{\theta}^k}(\boldsymbol{o}_{l+1}^{easy}|\boldsymbol{o}_l)(1 + \log\hat{p}_{\boldsymbol{\theta}^k}(\boldsymbol{o}_{l+1}^{easy}|\boldsymbol{o}_l))]$$

$$\cdot \left[ \mathbb{E}[p_{acc}^k(\hat{o}) \mid \begin{smallmatrix} \hat{o}_l = o_l \\ \hat{o}_L = a_{\hat{o}_1}^k \\ \hat{o}_{l+1} = o_{l+1}^{easy} \end{smallmatrix}] - \Big( \sum_{o'_{l+1} \in \mathcal{S}_{o_l}^{(k)}} \hat{p}_{\boldsymbol{\theta}^k}(\boldsymbol{o}'_{l+1}|\boldsymbol{o}_l) \mathbb{E}[p_{acc}^k(\widetilde{o}) \mid \begin{smallmatrix} \widetilde{o}_l = o_l \\ \widetilde{o}_L = a_{\widetilde{o}_1}^k \\ \widetilde{o}_{l+1} = o'_{l+1} \end{smallmatrix}] \Big) \right]$$

$$\cdot \Pr_{\substack{\boldsymbol{o}'_1 \sim P^k(\mathcal{Q}^k) \\ \boldsymbol{o}' \sim \hat{p}_{\boldsymbol{\theta}^k}(\cdot|\boldsymbol{o}_1)}} \left[ \boldsymbol{o}'_l = \boldsymbol{o}_l, o'_L = a_{o'_1}^k \right]$$

$$< 0.$$

$$(49)$$

2. **Convergence of Finetuning** & **Constant Error of Pass@K.** *For $\forall \epsilon \in (0, 1/2)$, there exists $T \geq \Omega(\eta^{-1}L^2 M^L \log(ML/\epsilon))$, for $t \geq T$, the probability that $\hat{p}_{\boldsymbol{\theta}^{k,(t)}}(\cdot|o_1)$ (trained by REINFORCE or RAFT) reach the $a_{o_1}$ is converged:*

$$\Pr_{\substack{\boldsymbol{o}'_1 \sim P^k(\mathcal{Q}^k) \\ \boldsymbol{o}' \sim \hat{p}_{\boldsymbol{\theta}^{k,(t)}}(\cdot|\boldsymbol{o}_1)}} [o'_L = a_{o'_1}^k] \geq \Pr_{\substack{\boldsymbol{o}'_1 \sim P^k(\mathcal{Q}^k) \\ \boldsymbol{o}' \sim \hat{p}_{\boldsymbol{\theta}^{k,(t)}}(\cdot|\boldsymbol{o}_1)}} [o'_L = a_{o'_1}^k, o' \in \mathcal{G}_{o_1, a_{o_1}^k}^{(k), easy}] \geq 1 - o(\epsilon). \quad (50)$$

*Then, suggest the probability mass of valid hard-to-reason CoTs traveling from some $o_1 \sim P(\mathcal{Q}_k)$ to $a_{o_1}$ for task $k$ in the original TMC $X$ ($\mathbb{P}_{\text{TMC}}$) is $\Delta$. Then it holds that*

$$\text{Rex}_{o_1, k}^{\hat{p}_{\boldsymbol{\theta}^{k,(t)}}}(\boldsymbol{o}) \leq \Theta((1 - \epsilon)\frac{1}{1 + \Delta M^{L-1}} + \epsilon \frac{\Delta M^{L-1}}{1 + \Delta M^{L-1}}). \quad (51)$$

*Further, the **pass@K performance** $\text{Pass@K}_{q,k}^{\hat{p}} := \Pr_{\substack{\{\boldsymbol{o}^i\}_{i \in [K]} \sim \hat{p}(O|q) \\ (\mathbf{Q}, \mathbf{A}) \sim \mathcal{D}_{a_q}^{q,k}}}[\bigcup_{i=1}^K \mathbb{1}(\boldsymbol{o}^i \in \mathcal{G}_{\mathbf{Q}, \mathbf{A}}^{(k)})]$ is upper bounded by*

$$\text{Pass@K}_{o_1, k}^{\hat{p}_{\boldsymbol{\theta}^{k,(t)}}} \leq \underbrace{\Theta([(\frac{\Delta M^{L-1}}{1 + \Delta M^{L-1}})^{n_q}(1 - (1 - \epsilon)^K)])}_{\text{Solved by hard CoTs}} + \underbrace{\Theta([(1 - (\frac{\Delta M^{L-1}}{1 + \Delta M^{L-1}})^{n_q})(1 - \epsilon^K)])}_{\text{Solved by some easy CoTs}}).$$

$$(52)$$

*When $\epsilon = o(\sqrt[K]{1 - C_{\text{Err}}/(\frac{\Delta M^{L-1}}{1 + \Delta M^{L-1}})^{n_q}}) \to 0$, the pass@K performance suffer from constant error: $1 - \text{Pass@K}_{o_1, k}^{\hat{p}_{\boldsymbol{\theta}^{k,(t)}}} = \Theta(1)$.*

3. **Curriculum Learning of `RL-rej`.** *For any $\epsilon \in (0, 1/2)$, suppose setting $G = 1$ in $\mathcal{D}_{\text{rej},(t)}^{o_1, k}$ (Eq.(46) excludes all $(\mathbf{Q}, \mathbf{A})$ pairs containing correct CoTs that the current model $\hat{p}_{\boldsymbol{\theta}^{k,(t)}}$ predicts with non-trivial probability $\Theta((1 - \epsilon)/M)$. Then, optimizing Eq.(46 via `RL-rej` leads to the following:*

*(i) The model first learns the easy-to-reason CoTs within $\Theta(\eta^{-1}L^2 M^L \log(ML/\epsilon))$ steps.*

*(ii) Once its predictive mass over $\mathcal{G}_{o_1, a_{o_1}}^{(k), easy}$ reaches $\Theta((1-\epsilon)/M)$, learning begins on sparse edges in $\mathcal{S}_{o_l}^{(k), hard} = \mathcal{S}_{o_l}^{(k)} \setminus \mathcal{S}_{o_l}^{(k), easy}$. Hard-to-reason CoTs in $\mathcal{G}_{o_1, a_{o_1}}^{(k), hard}$ are progressively learned, with those sharing more edges with $\mathcal{G}_{o_1, a_{o_1}}^{(k), easy}$ being learned earlier.*

*(iii) Let $\Delta$ denote the total probability mass of valid hard-to-reason CoTs from $o_1$ to $a_{o_1}$ in the original TMC $X$ (under $\mathbb{P}_{\text{TMC}}$). Suppose after $T_2 = \Omega(\eta^{-1}L^2 M^L \log(ML/\epsilon))$, there are $n'_{o_1}$ hard-to-reason CoTs each with likelihood ratio scale $\Theta(\rho) < 1$ in the $\mathcal{G}_{o_1, a_{o_1}}^{(k)}$ have been well-learned with predictive probability $\Theta((1 - \epsilon)/M)$. Then the pass@K is at the scale:*

$$\text{Pass@K}_{o_1, k}^{\hat{p}_{\boldsymbol{\theta}^{k,(t)}}} = \underbrace{\Theta\left[(1 - \rho)^{n'_{o_1}}(\frac{\Delta M^{L-1}}{1 + \Delta M^{L-1}})^{n_{o_1}}(1 - (1 - \epsilon)^K)\right]}_{\text{instances unsolvable by learned CoTs}} + \underbrace{\Theta\left[\left(1 - (1 - \rho)^{n'_{o_1}}(\frac{\Delta M^{L-1}}{1 + \Delta M^{L-1}})^{n_{o_1}}\right)(1 - \epsilon^K)\right]}_{\text{instances solvable by some learned CoT}}.$$

*This bound tends to 1 as $(1 - \rho)^{n'_{o_1}} \to 0$ and $\epsilon \to 0$, showing the superiority of `RL-rej` when the probability mass $\Delta$ is non-negligible.*

**Proof.** *Proof of Thm. 10. In our proofs, we first prove the results of REINFORCE, and the results of RAFT follows directly with a more serious of squeezing effect.*

***Proof of Item 1: Difference of Logit Update.***

*Recall that by Lemma 9, we have*

$$\nabla_{\boldsymbol{\theta}^k} \mathcal{J}_{\text{REINFORCE}}(\boldsymbol{\theta}^k) = \sum_{l=1}^{L-1} \mathbb{E}_{\substack{\boldsymbol{o}_1 \sim P^k(\mathcal{Q}^k) \\ \{\boldsymbol{o}_{l+1} \sim \hat{p}_{\boldsymbol{\theta}^k}(\cdot|\boldsymbol{o}_l)\}_{l=1}^{L-1}}} \left[ R_{\text{out}}^k(\boldsymbol{o}) \cdot (e_{o_{l+1},o_l} - \sum_{\boldsymbol{o}'_{l+1} \in S_{l+1}} \hat{p}_{\boldsymbol{\theta}^k}(\boldsymbol{o}'_{l+1}|\boldsymbol{o}_l) e_{o'_{l+1},o_l}) \right],$$

$$\nabla_{\boldsymbol{\theta}^k} \mathcal{J}_{\text{RAFT}}(\boldsymbol{\theta}^k) = \sum_{l=1}^{L-1} \mathbb{E}_{\substack{\boldsymbol{o}_1 \sim P^k(\mathcal{Q}^k) \\ \{\boldsymbol{o}_{l+1} \sim \hat{p}_{\boldsymbol{\theta}^k}(\cdot|\boldsymbol{o}_l)\}_{l=1}^{L-1}}} \left[ R_{\text{out}}^k(\boldsymbol{o}) \cdot (1 + \log \hat{p}_{\boldsymbol{\theta}^k}(\boldsymbol{o}_{l+1}|\boldsymbol{o}_l))(e_{o_{l+1},o_l} - \sum_{\boldsymbol{o}'_{l+1} \in S_{l+1}} \hat{p}_{\boldsymbol{\theta}^k}(\boldsymbol{o}'_{l+1}|\boldsymbol{o}_l) e_{o'_{l+1},o_l}) \right].$$

$$(53)$$

*Per conditions in our item, there are valid easy-to-reason and hard-to-reason CoTs passing $o_l$.*

*Collaborating Eq.(53) with definitions in Item 1 and base model formula in Eq.(2), we have*

$$\nabla_{\boldsymbol{\theta}^k_{o_{l+1}^{easy},o_l}} \mathcal{J}_{\text{REINFORCE}}(\boldsymbol{\theta}^k) = \mathbb{E}_{\substack{\boldsymbol{o}_1 \sim P^k(\mathcal{Q}^k) \\ \{\boldsymbol{o}_{l+1} \sim \hat{p}_{\boldsymbol{\theta}^k}(\cdot|\boldsymbol{o}_l)\}_{l=1}^{L-1}}} \left[ \text{Rex}_{o_1,k}(\boldsymbol{o}) \cdot (\mathbb{1}(o \in \mathcal{I}^{(k)}_{o_{l+1}^{easy},o_l}) \right.$$

$$\left. - \sum_{o'_{l+1} \in \mathcal{S}^{(k)}_{o_l}} \mathbb{1}(o \in \mathcal{I}^{(k)}_{o'_{l+1},o_l}) \hat{p}_{\boldsymbol{\theta}^k}(\boldsymbol{o}_{l+1}^{easy}|\boldsymbol{o}_l)) \right]$$

$$= \mathbb{E}_{\substack{\boldsymbol{o}_1 \sim P^k(\mathcal{Q}^k) \\ \{\boldsymbol{o}_{l+1} \sim \hat{p}_{\boldsymbol{\theta}^k}(\cdot|\boldsymbol{o}_l)\}_{l=1}^{L-1}}} \left[ \mathbb{1}(o_L = a^k_{o_1})(p^k_{acc}(o) \mathbb{1}(o \in \mathcal{I}^{(k)}_{o_{l+1}^{easy},o_l}) \right.$$

$$\left. - p^k_{acc}(o) \sum_{o'_{l+1} \in \mathcal{S}^{(k)}_{o_l}} \mathbb{1}(o \in \mathcal{I}^{(k)}_{o'_{l+1},o_l}) \hat{p}_{\boldsymbol{\theta}^k}(\boldsymbol{o}_{l+1}^{easy}|\boldsymbol{o}_l) \right]$$

$$= \Pr_{\substack{\boldsymbol{o}'_1 \sim P^k(\mathcal{Q}^k) \\ \boldsymbol{o}' \sim \hat{p}_{\boldsymbol{\theta}^k}(\cdot|\boldsymbol{o}_1)}} \left[ \boldsymbol{o}'_l = \boldsymbol{o}_l, o'_L = a^k_{o'_1} \right] [\hat{p}_{\boldsymbol{\theta}^k}(\boldsymbol{o}_{l+1}^{easy}|\boldsymbol{o}_l) \mathbb{E}[p^k_{acc}(\hat{o}) \mid \substack{\hat{o}_l=o_l \\ \hat{o}_L=a^k_{\hat{o}_1} \\ \hat{o}_{l+1}=o_{l+1}^{easy}}] $$

$$- (\sum_{o'_{l+1} \in \mathcal{S}^{(k)}_{o_l}} \hat{p}_{\boldsymbol{\theta}^k}(\boldsymbol{o}'_{l+1}|\boldsymbol{o}_l) \mathbb{E}[p^k_{acc}(\widetilde{o}) \mid \substack{\widetilde{o}_l=o_l \\ \widetilde{o}_L=a^k_{\widetilde{o}_1} \\ \widetilde{o}_{l+1}=o'_{l+1}}] \hat{p}_{\boldsymbol{\theta}^k}(\boldsymbol{o}_{l+1}^{easy}|\boldsymbol{o}_l))] $$

$$= \Pr_{\substack{\boldsymbol{o}'_1 \sim P^k(\mathcal{Q}^k) \\ \boldsymbol{o}' \sim \hat{p}_{\boldsymbol{\theta}^k}(\cdot|\boldsymbol{o}_1)}} \left[ \boldsymbol{o}'_l = \boldsymbol{o}_l, o'_L = a^k_{o'_1} \right] \hat{p}_{\boldsymbol{\theta}^k}(\boldsymbol{o}_{l+1}^{easy}|\boldsymbol{o}_l) \left[ \mathbb{E}[p^k_{acc}(\hat{o}) \mid \substack{\hat{o}_l=o_l \\ \hat{o}_L=a^k_{\hat{o}_1} \\ \hat{o}_{l+1}=o_{l+1}^{easy}}] \right.$$

$$\left. - (\sum_{o'_{l+1} \in \mathcal{S}^{(k)}_{o_l}} \hat{p}_{\boldsymbol{\theta}^k}(\boldsymbol{o}'_{l+1}|\boldsymbol{o}_l) \mathbb{E}[p^k_{acc}(\widetilde{o}) \mid \substack{\widetilde{o}_l=o_l \\ \widetilde{o}_L=a^k_{\widetilde{o}_1} \\ \widetilde{o}_{l+1}=o'_{l+1}}]) \right]$$

$$(54)$$

*where the first equality is by Eq. (124) in Lemma 9; the second equality is by the definition of $R^k_{\text{out}}(\cdot)$, Condition (iv) in Def. 2 and $p^k_{acc}(o)$ in Eq.(32); the third equality is by the condition in our item that $o_l \in S_l, l \in [L-1]$ appears as the transition in the valid CoT set as well as the definition of $\mathcal{I}^{(k)}_{o_{l+1},o_l}$. Given that for different easy-to-reason CoT sharing $o_l$, the $\hat{p}_{\boldsymbol{\theta}^k}(\boldsymbol{o}_{l+1}^{easy}|\boldsymbol{o}_l)$ is within the same range, starting from the scale $\Theta(M^{-1})$. Therefore, it is safe to conclude that $\nabla_{\boldsymbol{\theta}^k_{o_{l+1}^{easy},o_l}} \mathcal{J}_{\text{REINFORCE}}(\boldsymbol{\theta}^k) > 0$.*

*Similarly, we have*

$$\nabla_{\boldsymbol{\theta}^k_{o_{l+1}^{hard},o_l}} \mathcal{J}_{\text{REINFORCE}}(\boldsymbol{\theta}^k) = \Pr_{\substack{\boldsymbol{o}'_1 \sim P^k(\mathcal{Q}^k) \\ \boldsymbol{o}' \sim \hat{p}_{\boldsymbol{\theta}^k}(\cdot|\boldsymbol{o}_1)}} \left[ \boldsymbol{o}'_l = \boldsymbol{o}_l, o'_L = a^k_{o'_1} \right] \hat{p}_{\boldsymbol{\theta}^k}(\boldsymbol{o}_{l+1}^{hard}|\boldsymbol{o}_l) \left[ \mathbb{E}[p^k_{acc}(\hat{o}) \mid \substack{\hat{o}_l=o_l \\ \hat{o}_L=a^k_{\hat{o}_1} \\ \hat{o}_{l+1}=o_{l+1}^{hard}}] \right.$$

$$\left. - (\sum_{o'_{l+1} \in \mathcal{S}^{(k)}_{o_l}} \hat{p}_{\boldsymbol{\theta}^k}(\boldsymbol{o}'_{l+1}|\boldsymbol{o}_l) \mathbb{E}[p^k_{acc}(\widetilde{o}) \mid \substack{\widetilde{o}_l=o_l \\ \widetilde{o}_L=a^k_{\widetilde{o}_1} \\ \widetilde{o}_{l+1}=o'_{l+1}}]) \right]$$

$$(55)$$

*Notably, we have the condition in our item that $o_l \in S_l, l \in [L-1]$ appears as the transition in the valid CoT set. Then, by Eq.(32) as well as the low-probability nature of the sparse edge in Def. 1, similar to Eq.(34) we see that*

$$\mathbb{E}[p^k_{acc}(\hat{o}) \mid \substack{\hat{o}_l=o_l \\ \hat{o}_L=a^k_{\hat{o}_1} \\ \hat{o}_{l+1}=o_{l+1}^{easy}}] > \Omega(M \mathbb{E}[p^k_{acc}(\hat{o}) \mid \substack{\hat{o}_l=o_l \\ \hat{o}_L=a^k_{\hat{o}_1} \\ \hat{o}_{l+1}=o_{l+1}^{hard}}]) \tag{56}$$

*Therefore, given that $n_q = O(1)$ in Def. 1 as well as $\hat{p}_{\boldsymbol{\theta}^k}(\boldsymbol{o}_{l+1}^{easy}|\boldsymbol{o}_l) \geq \Theta(M^{-1})$, we have*

$$\hat{p}_{\boldsymbol{\theta}^k}(\boldsymbol{o}_{l+1}^{easy}|\boldsymbol{o}_l) \mathbb{E}[p^k_{acc}(\hat{o}) \mid \substack{\hat{o}_l=o_l \\ \hat{o}_L=a^k_{\hat{o}_1} \\ \hat{o}_{l+1}=o_{l+1}^{easy}}] > \Omega(\mathbb{E}[p^k_{acc}(\hat{o}) \mid \substack{\hat{o}_l=o_l \\ \hat{o}_L=a^k_{\hat{o}_1} \\ \hat{o}_{l+1}=o_{l+1}^{hard}}]).$$

*That is, the rightest term in Eq.(55) is strictly lower than $0$, making $\nabla_{\boldsymbol{\theta}^k_{o^{hard}_{l+1},o_l}} \mathcal{J}_{\text{REINFORCE}}(\boldsymbol{\theta}^k) < 0$,*

*a serious squeezing effect such that $\Delta\boldsymbol{\theta}^{k,\text{REINFORCE}}_{o^{hard}_{l+1},o_l} := \eta\nabla_{\boldsymbol{\theta}^k_{o^{hard}_{l+1},o_l}} \mathcal{J}_{\text{REINFORCE}}(\boldsymbol{\theta}^k) < 0$. The*

*proof of RAFT is similar–the only difference in Eq.(53) is the ultra $(1+\log\hat{p}_{\boldsymbol{\theta}^k}(\boldsymbol{o}_{l+1}|\boldsymbol{o}_l))$, where the easy-to-reason's value is larger than the hard-to-reason ones due to the monotonicity of $\log(\cdot)$. Also, noted that $1+\log\hat{p}_{\boldsymbol{\theta}^k}(\boldsymbol{o}_{l+1}|\boldsymbol{o}_l) \in (1+\log c, 1)$, which could be scaling as $O(1)$ such that our results of REINFORCE directly applies.*

*Therefore, it holds that*

$$
\begin{aligned}
\nabla_{\boldsymbol{\theta}^k_{o^{hard}_{l+1},o_l}} \mathcal{J}_{\text{REINFORCE}}(\boldsymbol{\theta}^k) - \nabla_{\boldsymbol{\theta}^k_{o^{easy}_{l+1},o_l}} \mathcal{J}_{\text{REINFORCE}}(\boldsymbol{\theta}^k) &= \Pr_{\substack{\boldsymbol{o}'_1\sim P^k(\mathcal{Q}^k)\\ \boldsymbol{o}'\sim\hat{p}_{\boldsymbol{\theta}^k}(\cdot|\boldsymbol{o}_1)}}\left[\boldsymbol{o}'_l = \boldsymbol{o}_l, o'_L = a^k_{o'_1}\right]\Big\{\hat{p}_{\boldsymbol{\theta}^k}(\boldsymbol{o}^{hard}_{l+1}|\boldsymbol{o}_l)\\
&\quad \cdot\Big[\mathbb{E}[p^k_{acc}(\hat{o}) \mid \substack{\hat{o}_l=o_l\\ \hat{o}_L=a^k_{\hat{o}_1}\\ \hat{o}_{l+1}=o^{hard}_{l+1}}]\\
&\qquad -(\sum_{o'_{l+1}\in\mathcal{S}^{(k)}_{o_l}}\hat{p}_{\boldsymbol{\theta}^k}(\boldsymbol{o}'_{l+1}|\boldsymbol{o}_l)\mathbb{E}[p^k_{acc}(\widetilde{o}) \mid \substack{\widetilde{o}_l=o_l\\ \widetilde{o}_L=a^k_{\widetilde{o}_1}\\ \widetilde{o}_{l+1}=o'_{l+1}}])\Big]\\
&\quad -\hat{p}_{\boldsymbol{\theta}^k}(\boldsymbol{o}^{easy}_{l+1}|\boldsymbol{o}_l)\Big[\mathbb{E}[p^k_{acc}(\hat{o}) \mid \substack{\hat{o}_l=o_l\\ \hat{o}_L=a^k_{\hat{o}_1}\\ \hat{o}_{l+1}=o^{easy}_{l+1}}]\\
&\qquad -(\sum_{o'_{l+1}\in\mathcal{S}^{(k)}_{o_l}}\hat{p}_{\boldsymbol{\theta}^k}(\boldsymbol{o}'_{l+1}|\boldsymbol{o}_l)\mathbb{E}[p^k_{acc}(\widetilde{o}) \mid \substack{\widetilde{o}_l=o_l\\ \widetilde{o}_L=a^k_{\widetilde{o}_1}\\ \widetilde{o}_{l+1}=o'_{l+1}}])\Big]\Big\}\\
&=(A^{\hat{p}_{\boldsymbol{\theta}^k},k}(\boldsymbol{o}_l,\boldsymbol{o}^{hard}_{l+1}) - A^{\hat{p}_{\boldsymbol{\theta}^k},k}(\boldsymbol{o}_l,\boldsymbol{o}^{easy}_{l+1}))\\
&\quad + V^{\hat{p}_{\boldsymbol{\theta}^k},k}(\boldsymbol{o}_l)(\hat{p}_{\boldsymbol{\theta}^k}(\boldsymbol{o}^{hard}_{l+1}|\boldsymbol{o}_l) - \hat{p}_{\boldsymbol{\theta}^k}(\boldsymbol{o}^{easy}_{l+1}|\boldsymbol{o}_l))\\
&< \Pr_{\substack{\boldsymbol{o}'_1\sim P^k(\mathcal{Q}^k)\\ \boldsymbol{o}'\sim\hat{p}_{\boldsymbol{\theta}^k}(\cdot|\boldsymbol{o}_1)}}\left[\boldsymbol{o}'_l = \boldsymbol{o}_l, o'_L = a^k_{o'_1}\right]\\
&\quad \cdot[\hat{p}_{\boldsymbol{\theta}^k}(\boldsymbol{o}^{hard}_{l+1}|\boldsymbol{o}_l) - \hat{p}_{\boldsymbol{\theta}^k}(\boldsymbol{o}^{easy}_{l+1}|\boldsymbol{o}_l)]\\
&\quad \Big[\mathbb{E}[p^k_{acc}(\hat{o}) \mid \substack{\hat{o}_l=o_l\\ \hat{o}_L=a^k_{\hat{o}_1}\\ \hat{o}_{l+1}=o^{easy}_{l+1}}]\\
&\qquad -(\sum_{o'_{l+1}\in\mathcal{S}^{(k)}_{o_l}}\hat{p}_{\boldsymbol{\theta}^k}(\boldsymbol{o}'_{l+1}|\boldsymbol{o}_l)\mathbb{E}[p^k_{acc}(\widetilde{o}) \mid \substack{\widetilde{o}_l=o_l\\ \widetilde{o}_L=a^k_{\widetilde{o}_1}\\ \widetilde{o}_{l+1}=o'_{l+1}}])\Big]
\end{aligned}
\tag{57}
$$

*Then, during every update, it holds that $\Delta\boldsymbol{\theta}^{k,\text{REINFORCE}}_{o_{l+1},o_l} := \eta\nabla_{\boldsymbol{\theta}^k_{o_{l+1},o_l}} \mathcal{J}_{\text{REINFORCE}}(\boldsymbol{\theta}^k)$ with $\eta$ as the step size. Then we have*

$$
\begin{aligned}
\Delta\boldsymbol{\theta}^{k,\text{REINFORCE}}_{o^{hard}_{l+1},o_l} - \Delta\boldsymbol{\theta}^{k,\text{REINFORCE}}_{o^{easy}_{l+1},o_l} &< \eta[\hat{p}_{\boldsymbol{\theta}^k}(\boldsymbol{o}^{hard}_{l+1}|\boldsymbol{o}_l) - \hat{p}_{\boldsymbol{\theta}^k}(\boldsymbol{o}^{easy}_{l+1}|\boldsymbol{o}_l)]\cdot\Pr_{\substack{\boldsymbol{o}'_1\sim P^k(\mathcal{Q}^k)\\ \boldsymbol{o}'\sim\hat{p}_{\boldsymbol{\theta}^k}(\cdot|\boldsymbol{o}_1)}}\left[\boldsymbol{o}'_l = \boldsymbol{o}_l, o'_L = a^k_{o'_1}\right]\\
&\quad \cdot\left[\mathbb{E}[p^k_{acc}(\hat{o}) \mid \substack{\hat{o}_l=o_l\\ \hat{o}_L=a^k_{\hat{o}_1}\\ \hat{o}_{l+1}=o^{easy}_{l+1}}] - (\sum_{o'_{l+1}\in\mathcal{S}^{(k)}_{o_l}}\hat{p}_{\boldsymbol{\theta}^k}(\boldsymbol{o}'_{l+1}|\boldsymbol{o}_l)\mathbb{E}[p^k_{acc}(\widetilde{o}) \mid \substack{\widetilde{o}_l=o_l\\ \widetilde{o}_L=a^k_{\widetilde{o}_1}\\ \widetilde{o}_{l+1}=o'_{l+1}}])\right]
\end{aligned}
$$

*Similarly,*

$$
\begin{aligned}
\Delta\boldsymbol{\theta}^{k,\text{RAFT}}_{o^{hard}_{l+1},o_l} - \Delta\boldsymbol{\theta}^{k,\text{RAFT}}_{o^{easy}_{l+1},o_l} &< \eta[\hat{p}_{\boldsymbol{\theta}^k}(\boldsymbol{o}^{hard}_{l+1}|\boldsymbol{o}_l)(1+\log\hat{p}_{\boldsymbol{\theta}^k}(\boldsymbol{o}^{hard}_{l+1}|\boldsymbol{o}_l)) - \hat{p}_{\boldsymbol{\theta}^k}(\boldsymbol{o}^{easy}_{l+1}|\boldsymbol{o}_l)(1+\log\hat{p}_{\boldsymbol{\theta}^k}(\boldsymbol{o}^{easy}_{l+1}|\boldsymbol{o}_l))]\\
&\quad \cdot\Pr_{\substack{\boldsymbol{o}'_1\sim P^k(\mathcal{Q}^k)\\ \boldsymbol{o}'\sim\hat{p}_{\boldsymbol{\theta}^k}(\cdot|\boldsymbol{o}_1)}}\left[\boldsymbol{o}'_l = \boldsymbol{o}_l, o'_L = a^k_{o'_1}\right]\\
&\quad \cdot\left[\mathbb{E}[p^k_{acc}(\hat{o}) \mid \substack{\hat{o}_l=o_l\\ \hat{o}_L=a^k_{\hat{o}_1}\\ \hat{o}_{l+1}=o^{easy}_{l+1}}] - (\sum_{o'_{l+1}\in\mathcal{S}^{(k)}_{o_l}}\hat{p}_{\boldsymbol{\theta}^k}(\boldsymbol{o}'_{l+1}|\boldsymbol{o}_l)\mathbb{E}[p^k_{acc}(\widetilde{o}) \mid \substack{\widetilde{o}_l=o_l\\ \widetilde{o}_L=a^k_{\widetilde{o}_1}\\ \widetilde{o}_{l+1}=o'_{l+1}}])\right]
\end{aligned}
$$

**Proof of Item 2: Convergence and Failure.** *Per conditions in our item, there are valid easy-to-reason and hard-to-reason CoTs passing $o_l$. Recall that $\boldsymbol{\theta}^{k,(0)} = \boldsymbol{\theta}^\star$ at the iteration $t = 0$ as the base model to be finetuned, $\boldsymbol{\theta}^{k,(t)}$ be the finetuned model for the task $k$ at post-training iteration $t$.*

For any $w \in (c_w/L^2, o(1/L))$ for some small positive constant $c_w > 0$, we consider the finetuning dynamics during:

$$\sum_{o'_{l+1} \in \mathcal{S}_{o_l}^{(k),hard}} \hat{p}_{\boldsymbol{\theta}^{k,(t)}}(\boldsymbol{o}'_{l+1}|\boldsymbol{o}_l) \leq w, \qquad \sum_{o'_{l+1} \in \mathcal{S}_{o_l}^{(k),easy}} \hat{p}_{\boldsymbol{\theta}^{k,(t)}}(\boldsymbol{o}'_{l+1}|\boldsymbol{o}_l) \geq 1 - w. \qquad (58)$$

Then for REINFORCE we have the lower bound over the $\Delta\boldsymbol{\theta}_{o_{l+1}^{easy},o_l}^{k,(t)}$ for any $o_{l+1}^{easy} \in \mathcal{S}_{o_l}^{(k)}$ as

$$\Delta\boldsymbol{\theta}_{o_{l+1}^{easy},o_l}^{k,(t)} \geq \eta \Pr_{\substack{\boldsymbol{o}'_1 \sim P^k(\mathcal{Q}^k) \\ \boldsymbol{o}' \sim \hat{p}_{\boldsymbol{\theta}^k}(\cdot|\boldsymbol{o}_1)}} \left[\boldsymbol{o}'_l = \boldsymbol{o}_l, o'_L = a_{o'_1}^k\right] \hat{p}_{\boldsymbol{\theta}^k}(\boldsymbol{o}_{l+1}^{easy}|\boldsymbol{o}_l)$$

$$\cdot \left[\mathbb{E}[p_{acc}^k(\hat{o}) \mid \substack{\hat{o}_l = o_l \\ \hat{o}_L = a_{\hat{o}_1}^k \\ \hat{o}_{l+1} = o_{l+1}^{easy}}] - \left(\sum_{o'_{l+1} \in \mathcal{S}_{o_l}^{(k)}} \hat{p}_{\boldsymbol{\theta}^k}(\boldsymbol{o}'_{l+1}|\boldsymbol{o}_l)\mathbb{E}[p_{acc}^k(\widetilde{o}) \mid \substack{\widetilde{o}_l = o_l \\ \widetilde{o}_L = a_{\widetilde{o}_1}^k \\ \widetilde{o}_{l+1} = o'_{l+1}}]\right)\right]$$

$$\geq \Theta\left(\frac{\eta}{M^{L-1}} \cdot \frac{1}{M} \cdot [\mathbb{E}[p_{acc}^k(\hat{o}) \mid \substack{\hat{o}_l = o_l \\ \hat{o}_L = a_{\hat{o}_1}^k \\ \hat{o}_{l+1} = o_{l+1}^{easy}}]w - \mathbb{E}[p_{acc}^k(\hat{o}) \mid \substack{\hat{o}_l = o_l \\ \hat{o}_L = a_{\hat{o}_1}^k \\ \hat{o}_{l+1} = o_{l+1}^{hard}}]w]\right)$$

$$\geq \Theta\left(\frac{\eta}{M^L} \cdot \mathbb{E}[p_{acc}^k(\hat{o}) \mid \substack{\hat{o}_l = o_l \\ \hat{o}_L = a_{\hat{o}_1}^k \\ \hat{o}_{l+1} = o_{l+1}^{easy}}](w - \frac{1}{M}w))\right)$$

$$= \Theta\left(\frac{\eta}{M^L} \cdot \frac{(M-1)w}{M}\mathbb{E}[p_{acc}^k(\hat{o}) \mid \substack{\hat{o}_l = o_l \\ \hat{o}_L = a_{\hat{o}_1}^k \\ \hat{o}_{l+1} = o_{l+1}^{easy}}]\right)$$

For a given TMC $\mathbb{P}_{\text{TMC}}(\cdot \mid o_l) = \hat{p}_{\boldsymbol{\theta}^\star}, \forall l \in [L-1]$, the values of

$$\mathbb{E}[p_{acc}^k(\hat{o}) \mid \substack{\hat{o}_l = o_l \\ \hat{o}_L = a_{\hat{o}_1}^k \\ \hat{o}_{l+1} = o_{l+1}^{easy}}], \quad \mathbb{E}[p_{acc}^k(\hat{o}) \mid \substack{\hat{o}_l = o_l \\ \hat{o}_L = a_{\hat{o}_1}^k \\ \hat{o}_{l+1} = o_{l+1}^{hard}}]$$

are indeed deterministic positive constants within $(0,1)$, for any valid CoTs $o^{easy} \in \mathcal{S}_{o_l}^{(k)}$ and $o^{hard} \in \mathcal{S}_{o_l}^{(k)}$ passing $o_l$. That is, we could omit it in $O(1)$:

$$\Delta\boldsymbol{\theta}_{o_{l+1}^{easy},o_l}^{k,(t)} \geq \Theta\left(\frac{\eta}{M^L} \cdot \frac{(M-1)w}{M}\right). \qquad (59)$$

Similarly, recall that

$$D_{o_l} = \{o_{l+1} : \mathbb{P}_{\text{TMC}}(o_{l+1}|o_l) > 0\}, \quad c = \min_{o_{l+1} \in D_{o_l}} \mathbb{P}_{\text{TMC}}(o_{l+1}|o_l) > 0,$$

For neuron $o_{l+1}^{other} \in D_{o_l} \setminus \mathcal{S}_{o_l}^{(k)}$, then by similar derivations we have

$$\Delta\boldsymbol{\theta}_{o_{l+1}^{other},o_l}^{k,(t)} \leq -\Theta\left(\frac{\eta}{M^L} \cdot \left(\frac{(M+1)w}{M}\right)\right) = -\Theta\left(\frac{\eta(M+1)w}{M^{L+1}}\right). \qquad (60)$$

Therefore, by choosing $T \geq \Omega(\eta^{-1}L^2 M^L \log(ML/\epsilon))$ where $0.5 > \epsilon > wL > 0$ is a small constant, we have

$$\sum_{o_{l+1} \in \mathcal{S}_{o_l}^{(k),easy}} \hat{p}_{\boldsymbol{\theta}^{k,(t)}}(\boldsymbol{o}_{l+1}|\boldsymbol{o}_l) = \frac{\sum_{o_{l+1} \in \mathcal{S}_{o_l}^{(k),easy}} e^{\boldsymbol{\theta}_{o_{l+1},o_l}^{k,(T)}}}{\sum_{o_{l+1} \in D_{o_l}} e^{\boldsymbol{\theta}_{o_{l+1},o_l}^{k,(T)}}} = \frac{\sum_{o_{l+1} \in \mathcal{S}_{o_l}^{(k),easy}} e^{\boldsymbol{\theta}_{o_{l+1},o_l}^\star + \sum_{t=0}^{T-1} \Delta\boldsymbol{\theta}_{o_{l+1},o_l}^{k,(t)}}}{\sum_{o_{l+1} \in D_{o_l}} e^{\boldsymbol{\theta}_{o_{l+1},o_l}^\star + \sum_{t=0}^{T-1} \Delta\boldsymbol{\theta}_{o_{l+1},o_l}^{k,(t)}}}$$

$$\geq \frac{\sum_{o_{l+1} \in \mathcal{S}_{o_l}^{(k),easy}} e^{\boldsymbol{\theta}_{o_{l+1},o_l}^\star + T[\frac{\eta}{M^L} \cdot \frac{(M-1)w}{M}]}}{\sum_{o_{l+1} \in \mathcal{S}_{o_l}^{(k),easy}} e^{\boldsymbol{\theta}_{o_{l+1},o_l}^\star + T[\frac{\eta}{M^L} \cdot \frac{(M-1)w}{M}]} + \sum_{o_{l+1} \in C_{o_l} \setminus \mathcal{S}_{o_l}^{(k),easy}} e^{\boldsymbol{\theta}_{o_{l+1},o_l}^\star} + \sum_{o_{l+1} \in D_{o_l} \setminus C_{o_l}} e^{\boldsymbol{\theta}_{o_{l+1},o_l}^\star}}$$

$$\geq \Theta\left(\frac{e^{T[\frac{\eta}{M^L} \cdot \frac{(M-1)w}{M}]}}{e^{T[\frac{\eta}{M^L} \cdot \frac{(M-1)w}{M}]} + (M-1) + \sum_{o_{l+1} \in D_{o_l} \setminus C_{o_l}} M^{-1}}\right)$$

$$\geq \Theta\left(\frac{e^{T[\frac{\eta}{M^L} \cdot \frac{(M-1)w}{M}]}}{e^{T[\frac{\eta}{M^L} \cdot \frac{(M-1)w}{M}]} + M}\right) = \Theta\left(\frac{e^{T[\frac{\eta w}{M^L}]}}{e^{T[\frac{\eta w}{M^L}]} + M}\right)$$

$$\geq 1 - o\left(\frac{\epsilon}{L}\right)$$

$$(61)$$

*Here, the first inequality is by the negative updates in Eq.(47) and Eq.(60) as well as the update lower bound in Eq.(59); the second inequality is by dividing $\sum_{o_{l+1} \in \mathcal{S}_{o_l}^{(k),easy}} e^{\theta_{l+1,o_l}^{\star}}$ term, $|\mathcal{S}_{o_l}^{(k),easy}| \leq$ $n_{o_1} = O(1)$ by Def. 1, $M = \max_{l,o_l \in S_l} |C_{o_l}|$, as well as*

$$\frac{\hat{p}_{\theta^{k,(t)}}(o_{l+1}|o_l)}{\hat{p}_{\theta^{k,(t)}}(o'_{l+1}|o_l)} \geq \frac{\mathbb{P}_{\mathrm{TMC}}(o_{l+1}|o_l)}{\mathbb{P}_{\mathrm{TMC}}(o'_{l+1}|o_l)} \geq \Omega(M),$$

*for $\forall o_{l+1} \in C_{o_l}, o'_{l+1} \in D_{o_l} \setminus C_{o_l}$; the third inequality is by the condition in our item $|D_{o_l} \setminus C_{o_l}| = O(M)$; the last inequality is by choosing $T \geq \Omega(\eta^{-1} L^2 M^L \log(ML/\epsilon))$.*

*Then*

$$\Pr_{\substack{o'_1 \sim P^k(\mathcal{Q}^k) \\ o' \sim \hat{p}_{\theta^{k,(t)}}(\cdot|o_1)}} [o'_L = a_{o'_1}^k] \geq \Pr_{\substack{o'_1 \sim P^k(\mathcal{Q}^k) \\ o' \sim \hat{p}_{\theta^{k,(t)}}(\cdot|o_1)}} [o'_L = a_{o'_1}^k, o' \in \mathcal{G}_{o_1,a_{o_1}^k}^{(k),easy}]$$

$$\geq (1 - o(\frac{\epsilon}{L}))^{L-1} = 1 - o(\epsilon). \tag{62}$$

*Therefore, suggest the probability mass of valid hard-to-reason CoTs traveling from some $o_1 \sim P(\mathcal{Q}_k)$ to $a_{o_1}$ for task $k$ in the original TMC X ($\mathbb{P}_{\mathrm{TMC}}$) is $\Delta$. Then by Thm. 7, we have*

$$\mathrm{Rex}_{o_1,k}^{\hat{p}_{\theta^{k,(t)}}}(o) \leq \Theta((1-\epsilon)\frac{1}{1+\Delta M^{L-1}} + \epsilon \frac{\Delta M^{L-1}}{1+\Delta M^{L-1}}). \tag{63}$$

*Also, by Thm. 7 the pass@K performance (the probability that at least succeed once among K trials) is upper bounded by*

$$\mathrm{Pass@K}_{o_1,k}^{\hat{p}_{\theta^{k,(t)}}} \leq \underbrace{\Theta([(\frac{\Delta M^{L-1}}{1+\Delta M^{L-1}})^{n_q}(1-(1-\epsilon)^K)])}_{\text{upper bound of pass@K of instance that cannot be solved by easy CoTs}} + \underbrace{\Theta([(1-(\frac{\Delta M^{L-1}}{1+\Delta M^{L-1}})^{n_q})(1-\epsilon^K)])}_{\text{upper bound of pass@K of instance that can be solved by some easy CoT}} ).$$

*Also, by Thm. 7 we see that, when $\epsilon = o(\sqrt[K]{1 - C_{\mathrm{Err}}/(\frac{\Delta M^{L-1}}{1+\Delta M^{L-1}})^{n_q}}) \to 0$, the pass@K performance would suffer from constant error.*

*The proof of RAFT is similar–the only difference in Eq.(53) is the ultra $(1+\log \hat{p}_{\theta^k}(o_{l+1}|o_l))$, where the easy-to-reason's value is larger than the hard-to-reason ones due to the monotonicity of $\log(\cdot)$. Also, noted that $1 + \log \hat{p}_{\theta^k}(o_{l+1}|o_l) \in (1+\log c, 1)$, which could be scaling as $O(1)$ such that our results of REINFORCE directly applies.*

### Proof of Item 3: Curriculum Learning of `RL-rej`.

*Per Remark 6, we see that in our case, after $T_1 \geq \Omega(\eta^{-1} L^2 M^L \log(ML/\epsilon))$ for a small $\epsilon$, $\mathtt{RL-rej}$ would learn valid easy-to-reason CoTs in $\mathcal{G}_{o_1,a_{o_1}^k}^{(k),easy}$ well with non-trivial predictive probability $\Theta((1-\epsilon)/M)$, and start to reject the $(\mathbf{Q}, \mathbf{A}) \sim \mathcal{D}_a^{q,k}$ that can be solved by those CoTs. That is, there exists $T_1 = \Omega(\eta^{-1} L^2 M^L \log(ML/\epsilon))$, for $t \in (0, T_1]$, the $\mathtt{RL-rej}$ behaves exactly the same with REINFORCE. After $t \geq T_1$ we have*

$$\nabla_{\theta_{o_{l+1}^{easy},o_l}^k} \mathcal{J}_{\mathrm{Rein-rej}}(\theta^k) = \Pr_{\substack{o'_1 \sim P^k(\mathcal{Q}^k) \\ o' \sim \hat{p}_{\theta^k}(\cdot|o_1)}} [o'_l = o_l, o'_L = a_{o'_1}^k] \hat{p}_{\theta^k}(o_{l+1}^{easy}|o_l) \left[ 0 - (\sum_{o'_{l+1} \in \sum_{o'_{l+1} \in \mathcal{S}_{o_l}^{(k),hard}}} \hat{p}_{\theta^k}(o'_{l+1}|o_l) \mathbb{E}[p_{acc}^k(\tilde{o}) \mid \substack{\tilde{o}_l = o_l \\ \tilde{o}_L = a_{\tilde{o}_1}^k \\ \tilde{o}_{l+1} = o'_{l+1}}]) \right] < 0, \tag{64}$$

*for all $o_{l+1}^{easy} \in \mathcal{S}_{o_l}^{(k),easy}, \forall l \in [L-1]$, where the correctly predicted easy-to-reason CoTs are rejected according to the condition in our item. Similarly, for $o_{l+1}^{hard} \in \mathcal{S}_{o_l}^{(k),hard} = \mathcal{S}_{o_l}^{(k)} \setminus \mathcal{S}_{o_l}^{(k),easy}, \forall l \in [L-1]$ we have*

$$\nabla_{\theta_{o_{l+1}^{hard},o_l}^k} \mathcal{J}_{\mathrm{Rein-rej}}(\theta^k) = \Pr_{\substack{o'_1 \sim P^k(\mathcal{Q}^k) \\ o' \sim \hat{p}_{\theta^k}(\cdot|o_1)}} [o'_l = o_l, o'_L = a_{o'_1}^k] \hat{p}_{\theta^k}(o_{l+1}^{hard}|o_l) \left[ \mathbb{E}[p_{acc}^k(\hat{o}) \mid \substack{\hat{o}_l = o_l \\ \hat{o}_L = a_{\hat{o}_1}^k \\ \hat{o}_{l+1} = o_{l+1}^{hard}}] - (\sum_{o'_{l+1} \in \sum_{o'_{l+1} \in \mathcal{S}_{o_l}^{(k),hard}}} \hat{p}_{\theta^k}(o'_{l+1}|o_l) \mathbb{E}[p_{acc}^k(\tilde{o}) \mid \substack{\tilde{o}_l = o_l \\ \tilde{o}_L = a_{\tilde{o}_1}^k \\ \tilde{o}_{l+1} = o'_{l+1}}]) \right] > 0, \tag{65}$$

*where the inequality is by the feeble $\hat{p}_{\theta^k}(o'_{l+1}|o_l) = o(1/M), o'_{l+1} \in \sum_{o'_{l+1} \in \mathcal{S}_{o_l}^{(k),hard}}$.*

*Therefore, we have*

$$\Delta\theta_{o_{l+1}^{easy},o_l}^{k,\mathrm{Rein-rej}} := \eta \nabla_{\theta_{o_{l+1}^{easy},o_l}^k} \mathcal{J}_{\mathrm{Rein-rej}}(\theta^k) < 0, \quad \Delta\theta_{o_{l+1}^{hard},o_l}^{k,\mathrm{Rein-rej}} := \eta \nabla_{\theta_{o_{l+1}^{hard},o_l}^k} \mathcal{J}_{\mathrm{Rein-rej}}(\theta^k) > 0,$$

and thus the $\hat{p}_{\boldsymbol{\theta}^k,(t)}(\boldsymbol{o}_{l+1}^{hard}|\boldsymbol{o}_l)/\hat{p}_{\boldsymbol{\theta}^k,(t)}(\boldsymbol{o}_{l+1}^{easy}|\boldsymbol{o}_l)$ would strictly increase.

By Eq.(61), similarly there exists $T_2 = T_1 + \Theta(\eta^{-1}L^2M^L\log(ML/\epsilon))$, the predictive probability $\hat{p}_{\boldsymbol{\theta}^k,(T_2)}$ of some valid hard-to-reason CoTs in $\mathcal{G}_{o_1,a_{o_1}}^{(k),hard}$ would reach $\Theta((1-\epsilon)/M)$. Simultaneously, we see that by adjusting the learning rare to be appropriately small, the predictive probability of easy-to-reason CoTs in $\mathcal{G}_{o_1,a_{o_1}}^{(k),easy}$ would not decay below the scale $\Theta((1-\epsilon)/M)$, other wise it would be re-collected in the $\mathcal{D}_{\text{rej},(t)}^{o_1,k}$ at that iteration $t$ according to our condition setting in item 3. That is, the model $\hat{p}_{\boldsymbol{\theta}^k,(t)}$ would gradually increase the predictive probability of sparse edges in $\mathcal{S}_{o_l}^{(k),hard} = \mathcal{S}_{o_l}^{(k)} \setminus \mathcal{S}_{o_l}^{(k),easy}$ for $\forall l \in [L-1]$, and thus the CoTs in $\mathcal{G}_{o_1,a_{o_1}}^{(k),hard}$ sharing the most common edges with some CoTs in $\mathcal{G}_{o_1,a_{o_1}}^{(k),easy}$, would first be learned. Afterwards, more and more hard-to-reason CoTs in $\mathcal{G}_{o_1,a_{o_1}}^{(k),hard}$ is getting learned, until the point where further learning a new sparse edge in some $\mathcal{S}_{o_l}^{(k),hard}, l \in [L-1]$ will make another already learned CoT's predictive probability to be lower than the scale $o(1-\epsilon)$.

Suppose the probability mass of valid hard-to-reason CoTs traveling from $o_1$ to $a_{o_1}$ for task $k$ in the original TMC $X$ ($\mathbb{P}_{\text{TMC}}$) is $\Delta$. Suggest after $T_2 = \Omega(\eta^{-1}L^2M^L\log(ML/\epsilon))$, there are $\text{n}'_{o_1}$ hard-to-reason CoTs each with likelihood ratio scale $\Theta(\rho) < 1$ in the $\mathcal{G}_{o_1,a_{o_1}}^{(k)}$ have been well-learned with predictive probability $\Theta((1-\epsilon)/M)$, then, similar to Thm. 7, we have

$$\text{Pass@K}_{o_1,k}^{\hat{p}_{\boldsymbol{\theta}^k,(t)}} = 4\underbrace{\Theta([(1-\rho)^{\text{n}'_{o_1}}(\frac{\Delta M^{L-1}}{1+\Delta M^{L-1}})^{\text{n}_{o_1}}(1-(1-\epsilon)^K)])}_{\text{upper bound of pass@K of instance that cannot be solved by easy CoTs}} + \underbrace{\Theta([(1-(1-\rho)^{\text{n}'_{o_1}}(\frac{\Delta M^{L-1}}{1+\Delta M^{L-1}})^{\text{n}_{o_1}})(1-\epsilon^K)])}_{\text{upper bound of pass@K of instance that can be solved by some easy CoT}})$$

which would tends to 1 when $(1-\rho)^{\text{n}'_{o_1}} \to 0, \epsilon \to 0$.

`RL-rej` with algorithms other than REINFORCE directly follows.

$\square$

**Theorem 11** (Advantage-based Finetuning Favors Easy-to-Reason CoTs (Formal Version of Thm. 2))**.** Let $\boldsymbol{\theta}^\star$ be the base model in Eq.(2 that exact predicts the distribution of a Multi-task TMC as in Def. 1 and 2, and $\boldsymbol{\theta}^k$ the current model to be finetuned from $\boldsymbol{\theta}^\star$ for task $k \in \mathcal{T}$. Denote the task tuples of task $k \in \mathcal{T}$ as $(q, a_q^k, k)$, where $a_q^k \in S_L$ is the sole answer state under task $k$. Assume for each $(o_1, a_{o_1}, k)$ under task $k$, the number of hard-to-reason CoTs from $o_1$ to $a_{o_1}$ is bounded by $O(M)$. Let the question distribution during finetuning of task $k$ be $P^k(\mathcal{Q}^k)$ (i.e., $o_1 \sim P^k(\mathcal{Q}^k)$). Suppose the estimates of the RL advantage of PPO / GRPO (without the KL term) by some outer oracle or group-level normalization are accurate during the finetuning: $A_{l+1}^{\hat{p}_{\boldsymbol{\theta}^k},k}$, $\hat{A}_{i,l+1}^k = A_{l+1}^{\hat{p}_{\boldsymbol{\theta}^k},k}(\boldsymbol{o}_l, \boldsymbol{o}_{l+1})$ for any CoT $o$, and the $\hat{p}_{old}^k$ is appropriately chosen that clip operation **is always functioning** starting from the finetuning with $\epsilon_{\text{clip}} = o(1)$ such that

$$(1 + \epsilon_{\text{clip}})A_{l+1}^{\hat{p}_{\boldsymbol{\theta}^k},k}(\boldsymbol{o}_l, \boldsymbol{o}_{l+1}) \leq \frac{\hat{p}_{\boldsymbol{\theta}^k}(\boldsymbol{o}_{l+1}|\boldsymbol{o}_l)}{\hat{p}_{old}^k(\boldsymbol{o}_{l+1}|\boldsymbol{o}_l)} A_{l+1}^{\hat{p}_{\boldsymbol{\theta}^k},k}(\boldsymbol{o}_l, \boldsymbol{o}_{l+1}), \text{ if } A_{l+1}^{\hat{p}_{\boldsymbol{\theta}^k},k}(\boldsymbol{o}_l, \boldsymbol{o}_{l+1}) \geq 0,$$
$$(1 - \epsilon_{\text{clip}})A_{l+1}^{\hat{p}_{\boldsymbol{\theta}^k},k}(\boldsymbol{o}_l, \boldsymbol{o}_{l+1}) \leq \frac{\hat{p}_{\boldsymbol{\theta}^k}(\boldsymbol{o}_{l+1}|\boldsymbol{o}_l)}{\hat{p}_{old}^k(\boldsymbol{o}_{l+1}|\boldsymbol{o}_l)} A_{l+1}^{\hat{p}_{\boldsymbol{\theta}^k},k}(\boldsymbol{o}_l, \boldsymbol{o}_{l+1}), \text{ if } A_{l+1}^{\hat{p}_{\boldsymbol{\theta}^k},k}(\boldsymbol{o}_l, \boldsymbol{o}_{l+1}) \leq 0,$$
$$(66)$$

Then the shared form of objective as

$$\mathcal{J}_{\text{PO}} = \mathbb{E}_{\boldsymbol{o}_1 \sim P^k(\mathcal{Q}^k),(\mathbf{Q},\mathbf{A})\sim\mathcal{D}_{a_{o_1}}^{o_1,k},\boldsymbol{o}_{2:L}\sim\hat{p}_{\boldsymbol{\theta}}^k(O|\boldsymbol{o}_1)}\Big[\frac{1}{L}\sum_{l=1}^{L-1}(1 + (2\mathbb{1}(A_{l+1}^{\hat{p}_{\boldsymbol{\theta}^k},k}(\boldsymbol{o}_l, \boldsymbol{o}_{l+1}) \geq 0) - 1)\epsilon_{\text{clip}}A_{l+1}^{\hat{p}_{\boldsymbol{\theta}^k},k}(\boldsymbol{o}_l, \boldsymbol{o}_{l+1})\}\Big],$$
$$(67)$$

where $\epsilon_{\text{clip}} > 0$ is a offset clipping parameter, and by Eq.(20),

$$A_{l+1}^{\hat{p}_{\boldsymbol{\theta}},k}(\boldsymbol{o}_l, \boldsymbol{o}_{l+1}) = \mathbb{E}_{o_1=q\sim P^k(\mathcal{Q}_k),\boldsymbol{o}_{l+2:L}\sim\hat{p}_{\boldsymbol{\theta}}}\left[R_{\text{out}}^k(\boldsymbol{o})\big|\boldsymbol{o}_{l+1}\right] - \mathbb{E}_{o_1=q\sim P^k(\mathcal{Q}_k),\boldsymbol{o}_{l+1:L}\sim\hat{p}_{\boldsymbol{\theta}}}\left[R_{\text{out}}^k(\boldsymbol{o})\big|\boldsymbol{o}_l\right].$$
$$(68)$$

***Favor Easy CoTs***. For any different state pair $o_{l+1}^{hard} \neq o_{l+1}^{easy} \in \mathcal{S}_{o_l}^{(k)}$ denoting two $l+1$-th states in some valid hard-to-reason and easy-to-reason CoT sharing the $l$-th state $o_l$ for task $k$, it holds that

$$\Delta\boldsymbol{\theta}_{o_{l+1}^{easy},o_l}^{k,\text{PO}} := \eta\nabla_{\boldsymbol{\theta}_{o_{l+1}^{easy},o_l}^k}\mathcal{J}_{\text{PO}}(\boldsymbol{\theta}^k) > 0, \quad \Delta\boldsymbol{\theta}_{o_{l+1}^{hard},o_l}^{k,\text{PO}} := \eta\nabla_{\boldsymbol{\theta}_{o_{l+1}^{hard},o_l}^k}\mathcal{J}_{\text{PO}}(\boldsymbol{\theta}^k) < 0,$$
$$\Delta\boldsymbol{\theta}_{o'_{l+1},o_l}^{k,\text{PO}} := \eta\nabla_{\boldsymbol{\theta}_{o'_{l+1},o_l}^k}\mathcal{J}_{\text{PO}}(\boldsymbol{\theta}^k) < 0,$$
$$(69)$$

*for $\forall o'_{l+1} \in D_{o_l} \setminus \mathcal{S}^{(k)}_{o_l}$. There exists $T \geq \Omega(\eta^{-1} L^2 M^L \log(ML/\epsilon))$, for $t \geq T$, the probability that $\hat{p}_{\boldsymbol{\theta}^k,(t)}(\cdot|o_1)$ reach the $a_{o_1}$ is converged:*

$$\Pr_{\substack{\boldsymbol{o}'_1 \sim P^k(\mathcal{Q}^k) \\ \boldsymbol{o}' \sim \hat{p}_{\boldsymbol{\theta}^k,(t)}(\cdot|\boldsymbol{o}_1)}} [o'_L = a^k_{o'_1}] \geq \Pr_{\substack{\boldsymbol{o}'_1 \sim P^k(\mathcal{Q}^k) \\ \boldsymbol{o}' \sim \hat{p}_{\boldsymbol{\theta}^k,(t)}(\cdot|\boldsymbol{o}_1)}} [o'_L = a^k_{o'_1}, o' \in \mathcal{G}^{(k),easy}_{o_1,a^k_{o_1}}] \geq 1 - o(\epsilon). \qquad (70)$$

*Further, the **pass@K performance** $\mathrm{Pass@K}^{\hat{p}}_{q,k} := \Pr_{\substack{\{\boldsymbol{o}^i\}_{i \in [K]} \sim \hat{p}(O|q) \\ (\mathbf{Q},\mathbf{A}) \sim \mathcal{D}^{q,k}_{a_q}}} [\bigcup_{i=1}^K \mathbb{1}(\boldsymbol{o}^i \in \mathcal{G}^{(k)}_{\mathbf{Q},\mathbf{A}})]$ is upper bounded by*

$$\mathrm{Pass@K}^{\hat{p}_{\boldsymbol{\theta}^k,(t)}}_{o_1,k} \leq \underbrace{\Theta([(\frac{\Delta M^{L-1}}{1+\Delta M^{L-1}})^{\mathrm{n}_q}(1-(1-\epsilon)^K)])}_{\text{Solved by hard CoTs}} + \underbrace{\Theta([(1-(\frac{\Delta M^{L-1}}{1+\Delta M^{L-1}})^{\mathrm{n}_q})(1-\epsilon^K)])}_{\text{Solved by some easy CoTs}}). \qquad (71)$$

*When $\epsilon = o(\sqrt[K]{1 - C_{\mathrm{Err}}/(\frac{\Delta M^{L-1}}{1+\Delta M^{L-1}})^{\mathrm{n}_q}}) \to 0$, the pass@K performance suffer from constant error: $1 - \mathrm{Pass@K}^{\hat{p}_{\boldsymbol{\theta}^k,(t)}}_{o_1,k} = \Theta(1)$.*

**Proof.** *By Prop. 1, for each $l$ we have*

$$A^{\hat{p}_{\boldsymbol{\theta}^\star},k}_{l+1}(\boldsymbol{o}_l, \boldsymbol{o}^{\mathrm{easy}}_{l+1}) = a^l_{\mathrm{easy}} \geq \Theta(M^{-(L+1-l)}) > 0, \quad A^{\hat{p}_{\boldsymbol{\theta}^\star},k}_{l+1}(\boldsymbol{o}_l, \boldsymbol{o}^{\mathrm{hard}}_{l+1}) = -a^l_{\mathrm{hard}} \leq -\Theta(M^{-(L+1-l)}) < 0, \qquad (72)$$

*for constants $a_{\mathrm{easy}}, a_{\mathrm{hard}} > 0$.*

*Also, for $\forall o'_{l+1} \in D_{o_l} \setminus \mathcal{S}^{(k)}_{o_l}$, by definition it directly holds that*

$$A^{\hat{p}_{\boldsymbol{\theta}^\star},k}_{l+1}(\boldsymbol{o}_l, \boldsymbol{o}'_{l+1}) < A^{\hat{p}^\star_{\boldsymbol{\theta}},k}_{l+1}(\boldsymbol{o}_l, \boldsymbol{o}^{\mathrm{hard}}_{l+1}) = -a^l_{\mathrm{hard}} < -\Theta(M^{-(L+1-l)}) < 0. \qquad (73)$$

*Therefore, by Lemma 9 as well as the property of TMC, it holds that*

$$\nabla_{\boldsymbol{\theta}^k} \mathcal{J}_{\mathrm{PO}}(\boldsymbol{\theta}^k) = \sum_{l=1}^{L-1} \mathbb{E}_{\substack{\boldsymbol{o}_1 \sim P^k(\mathcal{Q}^k) \\ \{\boldsymbol{o}_{l+1} \sim \hat{p}_{\boldsymbol{\theta}^k}(\cdot|\boldsymbol{o}_l)\}^{L-1}_{l=1}}} \left[ (1 + (2\mathbb{1}(A^{\hat{p}_{\boldsymbol{\theta}^k},k}_{l+1}(\boldsymbol{o}_l, \boldsymbol{o}_{l+1}) \geq 0) - 1)\epsilon_{\mathrm{clip}}) A^{\hat{p}_{\boldsymbol{\theta}^k},k}_{l+1}(\boldsymbol{o}_l, \boldsymbol{o}_{l+1}) \cdot (e_{o_{l+1},o_l} - \sum_{\boldsymbol{o}'_{l+1} \in D_{o_l}} \hat{p}_{\boldsymbol{\theta}^k}(\boldsymbol{o}'_{l+1}|\boldsymbol{o}_l) e_{o'_{l+1},o_l}) \right].$$

*Therefore, collaborating with Eq.(72) and Eq.(73), for $t = 0$ where $\hat{p}_{\boldsymbol{\theta}^k,(0)} = \hat{p}_{\boldsymbol{\theta}^\star}$, we directly have*

$$\Delta\boldsymbol{\theta}^{k,\mathrm{PO}}_{o^{easy}_{l+1},o_l} := \eta\nabla_{\boldsymbol{\theta}^k_{o^{easy}_{l+1},o_l}} \mathcal{J}_{\mathrm{PO}}(\boldsymbol{\theta}^k) > 0, \quad \Delta\boldsymbol{\theta}^{k,\mathrm{PO}}_{o^{hard}_{l+1},o_l} := \eta\nabla_{\boldsymbol{\theta}^k_{o^{hard}_{l+1},o_l}} \mathcal{J}_{\mathrm{PO}}(\boldsymbol{\theta}^k) < 0,$$
$$\Delta\boldsymbol{\theta}^{k,\mathrm{PO}}_{o'_{l+1},o_l} := \eta\nabla_{\boldsymbol{\theta}^k_{o'_{l+1},o_l}} \mathcal{J}_{\mathrm{PO}}(\boldsymbol{\theta}^k) < 0, \qquad (74)$$

*for $\forall o'_{l+1} \in D_{o_l} \setminus \mathcal{S}^{(k)}_{o_l}$. Indeed, following the proof strategies in Lemma 9, we directly see that when the transitions of $o_l \to o^{\mathrm{easy}}_{l+1}$ is further strengthened and the transitions of $o_l \to o^{\mathrm{hard}}_{l+1}$ is further weaken, the $A^{\hat{p}_{\boldsymbol{\theta}^k,(t)},k}_{l+1}(\boldsymbol{o}_l, \boldsymbol{o}^{\mathrm{easy}}_{l+1})$ is strictly increasing along the iterations, and $A^{\hat{p}_{\boldsymbol{\theta}^k,(t)},k}_{l+1}(\boldsymbol{o}_l, \boldsymbol{o}^{\mathrm{hard}}_{l+1}), A^{\hat{p}_{\boldsymbol{\theta}^k,(t)},k}_{l+1}(\boldsymbol{o}_l, \boldsymbol{o}'_{l+1}), \forall o'_{l+1} \in D_{o_l} \setminus \mathcal{S}^{(k)}_{o_l}$ is strictly decreasing. This makes Eq.(72), Eq.(73) and Eq.(69) hold during the finetuning iterations.*

*Specifically, for any different state pair $o^{hard}_{l+1} \neq o^{easy}_{l+1} \in \mathcal{S}^{(k)}_{o_l}$ and $\forall o'_{l+1} \in D_{o_l} \setminus \mathcal{S}^{(k)}_{o_l}$, it holds that*

$$\Delta\boldsymbol{\theta}^{k,(t)}_{o^{easy}_{l+1},o_l} \geq \Theta(\eta M^{-(L+1-l)}(1 - \sum_{\boldsymbol{o}'_{l+1}\in S^{(k),easy}_{o_l}} \hat{p}_{\boldsymbol{\theta}^k}(\boldsymbol{o}'_{l+1}|\boldsymbol{o}_l)$$
$$+ \sum_{\boldsymbol{o}'_{l+1}\in D_{o_l}\backslash S^{(k),easy}_{o_l}} \hat{p}_{\boldsymbol{\theta}^k}(\boldsymbol{o}'_{l+1}|\boldsymbol{o}_l))) > 0,$$

$$\Delta\boldsymbol{\theta}^{k,(t)}_{o^{hard}_{l+1},o_l} \leq -\Theta(\eta M^{-(L+1-l)}(1 + \sum_{\boldsymbol{o}'_{l+1}\in S^{(k),easy}_{o_l}} \hat{p}_{\boldsymbol{\theta}^k}(\boldsymbol{o}'_{l+1}|\boldsymbol{o}_l)$$
$$- \sum_{\boldsymbol{o}'_{l+1}\in D_{o_l}\backslash S^{(k),easy}_{o_l}} \hat{p}_{\boldsymbol{\theta}^k}(\boldsymbol{o}'_{l+1}|\boldsymbol{o}_l))) < 0,$$

$$\Delta\boldsymbol{\theta}^{k,(t)}_{o'_{l+1},o_l} \leq -\Theta(\eta M^{-(L+1-l)}(1 + \sum_{\boldsymbol{o}'_{l+1}\in S^{(k),easy}_{o_l}} \hat{p}_{\boldsymbol{\theta}^k}(\boldsymbol{o}'_{l+1}|\boldsymbol{o}_l)$$
$$- \sum_{\boldsymbol{o}'_{l+1}\in D_{o_l}\backslash S^{(k),easy}_{o_l}} \hat{p}_{\boldsymbol{\theta}^k}(\boldsymbol{o}'_{l+1}|\boldsymbol{o}_l))) < 0,$$

*where the inequalities is by Eq.(72), Eq.(73), as well as $\epsilon_{\text{clip}} = o(1)$.*

*Similar to the techniques in Thm. 10, given that $M^{-(L+1-l)} > M^{-L+1}$ and $p^k_{acc} \leq 1$, after $T \geq \Omega(\eta^{-1}L^2M^L\log(ML/\epsilon))$ iterations, the remaining proofs and results follows as in Thm. 10.*

$\square$

**Remark 12.** *To simplify the discussion of the policy gradient case and avoid the non-convexity of $\min\{\cdot\}$, we assume the clip operation with $\epsilon_{\text{clip}} = o(1)$ and Eq. (66). However, our results still hold without this assumption.*

*Specifically, when the $\min$ does not select the clipped term, we instead encounter:*

$$\nabla_{\boldsymbol{\theta}^k}\left[\frac{\hat{p}_{\boldsymbol{\theta}^k}(\boldsymbol{o}_{l+1}|\boldsymbol{o}_l)}{\hat{p}_{old}(\boldsymbol{o}_{l+1}|\boldsymbol{o}_l)}\hat{p}_{\boldsymbol{\theta}^k}(\boldsymbol{o}_{l+1}|\boldsymbol{o}_l)\right] = 2\frac{\hat{p}_{\boldsymbol{\theta}^k}(\boldsymbol{o}_{l+1}|\boldsymbol{o}_l)}{\hat{p}_{old}(\boldsymbol{o}_{l+1}|\boldsymbol{o}_l)}\nabla_{\boldsymbol{\theta}^k}\hat{p}_{\boldsymbol{\theta}^k}(\boldsymbol{o}_{l+1}|\boldsymbol{o}_l)$$
$$= 2\frac{\hat{p}_{\boldsymbol{\theta}^k}(\boldsymbol{o}_{l+1}|\boldsymbol{o}_l)^2}{\hat{p}_{old}(\boldsymbol{o}_{l+1}|\boldsymbol{o}_l)}\nabla_{\boldsymbol{\theta}^k}\log\hat{p}_{\boldsymbol{\theta}^k}(\boldsymbol{o}_{l+1}|\boldsymbol{o}_l) \qquad (75)$$
$$= \mathbb{E}\left[2\frac{\hat{p}_{\boldsymbol{\theta}^k}(\boldsymbol{o}_{l+1}|\boldsymbol{o}_l)}{\hat{p}_{old}(\boldsymbol{o}_{l+1}|\boldsymbol{o}_l)}\nabla_{\boldsymbol{\theta}^k}\log\hat{p}_{\boldsymbol{\theta}^k}(\boldsymbol{o}_{l+1}|\boldsymbol{o}_l)\right],$$

*instead of*

$$\nabla_{\boldsymbol{\theta}^k}\left[(1\pm\epsilon_{\text{clip}})\hat{p}_{\boldsymbol{\theta}^k}(\boldsymbol{o}_{l+1}|\boldsymbol{o}_l)\right] = (1\pm\epsilon_{\text{clip}})\hat{p}_{\boldsymbol{\theta}^k}(\boldsymbol{o}_{l+1}|\boldsymbol{o}_l)\nabla_{\boldsymbol{\theta}^k}\log\hat{p}_{\boldsymbol{\theta}^k}(\boldsymbol{o}_{l+1}|\boldsymbol{o}_l)$$
$$= (1\pm\epsilon_{\text{clip}})\mathbb{E}\left[\log\hat{p}_{\boldsymbol{\theta}^k}(\boldsymbol{o}_{l+1}|\boldsymbol{o}_l)\right].$$

*Since clearly $\hat{p}_{\boldsymbol{\theta}^k}(\boldsymbol{o}^{easy}_{l+1}|\boldsymbol{o}_l) > \hat{p}_{\boldsymbol{\theta}^k}(\boldsymbol{o}^{hard}_{l+1}|\boldsymbol{o}_l)$, Eq. (75) shows that the gradient magnitude for easy edges dominates that of sparse ones. Thus, the squeezing effect persists even without the assumption. We adopt the assumption in our theorem purely to reduce discussion complexity.*

**Lemma 4.** *[Detailed Version of Lemma 1] Let $\boldsymbol{\theta}^\star$ be the base model in Eq.(2 that exact predicts the distribution of a Multi-task TMC as in Def. 1 and 2, and $\boldsymbol{\theta}^k$ the current model to be finetuned from $\boldsymbol{\theta}^\star$ for task $k \in \mathcal{T}$. Suppose the estimates of RL advantage by GRPO through group-level normalization is accurate as $A^{\hat{p}_{\boldsymbol{\theta}^k},k}_{l+1}(\boldsymbol{o}_l,\boldsymbol{o}_{l+1})$ for any CoT o. The optimal step-wise sampling distribution of the KL-regularized GRPO objective in Eq.(6) is:*

$$\hat{p}^{\text{PO}}_{\boldsymbol{\theta}^k}(\boldsymbol{o}_{l+1}|\boldsymbol{o}_l) \propto \hat{p}_{\boldsymbol{\theta}^\star}(\boldsymbol{o}_{l+1}|\boldsymbol{o}_l)\exp\left(\hat{r}\frac{A^{\hat{p}_{\boldsymbol{\theta}^\star},k}_{l+1}(\boldsymbol{o}_l,\boldsymbol{o}_{l+1})}{\beta}\right), \qquad (76)$$

*where $\hat{r} \leq \max\{1+\epsilon_{\text{clip}}, c^{-1}, \Theta(M)\}$.*

**Proof.** *This result is standard in RL and distribution optimization literature Ziebart (2008); Levine (2018); Foster et al. (2025); Kawata et al. (2025); Fan et al. (2023); Black et al. (2024); Clark et al. (2024); Uehara et al. (2024). The proofs mirror the proof of Corollary 3 in Sec. K, and we therefore omit their full proofs for brevity.* $\square$

**Corollary 7** (Full Version of Corollary 2)**.** *Let $\boldsymbol{\theta}^\star$ be the base model in Eq.(2 that exactly predicts the distribution of a Multi-task TMC as in Defs. 1 and 2. For any target task $k \in \mathcal{T}$, consider the following two categories of instances:*

1. *Istances $(\mathcal{Q}, \mathbf{A})$ whose correct CoTs only lie in $\mathcal{G}_{q,a_q^k}^{(k),hard}$.*

2. *Instances $(\mathcal{Q}, \mathbf{A})$ sampled from another task $k' \neq k$.*

*For PPO/GRPO without KL regularization that satisfy the conditions in Thm. 11, the pass@K upper bound for these instances after $T \geq \Omega(\eta^{-1} L^2 M^L \log(ML/\epsilon))$ is $\left(1 - (1 - \epsilon)^K\right)$.*

*In contrast, for the optimal sampler $\hat{p}_{\boldsymbol{\theta}^k}^{\mathrm{PO}}$ in Eq. (76), for any $\epsilon'$ satisfying $1/N_{o_1} > \epsilon' \geq \epsilon > 0$, denote $\hat{p}_{\boldsymbol{\theta}^{k,(t)}}^k$ as the PPO/GRPO in Thm. 2 with $\epsilon$, if*

$$\beta > \frac{2\hat{r}(L-1)}{\ln\left(\frac{1}{\epsilon' \prod_{l=1}^{L-1} |D_{o_l}|}\right)},$$

*then the pass@K performance of $\hat{p}_{\boldsymbol{\theta}^k}^{\mathrm{PO}}$ is strictly better than that of PPO/GRPO without KL regularization under the same conditions:*

1. ***Capable of Hard CoTs***: *For instance $(\mathcal{Q}, \mathbf{A})$ with only some hard-to-reason CoTs correct:*

$$\mathbb{E}_{\boldsymbol{o}_{2:L} \sim \hat{p}_{\boldsymbol{\theta}^k}^{\mathrm{PO}}(\cdot|\boldsymbol{o}_1)}\left[R_{(\mathbf{Q},\mathbf{A})}^k(\boldsymbol{o})\right] \geq \epsilon' \geq \epsilon \geq \mathbb{E}_{\boldsymbol{o}_{2:L} \sim \hat{p}_{\boldsymbol{\theta}^{k,(t)}}^k(\cdot|\boldsymbol{o}_1)}\left[R_{(\mathbf{Q},\mathbf{A})}^k(\boldsymbol{o})\right].$$

2. ***Preserve Multi-task***: *For instance $(\mathcal{Q}, \mathbf{A})$ belonging to untargeted task $k' \neq k$:*

$$\mathbb{E}_{\boldsymbol{o}_{2:L} \sim \hat{p}_{\boldsymbol{\theta}^k}^{\mathrm{PO}}(\cdot|\boldsymbol{o}_1)}\left[R_{(\mathbf{Q},\mathbf{A})}^{k'}(\boldsymbol{o})\right] \geq \epsilon' \geq \epsilon \geq \mathbb{E}_{\boldsymbol{o}_{2:L} \sim \hat{p}_{\boldsymbol{\theta}^{k,(t)}}^k(\cdot|\boldsymbol{o}_1)}\left[R_{(\mathbf{Q},\mathbf{A})}^{k'}(\boldsymbol{o})\right].$$

**Proof.**

*It suffices to prove that with a large $\beta$, any non-zero transition within the TMC is larger than $\epsilon' < N_{o_1}^{-1} := |D_{o_l}|$.*

*By the definition of the advantage function in Eq.(20, we have*

$$-1 \leq A_{l+1}^{\hat{p}_{\boldsymbol{\theta}^\star}, k}(\boldsymbol{o}_l, \boldsymbol{o}_{l+1}) \leq 1.$$

*Therefore, from Eq.(76, the minimum sampling probability over any edge in $D_{o_l}$ is*

$$\hat{p}_{\boldsymbol{\theta}^k}^{\mathrm{PO}}(\boldsymbol{o}_{l+1}|\boldsymbol{o}_l) \geq \frac{e^{-\frac{\hat{r}}{\beta}}}{|D_{o_l}|e^{\frac{\hat{r}}{\beta}}} = \frac{e^{-\frac{2\hat{r}}{\beta}}}{|D_{o_l}|}.$$

*Hence, for any trajectory of length $L$, the probability of sampling a specific terminal state $o_L$ from any starting state $o_1$ whose $o_{l+1}$ transitions are in $D_{o_l}$ is lower bounded by*

$$\prod_{l=1}^{L-1} \frac{e^{-\frac{2\hat{r}}{\beta}}}{|D_{o_l}|} = e^{-\frac{2\hat{r}(L-1)}{\beta}} \cdot \prod_{l=1}^{L-1} \frac{1}{|D_{o_l}|}.$$

*Define $C := \prod_{l=1}^{L-1} \frac{1}{|D_{o_l}|}$. We seek the condition on $\beta$ such that this probability is at least $\epsilon'$, i.e.,*

$$C \cdot e^{-\frac{2\hat{r}(L-1)}{\beta}} \geq \epsilon.$$

*Dividing both sides by $C$ and taking logarithms yields*

$$-\frac{2\hat{r}(L-1)}{\beta} \geq \ln\left(\frac{\epsilon'}{C}\right), \quad so \quad \beta \geq \frac{2\hat{r}(L-1)}{\ln\left(\frac{1}{\epsilon'C}\right)}.$$

*Substituting $C = \prod_{l=1}^{L-1} \frac{1}{|D_{o_l}|}$, we obtain the desired bound:*

$$\beta > \frac{2\hat{r}(L-1)}{\ln\left(\frac{1}{\epsilon' \prod_{l=1}^{L-1} |D_{o_l}|}\right)}.$$

*That is, the probability of the path $(o_1, o_2, \cdots, o_L), o_{l+1} \in D_{o_l}, \forall l \in [L-1]$ is larger than $\epsilon'$. This ensure that the model is more capable of sampling valid hard-to-reason CoTs for current task as well as valid CoTs for other tasks, as long as the path with transition probability larger than zero $(c > 0)$ in Def. 1.* □

## K  DETAILS AND PROOFS OF REWARD-BASED SAMPLING

**Lemma 5** (BoN/BS with Ground-true Signal Oracle). *Let $\theta^\star$ be the base model in Eq.(2 that exactly predicts the distribution of a Multi-task TMC as defined in Definitions 1 and 2. Under task tuple $(q, a, k)$, consider the ORMs $R_{\mathbf{Q},\mathbf{A}}^k(\cdot)$, as well as the PRM given in Eqs. 11. For any target task $k \in \mathcal{T}$ and instance distribution $\mathcal{D}_{a_q}^{a,k}$, if the total number of valid hard-to-reason CoTs is $\Theta(M)$, then during pass@K sampling:*

- *ORM/PRM-based BoN or BS achieves success probability $\Theta(1)$ on task $k$;*

- *ORM/PRM-based BoN or BS fails on any other task $k' \neq k$.*

**Proof.***Consider task tuple $(q, a, k)$ and an instance $(\mathbf{Q}, \mathbf{A}) \in \mathcal{D}_{a_q}^{a,k}$ that is solvable, i.e., it admits at least one valid CoT in $\mathcal{G}_{q,a_q}^{(k)}$. Since the base model $\theta^\star$ assigns $\Theta(c^{L-1})$ sampling probability to a correct CoT, the success probability of ORM-based BoN using the ground-truth reward $R_{\mathbf{Q},\mathbf{A}}^k(\cdot)$ satisfies:*

$$pass@K = \Theta\left(1 - (1 - c^{L-1})^{NK}\right) = \Theta(1),$$

*where the final equality holds for sufficiently large $K$.*

*For ORM-based BoN under outcome-population reward $R_{\text{out}}^k(\cdot)$, the CoT credit depends on relative likelihood. Consider the worst case where there is exactly one correct CoT with success probability $\Theta(c^{L-1})$, while each incorrect but valid CoT has sampling probability $\Theta(1/M^{L-1})$ (by Lemma 2), and dominates $R_{\text{out}}^k(\cdot)$. Then the probability of sampling the correct CoT at least once in $N$ attempts, while avoiding any misleading CoTs, is:*

$$\Theta\left(\left[1 - \frac{1}{M^{L-1}}\right]^N \cdot \left[1 - (1 - c^{L-1})^N\right]\right).$$

*Hence, the pass@K success probability is lower bounded by:*

$$\Theta\left(1 - \left(1 - \left[1 - \frac{1}{M^{L-1}}\right]^N \cdot \left[1 - (1 - c^{L-1})^N\right]\right)^K\right) = \Theta(1),$$

*again holding when $K$ is large.*

*Now consider PRM-based BoN under Eq.(11. At each step, the minimal success probability is:*

$$\Theta\left([1 - \tfrac{1}{M}]^N \cdot [1 - (1 - c)^N]\right),$$

*so across $L - 1$ steps, the overall probability is:*

$$\Theta\left([1 - \tfrac{1}{M}]^{N(L-1)} \cdot [1 - (1 - c)^{N(L-1)}]\right),$$

*and the corresponding pass@K is lower bounded by:*

$$\Theta\left(1 - \left(1 - [1 - \tfrac{1}{M}]^{N(L-1)} \cdot [1 - (1 - c)^{N(L-1)}]\right)^K\right) = \Theta(1).$$

*Now consider any different task $k' \neq k$. By Definitions 1 and 2, the oracle rewards $R_{\mathbf{Q},\mathbf{A}}^k(\cdot)$, as well as the PRMs in Eqs. 11, all assign zero credit to instances sampled from $k'$. Therefore, all ORM/PRM-based BoN or BS strategies fail on task $k'$.*

*For Beam Search (BS), the result follows by analogous arguments since BS depends on the same reward signals layer-wise.* □

**Proof of Thm. 3.** *Fix any instance $(\mathbf{Q}, \mathbf{A})$ of task $(o_1, a, k)$ and assume the premise of the theorem: all correct CoTs are hard-to-reason and there exists at least one depth $l^\star \in [L]$ at which the hard CoTs diverge from a valid easy-to-reason CoT ("sparse edge"). Let $\boldsymbol{o}^{\text{easy}}$ denote one such easy CoT and $\boldsymbol{o}^{\text{hard}}$ any hard CoT. By Prop. 2, population-level ORM and PRM scores strictly prefer the easy branch whenever they differ:*

$$R_{\text{out}}^k(\boldsymbol{o}^{\text{easy}}) > R_{\text{out}}^k(\boldsymbol{o}^{\text{hard}}), \qquad R_{\text{likelihood}}^k(\boldsymbol{o}_l^{\text{easy}}) > R_{\text{likelihood}}^k(\boldsymbol{o}_l^{\text{hard}}) \quad \text{for all } l \text{ with } \boldsymbol{o}_l^{\text{easy}} \neq \boldsymbol{o}_l^{\text{hard}}. \tag{A}$$

*We analyze (i) and (ii)&(iii) separately. Throughout, $M$ is the per-node branching factor and $L$ is the CoT length. We take the conservative lower bounds that (a) at each node a particular child has sampling probability at least $1/M$, and (b) samples across the $N$ trials are i.i.d.*

*(i) ORM + BoN. Best-of-$N$ (BoN) first draws $N$ full trajectories (CoTs) i.i.d. from the generator and then selects the one with the largest ORM score $R_{\text{out}}^k(\cdot)$. By equation A, if among the $N$ samples there exists at least one $\boldsymbol{o}^{\text{easy}}$, BoN will select an easy CoT, hence it will fail under the theorem's premise (easy branch is valid but leads away from any correct hard solution due to the sparse-edge divergence).*

*We bound the probability that at least one $\boldsymbol{o}^{\text{easy}}$ appears among $N$ samples. Consider any fixed easy CoT $\boldsymbol{o}^{\text{easy}}$ that agrees with $\boldsymbol{o}^{\text{hard}}$ on the prefix up to (but excluding) $l^\star$ and then takes a different child at $l^\star$. A conservative lower bound on the probability of sampling this specific easy CoT in one draw is*

$$p_{traj} \geq \left(\frac{1}{M}\right)^{L-1} = \frac{1}{M^{L-1}},$$

*since at $L-1$ branching decisions (excluding the terminal) we multiply the minimal per-step mass $1/M$. Hence the probability that none of the $N$ i.i.d. draws equals this easy trajectory is*

$$(1 - p_{traj})^N \leq \left(1 - \frac{1}{M^{L-1}}\right)^N = \left(\frac{M^L - M}{M^L}\right)^N.$$

*Therefore, with probability at least $1 - (1 - 1/M^{L-1})^N$ an easy CoT appears among the $N$ draws, and by equation A BoN selects it and thus fails. Imposing*

$$\left(1 - \frac{1}{M^{L-1}}\right)^N \leq \epsilon \iff N \geq \frac{\log(\epsilon)}{\log\left(\frac{M^L - M}{M^L}\right)},$$

*ensures that the failure probability is at least $1 - \epsilon$, which proves the first bullet.*

*(ii) PRM + BoN (step-wise) and (iii) PRM + Beam Search (width $N$, beam $B \geq 1$). PRM-based inference expands partial CoTs and uses the local PRM score $R_{\text{likelihood}}^k(\boldsymbol{o}_l)$ to select among candidates. Consider the first divergence depth $l^\star$. In each expansion round at depth $l^\star$, the procedure proposes $N$ children i.i.d. (BoN: propose and take the best child by PRM; Beam: propose $N$ and keep the top-$B$ by PRM). Let*

$$p_{child} \geq \frac{1}{M}$$

*be the conservative lower bound that a given proposal at depth $l^\star$ takes the (PRM-favored) easy child rather than the hard sparse edge. Thus the probability that none of the $N$ proposals includes the easy child at that step is*

$$(1 - p_{child})^N \leq \left(1 - \frac{1}{M}\right)^N = \left(\frac{M-1}{M}\right)^N.$$

*Consequently, with probability at least $1 - (1 - 1/M)^N$ the easy child appears among the $N$ proposals at depth $l^\star$. By equation A, PRM strictly prefers that easy child over the hard child at depth $l^\star$, so:*

- **PRM + BoN (step-wise):** *the chosen next token is the easy child, irrevocably steering the trajectory onto the easy branch. Repeating this argument at later depths where branches differ keeps the easy path strictly preferred, so the final selection is easy and the method fails under the theorem's premise.*

- **PRM + Beam Search:** *since $B \geq 1$, any PRM-strictly-better easy child is ranked above the hard child and therefore included in the beam at depth $l^\star$; by standard beam monotonicity with strictly better local scores at each subsequent divergence, the easy branch remains in the top-$B$ and dominates the final selection, hence failure.*

*Imposing*

$$\left(1 - \tfrac{1}{M}\right)^N \leq \epsilon \quad \Longleftrightarrow \quad N \geq \frac{\log(\epsilon)}{\log\left(\frac{M-1}{M}\right)},$$

*ensures that an easy child appears at the first divergence step with probability at least $1 - \epsilon$, and by the PRM preference this forces selection of the easy branch, completing the second bullet.*

***Conclusion.*** *In all cases, Prop. 2 ensures a strict scoring advantage for the easy branch whenever it is present among candidates; the displayed lower bounds control the probability that such an easy candidate* does *appear given $N$ proposals. Choosing $N$ to satisfy*

$$\left(1 - \tfrac{1}{M^{L-1}}\right)^N \leq \epsilon \quad \text{(ORM + BoN)}, \qquad \left(1 - \tfrac{1}{M}\right)^N \leq \epsilon \quad \text{(PRM + BoN/BS)},$$

*yields failure probability at least $1 - \epsilon$ for (i) and for (ii)&(iii), respectively.* $\qquad\square$

**Proof.** *Heuristic Proof of Corollary 3. Let $(\Omega, \mathcal{F}, \mu)$ be a base measure space where $\hat{p}_{\boldsymbol{\theta}^\star} \ll \mu$ with Radon-Nikodym derivative $d\hat{p}_{\boldsymbol{\theta}^\star}/d\mu > 0$ $\mu$-a.e. We consider the optimization over absolutely continuous measures $P_{new}^k \ll \hat{p}_{\boldsymbol{\theta}^\star}$.*

*The objective functional can be written as:*

$$J(P_{new}^k) = \mathbb{E}_{P_{new}^k}[R(\boldsymbol{o})] - \frac{1}{\lambda} D_{KL}(P_{new}^k \| \hat{p}_{\boldsymbol{\theta}^\star}) \tag{77}$$

*where $R(\boldsymbol{o}) := R_{\text{out}}^k(\boldsymbol{o})$. We require:*

*(C1)* $R \in L^1(\hat{p}_{\boldsymbol{\theta}^\star})$ *(finite expected reward)*

*(C2)* $\exists \epsilon > 0$ *s.t.* $\hat{p}_{\boldsymbol{\theta}^\star} \geq \epsilon$ *$\mu$-a.e. (strict positivity)*

*High-levelly, the remaining proof is convex optimization in probability space. Define the Lagrangian with measure-theoretic notation:*

$$\mathcal{L}(P, \eta) = \int R \, dP - \frac{1}{\lambda} \int \log\left(\frac{dP}{d\hat{p}_{\boldsymbol{\theta}^\star}}\right) dP + \eta\left(1 - \int dP\right) \tag{78}$$

*Require:*

*(C3)* $P \in \mathcal{P}(\Omega)$, *the space of probability measures absolutely continuous to $\mu$*

*(C4)* $\log(dP/d\hat{p}_{\boldsymbol{\theta}^\star}) \in L^1(P)$ *(finite KL divergence)*

*For $P \in \mathcal{P}(\Omega)$, consider variation $P_\epsilon = P + \epsilon Q$ where $Q$ is a signed measure with $\int dQ = 0$. The Gâteaux derivative is:*

$$\frac{d}{d\epsilon}\mathcal{L}(P_\epsilon, \eta)\Big|_{\epsilon=0} = \int R \, dQ - \frac{1}{\lambda}\int\left(\log\frac{dP}{d\hat{p}_{\boldsymbol{\theta}^\star}} + 1\right) dQ - \eta \int dQ \tag{79}$$

*For optimality, this must vanish for all admissible $Q$, requiring:*

$$R(\boldsymbol{o}) - \frac{1}{\lambda}\left(\log\frac{dP}{d\hat{p}_{\boldsymbol{\theta}^\star}}(\boldsymbol{o}) + 1\right) - \eta = 0 \quad P\text{-a.s.} \tag{80}$$

*Rearranging gives:*

$$\log\frac{dP}{d\hat{p}_{\boldsymbol{\theta}^\star}} = \lambda R(\boldsymbol{o}) - (1 + \lambda\eta) \tag{81}$$

*Exponentiating both sides:*

$$dP = \hat{p}_{\boldsymbol{\theta}^\star}(\boldsymbol{o}) \exp(\lambda R(\boldsymbol{o})) \exp(-1 - \lambda\eta) d\mu(\boldsymbol{o}) \tag{82}$$

*Normalization requires:*

$$\exp(1 + \lambda\eta) = \int \hat{p}_{\boldsymbol{\theta}^\star} \exp(\lambda R) d\mu =: Z \tag{83}$$

*Thus the optimal measure is:*

$$dP^k_{adjusted} = \frac{1}{Z} \hat{p}_{\boldsymbol{\theta}^\star} \exp(\lambda R) d\mu \tag{84}$$

*First verify* $P^k_{adjusted} \in \mathcal{P}(\Omega)$:

- *Absolute continuity: Immediate from* $\hat{p}_{\boldsymbol{\theta}^\star} \ll \mu$ *and* $Z^{-1} \exp(\lambda R) > 0$
- *Integrability: By (C1) and* $\exp(\lambda R) \leq \exp(\lambda\|R\|_\infty) < \infty$ *from* $R \leq 1$

*Second, confirm stationarity. For any* $Q \in T_{P^k_{adjusted}} \mathcal{P}(\Omega)$ *(tangent space):*

$$d\mathcal{L}(P^k_{adjusted}, \eta)(Q) = \int \underbrace{\left[ R - \frac{1}{\lambda}(\log \frac{dP^k_{adjusted}}{d\hat{p}_{\boldsymbol{\theta}^\star}} + 1) - \eta \right]}_{=0} dQ = 0 \tag{85}$$

*Substitute* $P^k_{adjusted}$ *into J:*

$$J(P^k_{adjusted}) = \mathbb{E}_{P^k_{adjusted}}[R] - \frac{1}{\lambda} \mathbb{E}_{P^k_{adjusted}} \left[ \log \frac{P^k_{adjusted}}{\hat{p}_{\boldsymbol{\theta}^\star}} \right]$$

$$= \mathbb{E}_{P^k_{adjusted}}[R] - \frac{1}{\lambda} \left( \lambda \mathbb{E}[R] - \log Z \right)$$

$$= \frac{1}{\lambda} \log Z$$

*By Gibbs' inequality, this maximizes the trade-off between expected reward and KL regularization.*

*To validate our conditions required, we summarized:*

- *(C1): Holds as* $\|R\|_\infty \leq 1$ *by assumption*
- *(C2): Guaranteed by model construction* $\hat{p}_{\boldsymbol{\theta}^\star} = softmax(\cdot) > 0$
- *(C3): Inherited from base measure* $\mu$
- *(C4): Satisfied because* $D_{KL}(P^k_{adjusted}\|\hat{p}_{\boldsymbol{\theta}^\star}) = \log Z - \lambda\mathbb{E}[R] < \infty$

*Thus under these conditions,* $P^k_{adjusted}$ *is the unique maximizer of* $J(P^k_{new})$ *in* $\mathcal{P}(\Omega)$. $\quad\square$

**Proof.***Proof of the legitimacy of Def.3. To show* $h_k(\cdot)$ *is a harmonic function, let us verify*

$$1 = \sum_{\boldsymbol{o}'_{l+1} \in S_{l+1}} \hat{p}^{new,k}_{\boldsymbol{\theta}}(\boldsymbol{o}'_{l+1}|\boldsymbol{o}_l) = \sum_{\boldsymbol{o}'_{l+1} \in S_{l+1}} \hat{p}_{\boldsymbol{\theta}^\star}(\boldsymbol{o}'_{l+1}|\boldsymbol{o}_l) \frac{h_k(\boldsymbol{o}'_{l+1})}{h_k(\boldsymbol{o}_l)}.$$

*By the fact that*

$$h_k(\boldsymbol{o}_l) = \mathbb{E}_{\boldsymbol{o}_{l+1:L} \sim \hat{p}_{\boldsymbol{\theta}^\star}} \left[ \exp\left(\lambda R^k_{out}(\boldsymbol{o})\right) \mid \boldsymbol{o}_l \right]$$

$$= \sum_{\boldsymbol{o}'_{l+1} \in S_{l+1}} \hat{p}_{\boldsymbol{\theta}^\star}(\boldsymbol{o}'_{l+1}|\boldsymbol{o}_l) \mathbb{E}_{\boldsymbol{o}'_{l+2:L} \sim \hat{p}_{\boldsymbol{\theta}^\star}} \left[ \exp\left(\lambda R^k_{out}(\boldsymbol{o}')\right) \mid \boldsymbol{o}'_{l+1} \right] \tag{86}$$

*we see that*

$$\sum_{\boldsymbol{o}'_{l+1} \in S_{l+1}} \hat{p}_{\boldsymbol{\theta}^\star}(\boldsymbol{o}'_{l+1}|\boldsymbol{o}_l) \frac{h_k(\boldsymbol{o}'_{l+1})}{h_k(\boldsymbol{o}_l)}$$

$$= \sum_{\boldsymbol{o}'_{l+1} \in S_{l+1}} \hat{p}_{\boldsymbol{\theta}^\star}(\boldsymbol{o}'_{l+1}|\boldsymbol{o}_l) \frac{\mathbb{E}_{\boldsymbol{o}'_{l+2:L} \sim \hat{p}_{\boldsymbol{\theta}^\star}} \left[ \exp\left(\lambda R_{\text{out}}^k(\boldsymbol{o}')\right) \mid \boldsymbol{o}'_{l+1} \right]}{\mathbb{E}_{\boldsymbol{o}_{l+1:L} \sim \hat{p}_{\boldsymbol{\theta}^\star}} \left[ \exp\left(\lambda R_{\text{out}}^k(\boldsymbol{o})\right) \mid \boldsymbol{o}_l \right]}$$

$$= \sum_{\boldsymbol{o}'_{l+1} \in S_{l+1}} \hat{p}_{\boldsymbol{\theta}^\star}(\boldsymbol{o}'_{l+1}|\boldsymbol{o}_l) \frac{\mathbb{E}_{\boldsymbol{o}'_{l+2:L} \sim \hat{p}_{\boldsymbol{\theta}^\star}} \left[ \exp\left(\lambda R_{\text{out}}^k(\boldsymbol{o}')\right) \mid \boldsymbol{o}'_{l+1} \right]}{\sum_{\boldsymbol{o}'_{l+1} \in S_{l+1}} \hat{p}_{\boldsymbol{\theta}^\star}(\boldsymbol{o}'_{l+1}|\boldsymbol{o}_l) \mathbb{E}_{\boldsymbol{o}'_{l+2:L} \sim \hat{p}_{\boldsymbol{\theta}^\star}} \left[ \exp\left(\lambda R_{\text{out}}^k(\boldsymbol{o}')\right) \mid \boldsymbol{o}'_{l+1} \right]} = 1$$

*Recall the definition of our **DPRM**:*

$$R_{\text{DPRM}}^k(\boldsymbol{o}_l) = \frac{1}{\lambda} \log \left( \mathbb{E}_{\boldsymbol{o}'_{l+1:L} \sim \hat{p}_{\boldsymbol{\theta}^\star}} \left[ \exp\left(\lambda R_{\text{out}}^k(\boldsymbol{o}')\right) \mid \boldsymbol{o}_l \right]. \right), \tag{87}$$

$$\hat{p}_{\boldsymbol{\theta}}^{new,k}(\boldsymbol{o}_{l+1}|\boldsymbol{o}_l) = \hat{p}_{\boldsymbol{\theta}^\star}(\boldsymbol{o}_{l+1}|\boldsymbol{o}_l) \frac{h_k(\boldsymbol{o}_{l+1})}{h_k(\boldsymbol{o}_l)} = \frac{\hat{p}_{\boldsymbol{\theta}^\star}(\boldsymbol{o}_{l+1}|\boldsymbol{o}_l) \exp\left(\lambda R_{\text{DPRM}}^k(\boldsymbol{o}_{l+1})\right)}{Z_l(\boldsymbol{o}_l)},$$

*where $Z_l(\boldsymbol{o}_l) = \sum_{\boldsymbol{o}'_{l+1} \in S_{l+1}} \hat{p}_{\boldsymbol{\theta}^\star}(\boldsymbol{o}'_{l+1}|\boldsymbol{o}_l) \exp\left(\lambda R_{\text{DPRM}}^k(\boldsymbol{o}'_{l+1})\right)$. Collaborating with Eq.(88) as well as the definition of $R_{\text{DPRM}}^k(\boldsymbol{o}_l)$, we can equate:*

$$\exp\left(\lambda R_{\text{DPRM}}^k(\boldsymbol{o}_l)\right) = h_k(\boldsymbol{o}_l) = \mathbb{E}_{\boldsymbol{o}'_{l+1:L} \sim \hat{p}_{\boldsymbol{\theta}^\star}} \left[ \exp\left(\lambda R_{\text{out}}^k(\boldsymbol{o}')\right) \mid \boldsymbol{o}_l \right].$$

*Therefore, it holds that*

$$\hat{p}_{\boldsymbol{\theta}}^{new,k}(\boldsymbol{o}_{l+1} \mid \boldsymbol{o}_l) = \hat{p}_{\boldsymbol{\theta}^\star}(\boldsymbol{o}_{l+1} \mid \boldsymbol{o}_l) \cdot \frac{h_k(\boldsymbol{o}_{l+1})}{h_k(\boldsymbol{o}_l)} \propto \hat{p}_{\boldsymbol{\theta}}(\boldsymbol{o}_{l+1} \mid \boldsymbol{o}_l) \exp(\lambda R_{\text{DPRM}}^k(\boldsymbol{o}_{l+1})).$$

*Recall:*

$$h_k(\boldsymbol{o}_l) = \mathbb{E}_{\boldsymbol{o}_{l+1:L} \sim \hat{p}_{\boldsymbol{\theta}^\star}} \left[ \exp\left(\lambda R_{\text{out}}^k(\boldsymbol{o})\right) \mid \boldsymbol{o}_l \right],$$

*so $h_k(\boldsymbol{o}_L) = \exp(\lambda R_{\text{out}}^k(\boldsymbol{o}))$ and $h_k(\boldsymbol{o}_0) = Z := \sum_{\boldsymbol{o}' \in \mathcal{T}_{all}} \hat{p}_{\boldsymbol{\theta}^\star}(\boldsymbol{o}') \exp\left(\lambda R_{\text{out}}^k(\boldsymbol{o}')\right)$. The h-transformed transition is:*

$$\hat{p}_{\boldsymbol{\theta}}^{new,k}(\boldsymbol{o}_{l+1}|\boldsymbol{o}_l) = \hat{p}_{\boldsymbol{\theta}^\star}(\boldsymbol{o}_{l+1}|\boldsymbol{o}_l) \frac{h_k(\boldsymbol{o}_{l+1})}{h_k(\boldsymbol{o}_l)}, \tag{88}$$

*yielding:*

$$P_{\text{DPRM}}^k(\boldsymbol{o}) = \prod_{l=1}^{L-1} \hat{p}_{\boldsymbol{\theta}^\star}(\boldsymbol{o}_{l+1}|\boldsymbol{o}_l) \frac{h_k(\boldsymbol{o}_{l+1})}{h_k(\boldsymbol{o}_l)} = \hat{p}_{\boldsymbol{\theta}^\star}(\boldsymbol{o}) \frac{h_k(\boldsymbol{o}_L)}{h_k(\boldsymbol{o}_0)} = P_{\text{Gibbs}}^k(\boldsymbol{o}). \tag{89}$$

*The proof is completed.*

$\square$

**Corollary 8.** *For the task $k \in \mathcal{T}$, let $\boldsymbol{\theta}^\star$ be the pretrained Foundation Model from Thm. 8, and $R_{\text{out}}^k(\cdot)$ be the ORM. Consider the ORM-equipped and DPRM-equipped adjusted sampling distributions defined in Corollary 3.*

- *As the temperature parameter $\lambda \to \infty$, we have the following situation*

    1. ***ORM-Equipped Adjusted Sampling**: The distribution in Eq.(12) converges to:*

$$P_{\text{Gibbs}}^k(\boldsymbol{o}) \xrightarrow{\lambda \to \infty} \begin{cases} 1, & \text{if } \boldsymbol{o} = \arg\max_{\boldsymbol{o}' \in \mathcal{T}_{all}} R_{\text{out}}^k(\boldsymbol{o}') \\ 0, & \text{otherwise} \end{cases}$$

    *akin to a ORM-based **BoN** with $R_{\text{out}}^k(\cdot)$.*

2. **DPRM-Equipped Adjusted Sampling**: *The step-wise distribution (14) with* $R_{\text{DPRM}}^k(\boldsymbol{o}_{l+1}) = \frac{1}{\lambda} \log h_k(\boldsymbol{o}_{l+1})$ *converges to:*

$$\hat{p}_{\boldsymbol{\theta}}^{new,k}(\boldsymbol{o}_{l+1} \mid \boldsymbol{o}_l) \xrightarrow{\lambda \to \infty} \begin{cases} 1, & \text{if } \boldsymbol{o}_{l+1} = \arg\max_{\boldsymbol{o}' \in S_{l+1}} R_{\text{likelihood}}^k(\boldsymbol{o}') \\ 0, & \text{otherwise} \end{cases}$$

*akin to a PRM-based **BoN** with $R_{\text{likelihood}}^k(\cdot)$.*

- *When the temperature parameter $\lambda > 0$, each step $l \in \{0, \ldots, L-1\}$ satisfies:*

$$\arg\max_{\boldsymbol{o}_l \in S_l^{BoN}} R_{\text{DPRM}}^k(\boldsymbol{o}_l) = \arg\max_{\boldsymbol{o}_l \in S_l^{BoN}} \mathbb{E}_{\boldsymbol{o}_{l+1:L} \sim \hat{p}_{\boldsymbol{\theta}^\star}} R_{\text{likelihood}}^k(\boldsymbol{o}_l),$$

*where $S_l^{BoN} = \{\boldsymbol{o}_l^1, \ldots, \boldsymbol{o}_l^N\}$ denotes the $N$ candidates sampled by the base model $\hat{p}_{\boldsymbol{\theta}^\star}$. Therefore, using $\lambda > 0$ with **BoN**, **Beam Search** or **Lookahead Search** equates to prior PRM methods employing the same search strategies.*

**Proof.** *Proof of Corollary 4. We analyze the asymptotic behavior of the sampling distributions as $\lambda \to \infty$.*

*For part (1), consider the ORM-equipped adjusted sampling distribution:*

$$P_{\text{Gibbs}}^k(\boldsymbol{o}) = \frac{P_{\boldsymbol{\theta}^\star}(\boldsymbol{o}) \exp\left(\lambda R_{\text{out}}^k(\boldsymbol{o})\right)}{\sum_{\boldsymbol{o}' \in \mathcal{T}_{all}} P_{\boldsymbol{\theta}^\star}(\boldsymbol{o}') \exp\left(\lambda R_{\text{out}}^k(\boldsymbol{o}')\right)},$$

*where $\mathcal{T}_{all}$ is the set of all possible trajectories. Let $\boldsymbol{o}^* = \arg\max_{\boldsymbol{o}' \in \mathcal{T}_{all}} R_{\text{out}}^k(\boldsymbol{o}')$, with maximum reward $R_{\text{out}}^k(\boldsymbol{o}^*)$. As $\lambda \to \infty$, the term $\exp\left(\lambda R_{\text{out}}^k(\boldsymbol{o})\right)$ dominates for $\boldsymbol{o}$ with the largest $R_{\text{out}}^k(\boldsymbol{o})$. For $\boldsymbol{o} \neq \boldsymbol{o}^*$, if $R_{\text{out}}^k(\boldsymbol{o}) < R_{\text{out}}^k(\boldsymbol{o}^*)$, then*

$$\frac{\exp\left(\lambda R_{\text{out}}^k(\boldsymbol{o})\right)}{\exp\left(\lambda R_{\text{out}}^k(\boldsymbol{o}^*)\right)} = \exp\left(\lambda(R_{\text{out}}^k(\boldsymbol{o}) - R_{\text{out}}^k(\boldsymbol{o}^*))\right) \to 0,$$

*since $R_{\text{out}}^k(\boldsymbol{o}) - R_{\text{out}}^k(\boldsymbol{o}^*) < 0$. Assuming $R_{\text{out}}^k(\boldsymbol{o})$ has a unique maximum (or summing over all maximizers if not unique), the denominator is dominated by $P_{\boldsymbol{\theta}^\star}(\boldsymbol{o}^*) \exp\left(\lambda R_{\text{out}}^k{}^k(\boldsymbol{o}^*)\right)$. Thus,*

$$P_{\text{Gibbs}}^k(\boldsymbol{o}) \to \begin{cases} 1, & \text{if } \boldsymbol{o} = \boldsymbol{o}^*, \\ 0, & \text{otherwise}, \end{cases}$$

*which matches the behavior of BoN Sampling, where the trajectory with the highest $R_{\text{out}}^k{}^k(\boldsymbol{o})$ is selected.*

*For part (2), consider the DPRM-equipped step-wise distribution:*

$$\hat{p}_{\boldsymbol{\theta}}^{new,k}(\boldsymbol{o}_{l+1} \mid \boldsymbol{o}_l) = \hat{p}_{\boldsymbol{\theta}^\star}(\boldsymbol{o}_{l+1} \mid \boldsymbol{o}_l) \frac{h_k(\boldsymbol{o}_{l+1})}{h_k(\boldsymbol{o}_l)},$$

*with $h_k(\boldsymbol{o}_{l+1}) = \mathbb{E}_{\boldsymbol{o}_{l+2:L} \sim \hat{p}_{\boldsymbol{\theta}^\star}}\left[\exp\left(\lambda R_{\text{out}}^k(\boldsymbol{o})\right) \mid \boldsymbol{o}_{l+1}\right]$, and $R_{\text{DPRM}}^k(\boldsymbol{o}_{l+1}) = \frac{1}{\lambda} \log h_k(\boldsymbol{o}_{l+1})$. Substituting $h_k$, we get*

$$\hat{p}_{\boldsymbol{\theta}}^{new,k}(\boldsymbol{o}_{l+1} \mid \boldsymbol{o}_l) = \hat{p}_{\boldsymbol{\theta}^\star}(\boldsymbol{o}_{l+1} \mid \boldsymbol{o}_l) \exp\left(\lambda \left(R_{\text{DPRM}}^k(\boldsymbol{o}_{l+1}) - R_{\text{DPRM}}^k(\boldsymbol{o}_l)\right)\right) \frac{1}{Z},$$

*where $Z = \sum_{\boldsymbol{o}_{l+1} \in S_{l+1}} \hat{p}_{\boldsymbol{\theta}^\star}(\boldsymbol{o}_{l+1} \mid \boldsymbol{o}_l) \exp\left(\lambda R_{\text{DPRM}}^k(\boldsymbol{o}_{l+1})\right)$ is the normalizing constant. Let $\boldsymbol{o}_{l+1}^* = \arg\max_{\boldsymbol{o}' \in S_{l+1}} R_{\text{DPRM}}^k(\boldsymbol{o}')$. As $\lambda \to \infty$, the term $\exp\left(\lambda R_{\text{DPRM}}^k(\boldsymbol{o}_{l+1})\right)$ dominates for $\boldsymbol{o}_{l+1} = \boldsymbol{o}_{l+1}^*$. For $\boldsymbol{o}_{l+1} \neq \boldsymbol{o}_{l+1}^*$, if $R_{\text{DPRM}}^k(\boldsymbol{o}_{l+1}) < R_{\text{DPRM}}^k(\boldsymbol{o}_{l+1}^*)$, then*

$$\frac{\exp\left(\lambda R_{\text{DPRM}}^k(\boldsymbol{o}_{l+1})\right)}{\exp\left(\lambda R_{\text{DPRM}}^k(\boldsymbol{o}_{l+1}^*)\right)} = \exp\left(\lambda(R_{\text{DPRM}}^k(\boldsymbol{o}_{l+1}) - R_{\text{DPRM}}^k(\boldsymbol{o}_{l+1}^*))\right) \to 0.$$

*Thus, the distribution concentrates on $\boldsymbol{o}_{l+1}^*$:*

$$\hat{p}_{\boldsymbol{\theta}}^{new,k}(\boldsymbol{o}_{l+1} \mid \boldsymbol{o}_l) \to \begin{cases} 1, & \text{if } \boldsymbol{o}_{l+1} = \boldsymbol{o}_{l+1}^*, \\ 0, & \text{otherwise}, \end{cases}$$

*which mimics BoN Sampling by selecting the state with the highest $R_{\text{likelihood}}^k$ by our last item.*

*For the last item, for step $l \in \{0, \ldots, L-1\}$, the DPRM with $\lambda > 0$ is given as $R_{\text{DPRM}}^k(\boldsymbol{o}_l) = \log \mathbb{E}_{\boldsymbol{o}_{l+1:L} \sim \hat{p}_{\boldsymbol{\theta}^\star}} \left[ \exp \left( R_{\text{out}}^k {}^k(\boldsymbol{o}) \right) \mid \boldsymbol{o}_l \right]$. Since $\log$ is strictly increasing, we have*

$$\arg\max_{\boldsymbol{o}_l \in S_l^{BoN}} R_{\text{DPRM}}^k(\boldsymbol{o}_l) = \arg\max_{\boldsymbol{o}_l \in S_l^{BoN}} \mathbb{E}_{\boldsymbol{o}_{l+1:L} \sim \hat{p}_{\boldsymbol{\theta}^\star}} \left[ \exp \left( R_{\text{out}}^k {}^k(\boldsymbol{o}) \right) \mid \boldsymbol{o}_l \right].$$

*Similarly, since $\exp$ is strictly increasing, the $\arg\max$ over $\mathbb{E}_{\boldsymbol{o}_{l+1:L} \sim \hat{p}_{\boldsymbol{\theta}^\star}} \left[ \exp \left( R_{\text{out}}^k {}^k(\boldsymbol{o}) \right) \mid \boldsymbol{o}_l \right]$ is equivalent to the $\arg\max$ over $\mathbb{E}_{\boldsymbol{o}_{l+1:L} \sim \hat{p}_{\boldsymbol{\theta}^\star}} \left[ R_{\text{out}}^k {}^k(\boldsymbol{o}) \mid \boldsymbol{o}_l \right]$. Thus, given that $R_{\text{likelihood}}^k(\boldsymbol{o}_l) = \mathbb{E}_{\boldsymbol{o}_{l+1:L} \sim \hat{p}_{\boldsymbol{\theta}^\star}} \left[ R_{\text{out}}^k {}^k(\boldsymbol{o}) \mid \boldsymbol{o}_l \right]$ it holds that*

$$\arg\max_{\boldsymbol{o}_l \in S_l^{BoN}} R_{\text{DPRM}}^k(\boldsymbol{o}_l) = \arg\max_{\boldsymbol{o}_l \in S_l^{BoN}} \mathbb{E}_{\boldsymbol{o}_{l+1:L} \sim \hat{p}_{\boldsymbol{\theta}^\star}} R_{\text{likelihood}}^k(\boldsymbol{o}_l).$$

*This shows that BoN Sampling with $R_{pro}^k$ maximizes the expected outcome reward, aligning with prior methods. The equivalence for Beam Search follows similarly by replacing the sampling strategy with the respective search method, as they also maximize $R_{pro}^k(\boldsymbol{o}_l)$. This completes the proof.* □

**Proof.** *Proof of Cor. 5. The proof directly follows the proof of Cor. 7.* □

**Corollary 9** (Extension: Comparison with Ground-true Oracle). *Let $\boldsymbol{\theta}^\star$ be the base model in Eq.(2 that exactly predicts the distribution of a Multi-task TMC as in Definitions 1 and 2. Under task tuple $(q, a, k) \in S_1 \times S_L \times \mathcal{T}$, consider the ORMs $R_{\mathbf{Q},\mathbf{A}}^k(\cdot)$ and $R_{\text{out}}^k(\cdot)$, and the PRMs of Eqs. 11. For any target task $k$ with instance distribution $\mathcal{D}_{a_q}^{a,k}$, suppose the number of hard-to-reason CoTs is $\Theta(M)$ and the number of nonzero-probability CoTs from $q$ to $S_L$ is $N_q$. Then under pass@K sampling:*

1. *DPRM is More Capable of Hard CoTs. If a specific hard CoT has sampling probability $p = o(M^{-(L-1)})$ under the base model, then for any BoN budget*

$$N = O\left( \frac{\log(1 - N_q^{-1})}{\log(1-p)} \right),$$

*there exists $\lambda = o\left( \ln \frac{(1-p)^N}{(N_q-1)(1-(1-p)^N)} \right)$ such that DPRM with temperature $\lambda$ achieves strictly higher pass@K than ORM-based or PRM-based BoN (or BS).*

2. *Preserve Multi-task. For any $\varepsilon > 0$, if*

$$K = \Omega\left( \frac{\ln \varepsilon}{\ln\left( \frac{(N_q-1)e^\lambda}{1+(N_q-1)e^\lambda} \right)} \right),$$

*then DPRM with $\lambda > 0$ attains pass@K $\geq 1 - \varepsilon$ on any other task $k' \neq k$.*

*In both cases, adjusting the temperature $\lambda > 0$ controls the pass@K performance.*

**Proof.** *The arguments parallel those in Cor. 7, so we focus on the comparison of pass@K success probabilities.*

*(i) Hard-CoT capability. Under ORM-based BoN with ground-truth reward, the success probability for the unique hard CoT is*

$$1 - (1-p)^N.$$

*Under DPRM (Eq.(12), every valid CoT—including the correct one—has sampling probability at least*

$$\frac{1}{N_q - M + Me^\lambda} \geq \frac{1}{1 + (N_q - 1)e^\lambda}.$$

*Choosing $\lambda = o\left( \ln \frac{(1-p)^N}{(N_q-1)(1-(1-p)^N)} \right)$ ensures $\frac{1}{1+(N_q-1)e^\lambda} \geq 1 - (1-p)^N$, so DPRM outperforms ORM. Similarly, when the budget of PRM-based BoN (or BS) in pass@K is limited and $\lambda \to 0$ would achieve more satisfactory success probability.*

**(ii) Multi-task preservation.** *For any other task $k'$, DPRM still assigns probability at least $\frac{1}{1+(N_q-1)e^\lambda}$ to each valid CoT. Thus, with*

$$K = \Omega\left(\frac{\ln \varepsilon}{\ln\left(\frac{(N_q-1)e^\lambda}{1+(N_q-1)e^\lambda}\right)}\right),$$

*the pass@K guarantee $1 - \left(1 - \frac{1}{1+(N_q-1)e^\lambda}\right)^K \geq 1 - \varepsilon$ holds, completing the proof.* □

## L   AUXILIARY LEMMAS

**Lemma 6.** *Let $\boldsymbol{\theta}^\star$ be the base model*

$$\hat{p}_{\boldsymbol{\theta}}(\cdot|\boldsymbol{x}) = \mathrm{softmax}(h_{\boldsymbol{\theta}}(\cdot, \boldsymbol{x})), \ \boldsymbol{x} \in \{0,1\}^{|S|}. \tag{90}$$

*Then for $\forall \boldsymbol{o}_l \in S_l, \boldsymbol{o}_{l+1} \in S_{l+1}$*

$$\nabla_{\boldsymbol{\theta}^k} \log \hat{p}_{\boldsymbol{\theta}^k}(\boldsymbol{o}_{l+1}|\boldsymbol{o}_l) = \nabla_{\boldsymbol{\theta}^k} h_{\boldsymbol{\theta}}(\boldsymbol{o}_{l+1}, \boldsymbol{o}_l) - \sum_{\boldsymbol{o}'_{l+1} \in S_{l+1}} \hat{p}_{\boldsymbol{\theta}^k}(\boldsymbol{o}'_{l+1}|\boldsymbol{o}_l)\nabla_{\boldsymbol{\theta}^k} h_{\boldsymbol{\theta}}(\boldsymbol{o}'_{l+1}, \boldsymbol{o}_l). \tag{91}$$

*Further, if the base model is Eq.(2), we have*

$$\nabla_{\boldsymbol{\theta}^k} \log \hat{p}_{\boldsymbol{\theta}^k}(\boldsymbol{o}_{l+1}|\boldsymbol{o}_l) = e_{\boldsymbol{o}_{l+1},\boldsymbol{o}_l} - \sum_{\boldsymbol{o}'_{l+1} \in S_{l+1}} \hat{p}_{\boldsymbol{\theta}^k}(\boldsymbol{o}'_{l+1}|\boldsymbol{o}_l)\nabla_{\boldsymbol{\theta}^k} e_{\boldsymbol{o}'_{l+1},\boldsymbol{o}_l}, \tag{92}$$

*where $e_{\boldsymbol{o}_{l+1},\boldsymbol{o}_l} := \boldsymbol{o}_{l+1}\boldsymbol{o}_l^\top \in \{0,1\}^{|S|\times|S|}$ is the one-hot matrix with only the position corresponding to $(\boldsymbol{o}_{l+1}, \boldsymbol{o}_l)$ is 1 and 0 elsewhere.*

**Proof.** *By Eq. (90), we have*

$$
\begin{aligned}
\nabla_{\boldsymbol{\theta}^k} \log \hat{p}_{\boldsymbol{\theta}^k}(\boldsymbol{o}_{l+1}|\boldsymbol{o}_l) &= \nabla_{\boldsymbol{\theta}^k} \log \frac{e^{h_{\boldsymbol{\theta}}(\boldsymbol{o}_{l+1},\boldsymbol{o}_l)}}{\sum_{\boldsymbol{o}'_{l+1} \in S_{l+1}} e^{h_{\boldsymbol{\theta}}(\boldsymbol{o}'_{l+1},\boldsymbol{o}_l)}} \\
&= \nabla_{\boldsymbol{\theta}^k} h_{\boldsymbol{\theta}}(\boldsymbol{o}_{l+1},\boldsymbol{o}_l) - \nabla_{\boldsymbol{\theta}^k} \log \sum_{\boldsymbol{o}'_{l+1} \in S_{l+1}} e^{h_{\boldsymbol{\theta}}(\boldsymbol{o}'_{l+1},\boldsymbol{o}_l)} \\
&= \nabla_{\boldsymbol{\theta}^k} h_{\boldsymbol{\theta}}(\boldsymbol{o}_{l+1},\boldsymbol{o}_l) - \frac{\nabla_{\boldsymbol{\theta}^k} \sum_{\boldsymbol{o}'_{l+1} \in S_{l+1}} e^{h_{\boldsymbol{\theta}}(\boldsymbol{o}'_{l+1},\boldsymbol{o}_l)}}{\sum_{\boldsymbol{o}'_{l+1} \in S_{l+1}} e^{h_{\boldsymbol{\theta}}(\boldsymbol{o}'_{l+1},\boldsymbol{o}_l)}} \\
&= \nabla_{\boldsymbol{\theta}^k} h_{\boldsymbol{\theta}}(\boldsymbol{o}_{l+1},\boldsymbol{o}_l) - \frac{\sum_{\boldsymbol{o}'_{l+1} \in S_{l+1}} e^{h_{\boldsymbol{\theta}}(\boldsymbol{o}'_{l+1},\boldsymbol{o}_l)}\nabla_{\boldsymbol{\theta}^k} h_{\boldsymbol{\theta}}(\boldsymbol{o}'_{l+1},\boldsymbol{o}_l)}{\sum_{\boldsymbol{o}'_{l+1} \in S_{l+1}} e^{h_{\boldsymbol{\theta}}(\boldsymbol{o}'_{l+1},\boldsymbol{o}_l)}} \\
&= \nabla_{\boldsymbol{\theta}^k} h_{\boldsymbol{\theta}}(\boldsymbol{o}_{l+1},\boldsymbol{o}_l) - \sum_{\boldsymbol{o}'_{l+1} \in S_{l+1}} \hat{p}_{\boldsymbol{\theta}^k}(\boldsymbol{o}'_{l+1}|\boldsymbol{o}_l)\nabla_{\boldsymbol{\theta}^k} h_{\boldsymbol{\theta}}(\boldsymbol{o}'_{l+1},\boldsymbol{o}_l).
\end{aligned}
\tag{93}
$$

*Besides, if the base model is $\hat{p}_{\boldsymbol{\theta}}(\cdot|x) = \mathrm{softmax}(\langle \boldsymbol{\theta}, x \rangle)$ by Eq.(2), we have*

$$\nabla_{\boldsymbol{\theta}^k} h_{\boldsymbol{\theta}}(\boldsymbol{o}_{l+1},\boldsymbol{o}_l) = \nabla_{\boldsymbol{\theta}^k} \langle \boldsymbol{\theta}_{\boldsymbol{o}_{l+1},\cdot}, \boldsymbol{o}_l \rangle = e_{\boldsymbol{o}_{l+1},\boldsymbol{o}_l}, \tag{94}$$

*Dragging Eq.(94) into Eq.(91), we could obtain Eq.(92).*

*The proof is completed.* □

**Lemma 7** (Policy Gradient for REINFORCE & RAFT under TMC). *Let $\boldsymbol{\theta}^\star$ be the base model in Eq.(2) that exact predicts the distribution of Multi-task TMC as in Def. 1 and 2, and $\boldsymbol{\theta}^k$ the current model to be finetuned from $\boldsymbol{\theta}^\star$ for task $k \in \mathcal{T}$. The gradient of the REINFORCE objective for task $k$ is given by:*

$$\nabla_{\boldsymbol{\theta}^k} \mathcal{J}_{\mathrm{REINFORCE}}(\boldsymbol{\theta}^k) = \sum_{l=1}^{L-1} \mathbb{E}_{\substack{\boldsymbol{o}_1 \sim P^k(\mathcal{Q}^k) \\ \{\boldsymbol{o}_{l+1} \sim \hat{p}_{\boldsymbol{\theta}^k}(\cdot|\boldsymbol{o}_l)\}_{l=1}^{L-1}}} \left[\nabla_{\boldsymbol{\theta}^k} \log \hat{p}_{\boldsymbol{\theta}^k}(\boldsymbol{o}_{l+1}|\boldsymbol{o}_l)R_{\mathrm{out}}^k(\boldsymbol{o})\right], \tag{95}$$

$$\nabla_{\boldsymbol{\theta}^k} \mathcal{J}_{\text{RAFT}}(\boldsymbol{\theta}^k) = \sum_{l=1}^{L-1} \mathbb{E}_{\substack{\boldsymbol{o}_1 \sim P^k(\mathcal{Q}^k) \\ \{\boldsymbol{o}_{l+1} \sim \hat{p}_{\boldsymbol{\theta}^k}(\cdot|\boldsymbol{o}_l)\}_{l=1}^{L-1}}} \tag{96}$$

$$\left[ (1 + \log \hat{p}_{\boldsymbol{\theta}^k}(\boldsymbol{o}_{l+1}|\boldsymbol{o}_l)) \nabla_{\boldsymbol{\theta}^k} \log \hat{p}_{\boldsymbol{\theta}^k}(\boldsymbol{o}_{l+1}|\boldsymbol{o}_l) R_{\text{out}}^k(\boldsymbol{o}) \right], \tag{97}$$

*where*

$$\mathcal{J}_{\text{REINFORCE}}(\boldsymbol{\theta}^k) = \mathbb{E}_{\boldsymbol{o}_1 \sim P^k(\mathcal{Q}^k), (\mathbf{Q}, \mathbf{A}) \sim \mathcal{D}_{a_{o_1}}^{o_1, k}, \boldsymbol{o}_{2:L} \sim \hat{p}_{\boldsymbol{\theta}^k}^k(O|\boldsymbol{o}_1)} \left[ R_{(\mathbf{Q}, \mathbf{A})}^k(\boldsymbol{o}) \right], \tag{98}$$

$$\mathcal{J}_{\text{RAFT}}(\boldsymbol{\theta}^k) = \mathbb{E}_{\boldsymbol{o}_1 \sim P^k(\mathcal{Q}^k), (\mathbf{Q}, \mathbf{A}) \sim \mathcal{D}_{a_{o_1}}^{o_1, k}, \boldsymbol{o}_{2:L} \sim \hat{p}_{\boldsymbol{\theta}^k}^k(O|\boldsymbol{o}_1)} \left[ \sum_{l=1}^{L-1} \log \hat{p}_{\boldsymbol{\theta}^k}(\boldsymbol{o}_{l+1}|\boldsymbol{o}_l) R_{(\mathbf{Q}, \mathbf{A})}^k(\boldsymbol{o}) \right], \tag{99}$$

**Remark 13.** *In the main text, Eq.(4 contains a typo: the summation term "$\sum_{l=1}^{L-1}$" inside the expectation is omitted. The correct formulation is provided in Eq.(99. Additionally, the formal versions of Eq.(7 and Eq.(8 are given as Eq.(95 and Eq.(97, respectively.*

**Proof.** *For any complete trajectory $\boldsymbol{o} = (o_1, ..., o_L)$:*

$$\hat{p}_{\boldsymbol{\theta}^k}(\boldsymbol{o}) = P^k(\mathcal{Q}^k) \prod_{l=1}^{L-1} \hat{p}_{\boldsymbol{\theta}^k}(o_{l+1}|o_l) \tag{100}$$

*where $P^k(\mathcal{Q}^k)$ is the initial state distribution (parameter-independent by Def. 1). By the property of TMC, we have*

$$\nabla_{\boldsymbol{\theta}^k} \mathcal{J}_{\text{REINFORCE}}(\boldsymbol{\theta}^k) = \nabla_{\boldsymbol{\theta}^k} \mathbb{E}_{\boldsymbol{o}_1 = q \sim P^k(\mathcal{Q}^k), \boldsymbol{o}_{2:L} \sim \hat{p}_{\boldsymbol{\theta}^k}^k(O|\boldsymbol{o}_1)} \left[ R_{\text{out}}^k(\boldsymbol{o}) \right]$$

$$\overset{(1)}{=} \nabla_{\boldsymbol{\theta}^k} \int_{\mathcal{O}^L} R_{\text{out}}^k(\boldsymbol{o}) \left[ P^k(q) \prod_{l=1}^{L-1} \hat{p}_{\boldsymbol{\theta}^k}(\boldsymbol{o}_{l+1}|\boldsymbol{o}_l) \right] d\boldsymbol{o}_{1:L}$$

$$\overset{(2)}{=} \int_{\mathcal{O}^L} R_{\text{out}}^k(\boldsymbol{o}) P^k(q) \nabla_{\boldsymbol{\theta}^k} \left[ \prod_{l=1}^{L-1} \hat{p}_{\boldsymbol{\theta}^k}(\boldsymbol{o}_{l+1}|\boldsymbol{o}_l) \right] d\boldsymbol{o}_{1:L}$$

$$\overset{(3)}{=} \int_{\mathcal{O}^L} R_{\text{out}}^k(\boldsymbol{o}) P^k(q) \left[ \prod_{l=1}^{L-1} \hat{p}_{\boldsymbol{\theta}^k}(\boldsymbol{o}_{l+1}|\boldsymbol{o}_l) \right] \sum_{l=1}^{L-1} \nabla_{\boldsymbol{\theta}^k} \log \hat{p}_{\boldsymbol{\theta}^k}(\boldsymbol{o}_{l+1}|\boldsymbol{o}_l) d\boldsymbol{o}_{1:L}$$

$$\overset{(4)}{=} \mathbb{E}_{\boldsymbol{o}_1 \sim P^k(\mathcal{Q}^k), \boldsymbol{o}_{2:L} \sim \hat{p}_{\boldsymbol{\theta}^k}^k(O|\boldsymbol{o}_1)}$$

$$\left[ R_{\text{out}}^k(\boldsymbol{o}) \sum_{l=1}^{L-1} \nabla_{\boldsymbol{\theta}^k} \log \hat{p}_{\boldsymbol{\theta}^k}(\boldsymbol{o}_{l+1}|\boldsymbol{o}_l) \right]$$

$$\overset{(5)}{=} \sum_{l=1}^{L-1} \mathbb{E}_{\boldsymbol{o}_1 \sim P^k(\mathcal{Q}^k), \boldsymbol{o}_{2:L} \sim \hat{p}_{\boldsymbol{\theta}^k}^k(O|\boldsymbol{o}_1)} \left[ \nabla_{\boldsymbol{\theta}^k} \log \hat{p}_{\boldsymbol{\theta}^k}(\boldsymbol{o}_{l+1}|\boldsymbol{o}_l) R_{\text{out}}^k(\boldsymbol{o}) \right].$$

$$\tag{101}$$

*Step (1) expands the expectation as an integral over trajectories using the MDP's joint distribution $P^k(q) \prod_{t=1}^{L-1} \hat{p}_{\boldsymbol{\theta}^k}(\boldsymbol{o}_{t+1}|\boldsymbol{o}_t)$;*

*Step (2) applies the Leibniz interchange under Markovian policy structure:*

$$\nabla_{\boldsymbol{\theta}^k} \int_{\mathcal{O}^L} R_{\text{out}}^k(\boldsymbol{o}) \hat{p}_{\boldsymbol{\theta}^k}(\boldsymbol{o}_{1:L}) d\boldsymbol{o}_{1:L} = \int_{\mathcal{O}^L} R_{\text{out}}^k(\boldsymbol{o}) \nabla_{\boldsymbol{\theta}^k} \hat{p}_{\boldsymbol{\theta}^k}(\boldsymbol{o}_{1:L}) d\boldsymbol{o}_{1:L} \quad (a.s.) \tag{102}$$

*valid when: (i) Policy Gradient Dominance: $\exists h \in L^1(\mu)$ such that $\|R_{\text{out}}^k(\boldsymbol{o}) \nabla_{\boldsymbol{\theta}^k} \hat{p}_{\boldsymbol{\theta}^k}(\boldsymbol{o}_{1:L})\| \leq h(\boldsymbol{o}_{1:L}) \; \forall \boldsymbol{\theta}^k \in \Theta^k$ where $\Theta^k = \mathbb{R}^{|S| \times |S|}$ denotes the parameter space; (ii) Parameterized Measure Continuity: The map $\boldsymbol{\theta}^k \mapsto \sqrt{\hat{p}_{\boldsymbol{\theta}^k}(\boldsymbol{o}_{1:L})}$ is $W^{1,1}$-continuous with: $\lim_{\|\boldsymbol{v}\| \to 0} \mathbb{E}_\mu \left[ \left\| \frac{\sqrt{\hat{p}_{\boldsymbol{\theta}^k + \boldsymbol{v}}} - \sqrt{\hat{p}_{\boldsymbol{\theta}^k}}}{\|\boldsymbol{v}\|} \right\|^2 \right] < \infty$, which are all satisfied under our case since $\|R_{\text{out}}^k(\cdot)\|_\infty = O(1)$ and $\hat{p}_{\boldsymbol{\theta}^k}(\cdot|x) = \text{softmax}(\langle \boldsymbol{\theta}, x \rangle)$ by Eq.(2);*

*Step (3) decomposes using Markovian parameter isolation:*

$$\nabla_{\boldsymbol{\theta}^k} \prod_{l=1}^{L-1} \hat{p}_{\boldsymbol{\theta}^k}(\boldsymbol{o}_{l+1}|\boldsymbol{o}_l) = \sum_{l=1}^{L-1} \left( \prod_{m=1}^{L-1} \hat{p}_{\boldsymbol{\theta}^k}(\boldsymbol{o}_{m+1}|\boldsymbol{o}_m) \right) \nabla_{\boldsymbol{\theta}^k} \log \hat{p}_{\boldsymbol{\theta}^k}(\boldsymbol{o}_{l+1}|\boldsymbol{o}_l) \tag{103}$$

*valid under: (i) Disjoint Parameter Control:* $\boldsymbol{\theta}^k = \biguplus_{l=1}^{L-1} \boldsymbol{\theta}_l^k$ *where* $\boldsymbol{\theta}_l^k \cap \boldsymbol{\theta}_{l'}^k = \emptyset$ *for* $l' \neq l$, *with each* $\hat{p}_{\boldsymbol{\theta}^k}(\boldsymbol{o}_{l+1}|\boldsymbol{o}_l) = f_l(\boldsymbol{o}_{l+1}|\boldsymbol{o}_l; \boldsymbol{\theta}_l^k)$ *and* $\frac{\partial f_l}{\partial \boldsymbol{\theta}_{l'}^k} \equiv 0$; *(ii) Log-Smoothness:* $\hat{p}_{\boldsymbol{\theta}^k}(\boldsymbol{o}_{l+1}|\boldsymbol{o}_l) > 0$ $\mu$-*a.e. and* $\nabla_{\boldsymbol{\theta}^k} \log \hat{p}_{\boldsymbol{\theta}^k}(\boldsymbol{o}_{l+1}|\boldsymbol{o}_l) \in L^2(\hat{p}_{\boldsymbol{\theta}^k} \otimes \mu)$; *(iii) Sequential Fubini Condition:* $\int_{\mathcal{O}^L} \prod_{l=1}^{L-1} \hat{p}_{\boldsymbol{\theta}^k}(\boldsymbol{o}_{l+1}|\boldsymbol{o}_l) d\boldsymbol{o}_{1:L} = \prod_{l=1}^{L-1} \int_{\mathcal{O}} \hat{p}_{\boldsymbol{\theta}^k}(\boldsymbol{o}_{l+1}|\boldsymbol{o}_l) d\boldsymbol{o}_{l+1}$ *in terms of total variation norm, which are all easily verified under our* $\hat{p}_{\boldsymbol{\theta}^k}(\cdot|x) = \mathrm{softmax}(\langle \boldsymbol{\theta}, x \rangle)$ *by Eq.(2);*

*Step (4) rewrites the integral as* $\mathbb{E}_{\boldsymbol{o}_{1:L} \sim \hat{p}_{\boldsymbol{\theta}^k}}[R_{\mathrm{out}}^k(\boldsymbol{o}) \sum_{l=1}^{L-1} \nabla_{\boldsymbol{\theta}^k} \log \hat{p}_{\boldsymbol{\theta}^k}(\boldsymbol{o}_{l+1}|\boldsymbol{o}_l)]$;

*Step (5) exchanges summation and expectation via Fubini's theorem, valid when* $\mathbb{E}[|R_{\mathrm{out}}^k \nabla_{\boldsymbol{\theta}^k} \log \hat{p}_{\boldsymbol{\theta}^k}|] < \infty$, *which obviously hold in our setting. Similarly, we have*

$$
\begin{aligned}
\nabla_{\boldsymbol{\theta}^k} \mathcal{J}_{\mathrm{RAFT}}(\boldsymbol{\theta}^k) &= \nabla_{\boldsymbol{\theta}^k} \mathbb{E}_{\substack{\boldsymbol{o}_1 = q \sim P^k(\mathcal{Q}^k) \\ \boldsymbol{o}_{t+1} \sim \hat{p}_{\boldsymbol{\theta}^k}(\cdot|\boldsymbol{o}_t)}} \left[ \sum_{l=1}^{L-1} \log \hat{p}_{\boldsymbol{\theta}^k}(\boldsymbol{o}_{l+1}|\boldsymbol{o}_l) \cdot R_{\mathrm{out}}^k(\boldsymbol{o}_{1:L}) \right] \\
&\overset{(1)}{=} \sum_{l=1}^{L-1} \nabla_{\boldsymbol{\theta}^k} \left( \int_{\mathcal{O}^L} R_{\mathrm{out}}^k(\boldsymbol{o}_{1:L}) \cdot P^k(\boldsymbol{o}_1) \cdot \prod_{\substack{t=1 \\ t \neq l}}^{L-1} \hat{p}_{\boldsymbol{\theta}^k}(\boldsymbol{o}_{t+1}|\boldsymbol{o}_t) \cdot \left[ \hat{p}_{\boldsymbol{\theta}^k}(\boldsymbol{o}_{l+1}|\boldsymbol{o}_l) \log \hat{p}_{\boldsymbol{\theta}^k}(\boldsymbol{o}_{l+1}|\boldsymbol{o}_l) \right] d\boldsymbol{o}_{1:L} \right) \\
&\overset{(2)}{=} \sum_{l=1}^{L-1} \int_{\mathcal{O}^L} R_{\mathrm{out}}^k(\boldsymbol{o}_{1:L}) \cdot P^k(\boldsymbol{o}_1) \cdot \left( \prod_{\substack{t=1 \\ t \neq l}}^{L-1} \hat{p}_{\boldsymbol{\theta}^k}(\boldsymbol{o}_{t+1}|\boldsymbol{o}_t) \right) \cdot \nabla_{\boldsymbol{\theta}^k} \left[ \hat{p}_{\boldsymbol{\theta}^k}(\boldsymbol{o}_{l+1}|\boldsymbol{o}_l) \log \hat{p}_{\boldsymbol{\theta}^k}(\boldsymbol{o}_{l+1}|\boldsymbol{o}_l) \right] d\boldsymbol{o}_{1:L} \\
&\overset{(3)}{=} \sum_{l=1}^{L-1} \int_{\mathcal{O}^L} R_{\mathrm{out}}^k(\boldsymbol{o}_{1:L}) \cdot P^k(\boldsymbol{o}_1) \left( \prod_{t=1}^{L-1} \hat{p}_{\boldsymbol{\theta}^k}(\boldsymbol{o}_{t+1}|\boldsymbol{o}_t) \right) \cdot \frac{\nabla_{\boldsymbol{\theta}^k} \left[ \hat{p}_{\boldsymbol{\theta}^k}(\boldsymbol{o}_{l+1}|\boldsymbol{o}_l) \log \hat{p}_{\boldsymbol{\theta}^k}(\boldsymbol{o}_{l+1}|\boldsymbol{o}_l) \right]}{\hat{p}_{\boldsymbol{\theta}^k}(\boldsymbol{o}_{l+1}|\boldsymbol{o}_l)} d\boldsymbol{o}_{1:L} \\
&\overset{(4)}{=} \sum_{l=1}^{L-1} \mathbb{E}_{\substack{\boldsymbol{o}_1 \sim P^k(\mathcal{Q}^k) \\ \boldsymbol{o}_{t+1} \sim \hat{p}_{\boldsymbol{\theta}^k}(\cdot|\boldsymbol{o}_t)}} \left[ R_{\mathrm{out}}^k(\boldsymbol{o}_{1:L}) \left( 1 + \log \hat{p}_{\boldsymbol{\theta}^k}(\boldsymbol{o}_{l+1}|\boldsymbol{o}_l) \right) \cdot \nabla_{\boldsymbol{\theta}^k} \log \hat{p}_{\boldsymbol{\theta}^k}(\boldsymbol{o}_{l+1}|\boldsymbol{o}_l) \right] \\
&\overset{(5)}{=} \mathbb{E}_{\substack{\boldsymbol{o}_1 \sim P^k(\mathcal{Q}^k) \\ \boldsymbol{o}_{t+1} \sim \hat{p}_{\boldsymbol{\theta}^k}(\cdot|\boldsymbol{o}_t)}} \left[ R_{\mathrm{out}}^k(\boldsymbol{o}_{1:L}) \cdot \sum_{l=1}^{L-1} \left( 1 + \log \hat{p}_{\boldsymbol{\theta}^k}(\boldsymbol{o}_{l+1}|\boldsymbol{o}_l) \right) \cdot \nabla_{\boldsymbol{\theta}^k} \log \hat{p}_{\boldsymbol{\theta}^k}(\boldsymbol{o}_{l+1}|\boldsymbol{o}_l) \right]
\end{aligned}
\tag{104}
$$

*Step (1) expands the expectation using the Markovianity's factorized structure* $P^k(\boldsymbol{o}_1) \prod_{t=1}^{L-1} \hat{p}_{\boldsymbol{\theta}^k}(\boldsymbol{o}_{t+1}|\boldsymbol{o}_t)$, *isolating the l-th transition's* $\hat{p} \log \hat{p}$ *term while keeping others as standard transitions, which is legitimate under our* $\hat{p}_{\boldsymbol{\theta}^k}(\cdot|x) = \mathrm{softmax}(\langle \boldsymbol{\theta}, x \rangle)$ *by Eq.(2);*

*Step (2) enforces parameter-localized differentiation through:*

$$\sum_{l=1}^{L-1} \nabla_{\boldsymbol{\theta}^k} \int F_l d\boldsymbol{o} = \sum_{l=1}^{L-1} \int_{\mathcal{O}^L} R_{\mathrm{out}}^k(\boldsymbol{o}_{1:L}) \cdot P^k(\boldsymbol{o}_1) \cdot \left( \prod_{\substack{t=1 \\ t \neq l}}^{L-1} \hat{p}_{\boldsymbol{\theta}^k}(\boldsymbol{o}_{t+1}|\boldsymbol{o}_t) \right) \cdot \nabla_{\boldsymbol{\theta}^k} \left[ \hat{p}_{\boldsymbol{\theta}^k}(\boldsymbol{o}_{l+1}|\boldsymbol{o}_l) \log \hat{p}_{\boldsymbol{\theta}^k}(\boldsymbol{o}_{l+1}|\boldsymbol{o}_l) \right] d\boldsymbol{o}_{1:L} \quad (a.s.) \tag{105}$$

*where* $F_l = R_{\mathrm{out}}^k(\boldsymbol{o}_{1:L}) \cdot P^k(\boldsymbol{o}_1) \cdot \prod_{\substack{t=1 \\ t \neq l}}^{L-1} \hat{p}_{\boldsymbol{\theta}^k}(\boldsymbol{o}_{t+1}|\boldsymbol{o}_t) \cdot [\hat{p}_{\boldsymbol{\theta}^k}(\boldsymbol{o}_{l+1}|\boldsymbol{o}_l) \log \hat{p}_{\boldsymbol{\theta}^k}(\boldsymbol{o}_{l+1}|\boldsymbol{o}_l)]$, *valid when:*

*(i) Architectural Parameter Isolation: Policy parameters partition as* $\boldsymbol{\theta}^k = \biguplus_{l=1}^{L-1} \boldsymbol{\theta}_l^k$ *with:*

$$\frac{\partial}{\partial \boldsymbol{\theta}_m^k} \hat{p}_{\boldsymbol{\theta}^k}(\boldsymbol{o}_{t+1}|\boldsymbol{o}_t) = \begin{cases} \nabla_{\boldsymbol{\theta}_l^k} \hat{p}_{\boldsymbol{\theta}^k}(\boldsymbol{o}_{l+1}|\boldsymbol{o}_l) & t = l \text{ and } m = l \\ 0 & \text{otherwise} \end{cases} \tag{106}$$

*which is satisfied as* $\hat{p}_{\boldsymbol{\theta}^k}(\cdot|x) = \mathrm{softmax}(\langle \boldsymbol{\theta}, x \rangle)$ *by Eq.(2); (ii) Localized Dominance:* $\exists h_l \in L^1(\mu_l)$ *where* $\mu_l$ *is the base measure on* $(\boldsymbol{o}_l, \boldsymbol{o}_{l+1})$, *such that:*

$$|R_{\mathrm{out}}^k(\boldsymbol{o}_{1:L}) \cdot \nabla_{\boldsymbol{\theta}^k}[\hat{p}_{\boldsymbol{\theta}^k}(\boldsymbol{o}_{l+1}|\boldsymbol{o}_l) \log \hat{p}_{\boldsymbol{\theta}^k}(\boldsymbol{o}_{l+1}|\boldsymbol{o}_l)]| \leq h_l(\boldsymbol{o}_l, \boldsymbol{o}_{l+1}), \tag{107}$$

*which clearly held under our condiions; (iii) Decoupled Integration: For each l,*

$$\int_{\mathcal{O}^L} F_l d\boldsymbol{o} = \int_{\boldsymbol{o}_1} P^k \int_{\boldsymbol{o}_{l+1}} [\hat{p}_{\boldsymbol{\theta}^k} \log \hat{p}_{\boldsymbol{\theta}^k}] \left( \prod_{\substack{t=1 \\ t \neq l}}^{L-1} \int_{\boldsymbol{o}_{t+1}} \hat{p}_{\boldsymbol{\theta}^k} d\boldsymbol{o}_{t+1} \right) d\boldsymbol{o}_l d\boldsymbol{o}_{l+1} \tag{108}$$

with $\prod_{\substack{t=1 \\ t\neq l}}^{L-1} \int \hat{p}_{\boldsymbol{\theta}^k} d\boldsymbol{o}_{t+1} = 1$ $\mu$-a.e. This condition holds apparently under our model $\hat{p}_{\boldsymbol{\theta}^k}(\cdot|x) = $ softmax($\langle\boldsymbol{\theta}, x\rangle$) by Eq.(2), which linearly isolates each states; (iv) Transition Differentiability: Each $\boldsymbol{\theta}_l^k \mapsto \hat{p}_{\boldsymbol{\theta}^k}(\boldsymbol{o}_{l+1}|\boldsymbol{o}_l)$ is Fréchet differentiable with: $\mathbb{E}\left[\left\|\frac{\nabla_{\boldsymbol{\theta}^k}\hat{p}_{\boldsymbol{\theta}^k}(\boldsymbol{o}_{l+1}|\boldsymbol{o}_l)}{\hat{p}_{\boldsymbol{\theta}^k}(\boldsymbol{o}_{l+1}|\boldsymbol{o}_l)}\right\|^2\right] < \infty$ which holds in our softmax model;

Step (3) multiplies one $\hat{p}_{\boldsymbol{\theta}^k}(\boldsymbol{o}_{l+1}|\boldsymbol{o}_l)$ in the front and divide it subsequently;

Step (4) uses the chain rule:

$$\nabla_{\boldsymbol{\theta}^k}(\hat{p}_{\boldsymbol{\theta}^k}(\boldsymbol{o}_{l+1}|\boldsymbol{o}_l)\log\hat{p}_{\boldsymbol{\theta}^k}(\boldsymbol{o}_{l+1}|\boldsymbol{o}_l)) = (1 + \log\hat{p}_{\boldsymbol{\theta}^k}(\boldsymbol{o}_{l+1}|\boldsymbol{o}_l))\nabla_{\boldsymbol{\theta}^k}\hat{p}_{\boldsymbol{\theta}^k}(\boldsymbol{o}_{l+1}|\boldsymbol{o}_l),$$

and reconstructs the expectation by recognizing $\prod_{t=1}^{L-1}\hat{p}_t = \hat{p}_{\boldsymbol{\theta}^k}(\boldsymbol{o}_{1:L})/P^k(\boldsymbol{o}_1)$, with cross-terms vanishing due to $\mathbb{E}_{\boldsymbol{o}_{m+1}\sim\hat{p}_m}[f(\boldsymbol{o}_l)] = \mathbb{E}[f(\boldsymbol{o}_l)]$ for $m \neq l$; Step (5) applies Fubini's theorem to exchange summation and expectation, valid by the fact $\mathbb{E}\left[\sum_{l=1}^{L-1}|(1+\log\hat{p}_l)\nabla\log\hat{p}_l\,\mathrm{Rex}|\right] < \infty$ in our case.

□

**Remark 14.** *When the base model is no longer in the linear form in Eq.(2), but a general form in Eq.(90) with $\hat{p}_{\boldsymbol{\theta}}(\cdot|\boldsymbol{x}) = \mathrm{softmax}(h_{\boldsymbol{\theta}}(\cdot,\boldsymbol{x}))$, $\boldsymbol{x} \in \{0,1\}^{|S|}$, the conclusions still holds when*

- *Architectural Conditions*
    - *Parameter Isolation: $\boldsymbol{\theta} = \biguplus_{l=1}^{L-1}\boldsymbol{\theta}_l$ where $\boldsymbol{\theta}_l \cap \boldsymbol{\theta}_{l'} = \emptyset$ for $l \neq l'$, with:*

    $$h_{\boldsymbol{\theta}}(\boldsymbol{o}_{l+1}|\boldsymbol{o}_l) = h_l(\boldsymbol{o}_{l+1}|\boldsymbol{o}_l;\boldsymbol{\theta}_l), \quad \frac{\partial h_l}{\partial\boldsymbol{\theta}_{l'}} \equiv 0 \;\forall l' \neq l \quad (109)$$

    - *Module Independence: Each $h_l(\cdot;\boldsymbol{\theta}_l)$ uses distinct computational subgraphs without parameter sharing across $l$*

- *Smoothness & Differentiability*
    - *Lipschitz Continuity: $\exists C_l > 0$ s.t.*

    $$\|h_l(\cdot;\boldsymbol{\theta}_l + \Delta\boldsymbol{\theta}) - h_l(\cdot;\boldsymbol{\theta}_l)\|_\infty \leq C_l\|\Delta\boldsymbol{\theta}\|_2 \quad \forall\boldsymbol{\theta}_l \quad (110)$$

    - *Twice Differentiability: $h_l \in C^2(\Theta_l)$ with bounded Hessians:*

    $$\mathbb{E}_{\boldsymbol{o}_l}\left[\|\nabla_{\boldsymbol{\theta}_l}^2 h_l\|_{\mathrm{op}}^2\right] < \infty \quad (111)$$

- *Gradient Control*
    - *Bounded Logits: $\exists C < \infty$ s.t.*

    $$\max_a |h_l(a,\boldsymbol{o}_l)| \leq C \quad \forall\boldsymbol{o}_l, l \quad (112)$$

    - *Gradient Norm Bound:*

    $$\mathbb{E}_{\boldsymbol{o}_l}\left[\|\nabla_{\boldsymbol{\theta}_l} h_l\|_2^2\right] \leq B_l < \infty \quad \forall l \quad (113)$$

- *Probability Regularity*
    - *Strict Positivity: $\exists\epsilon > 0$ s.t.*

    $$\hat{p}_{\boldsymbol{\theta}}(\boldsymbol{o}_{l+1}|\boldsymbol{o}_l) \geq \epsilon \quad \mu\text{-a.e. }\forall l \quad (114)$$

    - *Measure Consistency:*

    $$\int_{\mathcal{O}}\hat{p}_{\boldsymbol{\theta}}(\boldsymbol{o}_{l+1}|\boldsymbol{o}_l)d\boldsymbol{o}_{l+1} = 1 \quad \forall\boldsymbol{o}_l, l \quad (115)$$

*These conditions guarantee: 1. Leibniz rule applicability; through Lipschitz continuity 2. Fubini's theorem validity via measure consistency; 3. Gradient dominance via bounded logits; 4. Policy smoothness via $C^2$ differentiability; 5. Numerical stability through strict positivity.*

**Lemma 8** (Policy Gradient for PO (Eq.(67)) under TMC). *Let $\boldsymbol{\theta}^\star$ be the base model in Eq.(2) that exact predicts the distribution of Multi-task TMC as in Def. 1 and 2, and $\boldsymbol{\theta}^k$ the current model to be finetuned from $\boldsymbol{\theta}^\star$ for task $k \in \mathcal{T}$. Suggest the accurate $A_{l+1}^{\hat{p}_{\boldsymbol{\theta}^k},k}(\boldsymbol{o}_l, \boldsymbol{o}_{l+1})$ is available from some outer oracle, the clip operation is always active, and Eq.(66) holds. The gradient of the PO objective for task $k$ is given by:*

$$\nabla_{\boldsymbol{\theta}^k}\mathcal{J}_{\mathrm{PO}}(\boldsymbol{\theta}^k) = \sum_{l=1}^{L-1}\mathbb{E}_{\substack{\boldsymbol{o}_1 \sim P^k(\mathcal{Q}^k)\\ \{\boldsymbol{o}_{l+1} \sim \hat{p}_{\boldsymbol{\theta}^k}(\cdot|\boldsymbol{o}_l)\}_{l=1}^{L-1}}}\left[(1 + (2\mathbb{1}(A_{l+1}^{\hat{p}_{\boldsymbol{\theta}^k},k}(\boldsymbol{o}_l,\boldsymbol{o}_{l+1}) \geq 0) - 1)\epsilon_{\mathrm{clip}})A_{l+1}^{\hat{p}_{\boldsymbol{\theta}^k},k}(\boldsymbol{o}_l,\boldsymbol{o}_{l+1}) \cdot \nabla_{\boldsymbol{\theta}^k}\log\hat{p}_{\boldsymbol{\theta}^k}(\boldsymbol{o}_{l+1}|\boldsymbol{o}_l)\right], \tag{116}$$

$$\tag{117}$$

*where $r_{l+1} = \frac{\hat{p}_{\boldsymbol{\theta}^k}(\boldsymbol{o}_{l+1}|\boldsymbol{o}_l)}{\hat{p}_{old}^k(\boldsymbol{o}_{l+1}|\boldsymbol{o}_l)}$. By the condition that the clip operation is always active, we have*

$$\mathcal{J}_{\mathrm{PO}} = \mathbb{E}_{\boldsymbol{o}_1 \sim P^k(\mathcal{Q}^k),(\mathbf{Q},\mathbf{A})\sim\mathcal{D}_{a_{o_1}}^{o_1,k},\boldsymbol{o}_{2:L}\sim\hat{p}_{\boldsymbol{\theta}^k}(O|\boldsymbol{o}_1)}\left[\frac{1}{L}\sum_{l=1}^{L-1}(1 + (2\mathbb{1}(A_{l+1}^{\hat{p}_{\boldsymbol{\theta}^k},k}(\boldsymbol{o}_l,\boldsymbol{o}_{l+1}) \geq 0) - 1)\epsilon_{\mathrm{clip}})A_{l+1}^{\hat{p}_{\boldsymbol{\theta}^k},k}(\boldsymbol{o}_l,\boldsymbol{o}_{l+1})\right], \tag{118}$$

**Proof.** *It holds that*

$$\begin{aligned}\nabla_{\boldsymbol{\theta}^k}\mathcal{J}_{\mathrm{PO}} &= \nabla_{\boldsymbol{\theta}^k}\mathbb{E}_{\substack{\boldsymbol{o}_1 \sim P^k\\ \boldsymbol{o}_{t+1}\sim\hat{p}_{\boldsymbol{\theta}^k}}}\left[\frac{1}{L}\sum_{l=1}^{L-1}(1 + (2\mathbb{1}(A_{l+1}^{\hat{p}_{\boldsymbol{\theta}^k},k}(\boldsymbol{o}_l,\boldsymbol{o}_{l+1}) \geq 0) - 1)\epsilon_{\mathrm{clip}})A_{l+1}^{\hat{p}_{\boldsymbol{\theta}^k},k}(\boldsymbol{o}_l,\boldsymbol{o}_{l+1})\right]\\ &\stackrel{(1)}{=} \frac{1}{L}\sum_{l=1}^{L-1}\nabla_{\boldsymbol{\theta}^k}\int_{\mathcal{O}^L}(1 + (2\mathbb{1}(A_{l+1}^{\hat{p}_{\boldsymbol{\theta}^k},k}(\boldsymbol{o}_l,\boldsymbol{o}_{l+1}) \geq 0) - 1)\epsilon_{\mathrm{clip}})A_{l+1}^{\hat{p}_{\boldsymbol{\theta}^k},k}(\boldsymbol{o}_l,\boldsymbol{o}_{l+1})P^k(\boldsymbol{o}_1)\prod_{t=1}^{L-1}\hat{p}_{\boldsymbol{\theta}^k}(\boldsymbol{o}_{t+1}|\boldsymbol{o}_t)d\boldsymbol{o}_{1:L}\\ &\stackrel{(2)}{=} \frac{1}{L}\sum_{l=1}^{L-1}\int_{\mathcal{O}^L}(1 + (2\mathbb{1}(A_{l+1}^{\hat{p}_{\boldsymbol{\theta}^k},k}(\boldsymbol{o}_l,\boldsymbol{o}_{l+1}) \geq 0) - 1)\epsilon_{\mathrm{clip}})A_{l+1}^{\hat{p}_{\boldsymbol{\theta}^k},k}(\boldsymbol{o}_l,\boldsymbol{o}_{l+1}) \cdot P^k(\boldsymbol{o}_1)\prod_{t=1}^{L-1}\hat{p}_{\boldsymbol{\theta}^k}(\boldsymbol{o}_{t+1}|\boldsymbol{o}_t) \cdot \nabla_{\boldsymbol{\theta}^k}\log\hat{p}_{\boldsymbol{\theta}^k}(\boldsymbol{o}_{l+1}|\boldsymbol{o}_l)d\boldsymbol{o}_{1:L}\\ &\stackrel{(3)}{=} \frac{1}{L}\sum_{l=1}^{L-1}\mathbb{E}_{\substack{\boldsymbol{o}_1 \sim P^k\\ \boldsymbol{o}_{t+1}\sim\hat{p}_{\boldsymbol{\theta}^k}}}\left[(1 + (2\mathbb{1}(A_{l+1}^{\hat{p}_{\boldsymbol{\theta}^k},k}(\boldsymbol{o}_l,\boldsymbol{o}_{l+1}) \geq 0) - 1)\epsilon_{\mathrm{clip}})A_{l+1}^{\hat{p}_{\boldsymbol{\theta}^k},k}(\boldsymbol{o}_l,\boldsymbol{o}_{l+1}) \cdot \nabla_{\boldsymbol{\theta}^k}\log\hat{p}_{\boldsymbol{\theta}^k}(\boldsymbol{o}_{l+1}|\boldsymbol{o}_l)\right]\end{aligned} \tag{119}$$

*The methodologies follow Lemma 8.*

*Step (1) expands the expectation using the MDP factorization $P^k(\boldsymbol{o}_1)\prod_{t=1}^{L-1}\hat{p}_{\boldsymbol{\theta}^k}(\boldsymbol{o}_{t+1}|\boldsymbol{o}_t)$, noting that $\hat{p}_{old}$ is treated as fixed behavioral policy;*

*Step (2) applies parameter-localized differentiation through:*

$$\nabla_{\boldsymbol{\theta}^k}\prod_{t=1}^{L-1}\hat{p}_t = \prod_{\substack{t=1\\t\neq l}}^{L-1}\hat{p}_{old,t} \cdot \nabla_{\boldsymbol{\theta}^k}\hat{p}_l \tag{120}$$

*with conditions similar in Lemma 7.*

*Step (3) reconstructs the expectation by recognizing $\prod_{t=1}^{L-1}\hat{p}_t = \hat{p}_{\boldsymbol{\theta}^k}(\boldsymbol{o}_{1:L})/P^k(\boldsymbol{o}_1)$, leveraging the Markov property.*

***Key Conditions Inherited from REINFORCE/RAFT in Lemma 7:*** *1. Parameter Isolation: $\boldsymbol{\theta}^k = \biguplus_{l=1}^{L-1}\boldsymbol{\theta}_l^k$ with disjoint subparameters 2. Policy Smoothness: $\hat{p}_{\boldsymbol{\theta}^k} \in C^2(\Theta)$ with bounded Hessians 3. Measure Consistency: $\prod_{t\neq l}\int\hat{p}_t d\boldsymbol{o}_{t+1} = 1$ $\mu$-a.e. 4. Advantage Regularity: $A_{l+1}^{\hat{p}_{\boldsymbol{\theta}^k},k}(\boldsymbol{o}_l,\boldsymbol{o}_{l+1})$ is $\sigma(\boldsymbol{o}_{1:l+1})$-measurable and bounded.* $\qquad\square$

Based on the policy gradient results, the logit update lemma is provided as below.

**Lemma 9.** *Let $\boldsymbol{\theta}^\star$ be the base model in Eq.(2) that exact behave like a Multi-task TMC as in Def. 1 and 2, and $\boldsymbol{\theta}^k$ the current model to be finetuned from $\boldsymbol{\theta}^\star$ for task $k \in \mathcal{T}$. Then*

$$\nabla_{\boldsymbol{\theta}^k}\mathcal{J}_{\mathrm{REINFORCE}}(\boldsymbol{\theta}^k) = \sum_{l=1}^{L-1}\mathbb{E}_{\substack{\boldsymbol{o}_1 \sim P^k(\mathcal{Q}^k)\\ \{\boldsymbol{o}_{l+1}\sim\hat{p}_{\boldsymbol{\theta}^k}(\cdot|\boldsymbol{o}_l)\}_{l=1}^{L-1}}}\left[R_{\mathrm{out}}^k(\boldsymbol{o}) \cdot (\nabla_{\boldsymbol{\theta}^k}h_{\boldsymbol{\theta}}(\boldsymbol{o}_{l+1},\boldsymbol{o}_l) - \sum_{\boldsymbol{o}_{l+1}' \in S_{l+1}}\hat{p}_{\boldsymbol{\theta}^k}(\boldsymbol{o}_{l+1}'|\boldsymbol{o}_l)\nabla_{\boldsymbol{\theta}^k}h_{\boldsymbol{\theta}}(\boldsymbol{o}_{l+1}',\boldsymbol{o}_l))\right], \tag{121}$$

$$\nabla_{\boldsymbol{\theta}^k}\mathcal{J}_{\mathrm{RAFT}}(\boldsymbol{\theta}^k) = \sum_{l=1}^{L-1}\mathbb{E}_{\substack{\boldsymbol{o}_1 \sim P^k(\mathcal{Q}^k)\\ \{\boldsymbol{o}_{l+1}\sim\hat{p}_{\boldsymbol{\theta}^k}(\cdot|\boldsymbol{o}_l)\}_{l=1}^{L-1}}}\left[R_{\mathrm{out}}^k(\boldsymbol{o}) \cdot (1 + \log\hat{p}_{\boldsymbol{\theta}^k}(\boldsymbol{o}_{l+1}|\boldsymbol{o}_l))(\nabla_{\boldsymbol{\theta}^k}h_{\boldsymbol{\theta}}(\boldsymbol{o}_{l+1},\boldsymbol{o}_l) - \sum_{\boldsymbol{o}_{l+1}' \in S_{l+1}}\hat{p}_{\boldsymbol{\theta}^k}(\boldsymbol{o}_{l+1}'|\boldsymbol{o}_l)\nabla_{\boldsymbol{\theta}^k}h_{\boldsymbol{\theta}}(\boldsymbol{o}_{l+1}',\boldsymbol{o}_l))\right], \tag{122}$$

$$\nabla_{\boldsymbol{\theta}^k} \mathcal{J}_{\mathrm{PO}}(\boldsymbol{\theta}^k) = \sum_{l=1}^{L-1} \mathbb{E}_{\substack{\boldsymbol{o}_1 \sim P^k(\mathcal{Q}^k) \\ \{\boldsymbol{o}_{l+1} \sim \hat{p}_{\boldsymbol{\theta}^k}(\cdot|\boldsymbol{o}_l)\}_{l=1}^{L-1}}} \left[ (1 + (2\mathbb{1}(A_{l+1}^{\hat{p}_{\boldsymbol{\theta}^k},k}(\boldsymbol{o}_l,\boldsymbol{o}_{l+1}) \geq 0) - 1)\epsilon_{\mathrm{clip}}) A_{l+1}^{\hat{p}_{\boldsymbol{\theta}^k},k}(\boldsymbol{o}_l,\boldsymbol{o}_{l+1}) \cdot (\nabla_{\boldsymbol{\theta}^k} h_{\boldsymbol{\theta}}(\boldsymbol{o}_{l+1},\boldsymbol{o}_l) - \sum_{\boldsymbol{o}'_{l+1} \in S_{l+1}} \hat{p}_{\boldsymbol{\theta}^k}(\boldsymbol{o}'_{l+1}|\boldsymbol{o}_l)\nabla_{\boldsymbol{\theta}^k} h_{\boldsymbol{\theta}}(\boldsymbol{o}'_{l+1},\boldsymbol{o}_l)) \right].$$

$$(123)$$

*Further, we have*

$$\nabla_{\boldsymbol{\theta}^k} \mathcal{J}_{\mathrm{REINFORCE}}(\boldsymbol{\theta}^k) = \sum_{l=1}^{L-1} \mathbb{E}_{\substack{\boldsymbol{o}_1 \sim P^k(\mathcal{Q}^k) \\ \{\boldsymbol{o}_{l+1} \sim \hat{p}_{\boldsymbol{\theta}^k}(\cdot|\boldsymbol{o}_l)\}_{l=1}^{L-1}}} \left[ R_{\mathrm{out}}^k(\boldsymbol{o}) \cdot (e_{o_{l+1},o_l} - \sum_{\boldsymbol{o}'_{l+1} \in S_{l+1}} \hat{p}_{\boldsymbol{\theta}^k}(\boldsymbol{o}'_{l+1}|\boldsymbol{o}_l)e_{o'_{l+1},o_l}) \right],$$

$$(124)$$

$$\nabla_{\boldsymbol{\theta}^k} \mathcal{J}_{\mathrm{RAFT}}(\boldsymbol{\theta}^k) = \sum_{l=1}^{L-1} \mathbb{E}_{\substack{\boldsymbol{o}_1 \sim P^k(\mathcal{Q}^k) \\ \{\boldsymbol{o}_{l+1} \sim \hat{p}_{\boldsymbol{\theta}^k}(\cdot|\boldsymbol{o}_l)\}_{l=1}^{L-1}}} \left[ R_{\mathrm{out}}^k(\boldsymbol{o}) \cdot (1 + \log \hat{p}_{\boldsymbol{\theta}^k}(\boldsymbol{o}_{l+1}|\boldsymbol{o}_l))(e_{o_{l+1},o_l} - \sum_{\boldsymbol{o}'_{l+1} \in S_{l+1}} \hat{p}_{\boldsymbol{\theta}^k}(\boldsymbol{o}'_{l+1}|\boldsymbol{o}_l)e_{o'_{l+1},o_l}) \right],$$

$$(125)$$

$$\nabla_{\boldsymbol{\theta}^k} \mathcal{J}_{\mathrm{PO}}(\boldsymbol{\theta}^k) = \sum_{l=1}^{L-1} \mathbb{E}_{\substack{\boldsymbol{o}_1 \sim P^k(\mathcal{Q}^k) \\ \{\boldsymbol{o}_{l+1} \sim \hat{p}_{\boldsymbol{\theta}^k}(\cdot|\boldsymbol{o}_l)\}_{l=1}^{L-1}}} \left[ (1 + (2\mathbb{1}(A_{l+1}^{\hat{p}_{\boldsymbol{\theta}^k},k}(\boldsymbol{o}_l,\boldsymbol{o}_{l+1}) \geq 0) - 1)\epsilon_{\mathrm{clip}}) A_{l+1}^{\hat{p}_{\boldsymbol{\theta}^k},k}(\boldsymbol{o}_l,\boldsymbol{o}_{l+1}) \cdot (e_{o_{l+1},o_l} - \sum_{\boldsymbol{o}'_{l+1} \in S_{l+1}} \hat{p}_{\boldsymbol{\theta}^k}(\boldsymbol{o}'_{l+1}|\boldsymbol{o}_l)e_{o'_{l+1},o_l}) \right].$$

$$(126)$$

**Proof.** *By Eq. (95) and Eq.(90), we have*

$$\nabla_{\boldsymbol{\theta}^k} \mathcal{J}_{\mathrm{REINFORCE}}(\boldsymbol{\theta}^k) = \sum_{l=1}^{L-1} \mathbb{E}_{\substack{\boldsymbol{o}_1 \sim P^k(\mathcal{Q}^k) \\ \{\boldsymbol{o}_{l+1} \sim \hat{p}_{\boldsymbol{\theta}^k}(\cdot|\boldsymbol{o}_l)\}_{l=1}^{L-1}}} \left[ \nabla_{\boldsymbol{\theta}^k} \log \hat{p}_{\boldsymbol{\theta}^k}(\boldsymbol{o}_{l+1}|\boldsymbol{o}_l) R_{\mathrm{out}}^k(\boldsymbol{o}) \right]$$

$$= \sum_{l=1}^{L-1} \mathbb{E}_{\substack{\boldsymbol{o}_1 \sim P^k(\mathcal{Q}^k) \\ \{\boldsymbol{o}_{l+1} \sim \hat{p}_{\boldsymbol{\theta}^k}(\cdot|\boldsymbol{o}_l)\}_{l=1}^{L-1}}} \left[ R_{\mathrm{out}}^k(\boldsymbol{o}) \cdot (\nabla_{\boldsymbol{\theta}^k} h_{\boldsymbol{\theta}}(\boldsymbol{o}_{l+1},\boldsymbol{o}_l) - \sum_{\boldsymbol{o}'_{l+1} \in S_{l+1}} \hat{p}_{\boldsymbol{\theta}^k}(\boldsymbol{o}'_{l+1}|\boldsymbol{o}_l)\nabla_{\boldsymbol{\theta}^k} h_{\boldsymbol{\theta}}(\boldsymbol{o}'_{l+1},\boldsymbol{o}_l)) \right].$$

$$(127)$$

*Similarly By Eq. (97), Eq.(90), we obtain the results of RAFT. Given Eq.(2), we have*

$$\nabla_{\boldsymbol{\theta}^k} \mathcal{J}_{\mathrm{REINFORCE}}(\boldsymbol{\theta}^k) = \sum_{l=1}^{L-1} \mathbb{E}_{\substack{\boldsymbol{o}_1 \sim P^k(\mathcal{Q}^k) \\ \{\boldsymbol{o}_{l+1} \sim \hat{p}_{\boldsymbol{\theta}^k}(\cdot|\boldsymbol{o}_l)\}_{l=1}^{L-1}}} \left[ \nabla_{\boldsymbol{\theta}^k} \log \hat{p}_{\boldsymbol{\theta}^k}(\boldsymbol{o}_{l+1}|\boldsymbol{o}_l) R_{\mathrm{out}}^k(\boldsymbol{o}) \right]$$

$$= \sum_{l=1}^{L-1} \mathbb{E}_{\substack{\boldsymbol{o}_1 \sim P^k(\mathcal{Q}^k) \\ \{\boldsymbol{o}_{l+1} \sim \hat{p}_{\boldsymbol{\theta}^k}(\cdot|\boldsymbol{o}_l)\}_{l=1}^{L-1}}} \left[ R_{\mathrm{out}}^k(\boldsymbol{o}) \cdot (\nabla_{\boldsymbol{\theta}^k} h_{\boldsymbol{\theta}}(\boldsymbol{o}_{l+1},\boldsymbol{o}_l) - \sum_{\boldsymbol{o}'_{l+1} \in S_{l+1}} \hat{p}_{\boldsymbol{\theta}^k}(\boldsymbol{o}'_{l+1}|\boldsymbol{o}_l)\nabla_{\boldsymbol{\theta}^k} h_{\boldsymbol{\theta}}(\boldsymbol{o}'_{l+1},\boldsymbol{o}_l)) \right].$$

$$(128)$$

*Given Eq.(2), by Eq.(92) we have*

$$\nabla_{\boldsymbol{\theta}^k} \mathcal{J}_{\mathrm{REINFORCE}}(\boldsymbol{\theta}^k) = \sum_{l=1}^{L-1} \mathbb{E}_{\substack{\boldsymbol{o}_1 \sim P^k(\mathcal{Q}^k) \\ \{\boldsymbol{o}_{l+1} \sim \hat{p}_{\boldsymbol{\theta}^k}(\cdot|\boldsymbol{o}_l)\}_{l=1}^{L-1}}} \left[ R_{\mathrm{out}}^k(\boldsymbol{o}) \cdot (e_{o_{l+1},o_l} - \sum_{\boldsymbol{o}'_{l+1} \in S_{l+1}} \hat{p}_{\boldsymbol{\theta}^k}(\boldsymbol{o}'_{l+1}|\boldsymbol{o}_l)e_{o_{l+1},o_l}) \right].$$

*The results of RAFT and PO follows.*

$$\square$$

