# OpenReview forum: "Consistency Is Not Always Correct: Towards Understanding the Role of Exploration in Post-Training Reasoning"
_ICLR.cc/2026/Conference — ICLR 2026 Conference Desk Rejected Submission_

### Official Review · Reviewer_vABH · 2025-10-31

**Soundness:** 3
**Presentation:** 3
**Contribution:** 2
**Rating:** 2
**Confidence:** 4

**Summary:**

Standard RL training methods sharpen and do not encourage exploration. The authors propose a theoretical framework, TMC, to model the sequential structure of reasoning, viewing pretraining as tree-graph discovering and post-training as Chain-of-Thought (CoT) reweighting. Within this framework, they distinguish between **easy-to-reason CoTs** (common, high-probability transitions) and **hard-to-reason CoTs** (rare, low-probability transitions), linking instance difficulty to the base model's pass rate.

The central contribution is the rigorous formalization and proof that standard post-training methods introduce a **simplicity bias**, thereby failing to maintain access to rare but crucial CoTs needed for difficult problem instances. Some experimentation is performed to validate the theoretical findings.

**Strengths:**

- clear presentation
- rigorous proofs

**Weaknesses:**

All effects stated have been well-noted before:
- the squeezing effect has already been noted many times in literature before, often referred to as sharpening [1]
- entropy decrease has also been a phenomenon commonly noted [2]
- "certain rare, high-uncertainty CoTs by base model are responsible for solving hard problem instances" isn't this true by definition of hard problems?

Moreover, the strategies of rejecting easy instances [3] and KL regularization have also been explored before as ways to regulate exploration. Are you suggesting any novel strategies?

Lack of experiments to demonstrate the correctness of theoretical results. There is currently only a single set of experiments showing a toy task, but how do they hold up on Math, QA, coding, etc.?

[1] https://arxiv.org/abs/2506.02355
[2] https://arxiv.org/abs/2505.22617
[3] https://arxiv.org/abs/2506.09026

**Questions:**

1. Entropy **does not** necessarily decrease when running on difficult problems. How would youaddress this class of fine-tuning?
2. Is the goal of this paper to provide theory behind existing known phenomenons? If so, are there any training strategies outside of common existing ones that theory sheds light upon?

---

> ### Author Response · Authors · 2025-11-21
>
> Thank you for your time and feedback. We sincerely appreciate your recognition of the paper’s clarity and the rigor of its proofs—both essential qualities for a theoretical contribution. However, we respectfully clarify that the reviewer confuses a methodology/empirical paper (e.g., [1-3]) and a theoretical paper (ours). Below, we address each of your comments point by point.
>
> ---
>
> ***W1 & Q2.*** *All effects stated have been well-noted before; Is the goal of this paper to provide theory behind existing known phenomena?*
>
> **A1:**
> Thank you for raising this question. Yes—as stated in both the abstract and introduction—our work is positioned as a **theoretical contribution** that *unifies and explains* several **empirically observed** yet previously **disconnected** phenomena, *rather than* as an empirical or methodological paper proposing new training algorithms such as [1–3].
>
> Indeed, theoretical studies have intrinsic value in explaining real-world observations through the analysis of models that, while simplified, capture essential underlying principles. Both **Reviewer j2oD** and **Reviewer y8Az** explicitly recognized the novelty of our theoretical contribution, highlighting our solid mathematical grounding that connects multiple empirical findings and our insightful linkage between the proposed *Doob’s h-transform–induced Process Reward Model (DPRM)* framework and heuristic PRMs. They also noted the theory’s strength in addressing a paradox repeatedly observed in empirical literature:
>
> > Why can exploration improve reasoning (e.g., [2–3]), even when RL finetuning or 0–1 reward scaling often reinforce existing solution strategies rather than discovering new ones (e.g., Yue et al., NeurIPS 2025, Oral)?
>
> We address this question by rigorously connecting and explaining  **three empirically established phenomena** (Phenomena 1-3 in our introduction):
>
> * **Ph. 1. Squeezing (Sharpening) effect of RL finetuning** — rare but correct CoTs are forgotten, a phenomenon *empirically observed* in math and coding tasks (e.g., [1]) but lacking theoretical grounding.
> * **Ph. 2. Neural verifier consistency bias** — verifiers tend to prefer internally consistent reasoning over factual correctness (Xu et al., 2025).
> * **Ph. 3. Hard problems rely on rare CoTs** — when “hardness” is defined by low base-model pass rates (Tong et al., 2025).
>
> While previous empirical works documented these effects independently, none provided a unifying theoretical explanation. Our analysis integrates them under one framework and reveals that  **exploration** even when  *confined within the base model’s reasoning tree* can prevent the forgetting of rare-but-correct CoTs crucial for difficult tasks—thereby  **resolving the paradox above**
>
> To formalize this connection, we adopt a theoretical setup similar to Kim et al. (2025), modeling easy (high-probability) and hard (low-probability) reasoning steps as  **Markov transitions** In this abstraction, pretraining corresponds to  *tree discovery*, and post-training to  *CoT reweighting* We prove that this minimal model captures **Ph. 1–3** and further provides theoretical justification for the effectiveness of exploration-based strategies such as **rejection of easy instances** and  **KL regularization**
>
> In summary, our work aims to provide a **unified and analytically tractable theoretical framework** that helps the community interpret and understand existing empirical phenomena—a standard and valuable objective for theoretical contributions (e.g., Foster et al., COLT 2025; Kim et al., ICML 2025).
>
> ---
>
> **Remark on References [1–3].**
>
> We thank the reviewer for these helpful suggestions. Specifically:
>
> * **[1]** reinforces our **Ph. 1** discussion: it empirically observes the GRPO sharpening bias but offers  **no theoretical explanation** Our work generalizes this to many RL finetuning and inference-scaling regimes.
> * **[2]** (already cited in our original text) proposes exploration mechanisms like Clip-Cov, but it **does not theoretically link** entropy collapse to the sharpening effect. We explicitly establish this link, along with other phenomena.
> * **[3]** provides **additional empirical evidence** for exploration benefits, complementing our Section D discussion alongside Yu et al. (2025) and Xiong et al. (2025).
>
> We’d incorporate [1] and [3] into the revised “Related Work” section and expanded the discussion accordingly. Thank you.
>
> ---

---

> ### Author Response · Authors · 2025-11-21
>
> ***W2.*** *Lack of experiments to demonstrate the correctness of theoretical results. There is currently only a single set of experiments showing a toy task, but how do they hold up on Math, QA, coding, etc.?*
>
> **A2:** Thank you for raising this question. We respectfully disagree for the following reasons:
>
> 1. **For validating a theory, it is both natural and sufficient to evaluate it within its intended theoretical setting.**
>    Our results are derived under a  **textbook-style Markov Chain framework** where all assumptions are explicitly controlled and observable. Accordingly, we validate our theory  *within this framework* as done comprehensively in **Sec. 5** and  **App. D** which confirm all theoretical predictions.
>    Extending these experiments to full real-world domains would inevitably introduce **unmodeled complexities** beyond the scope of the theory. In particular, the observations underlying **Ph. 2–3** remain valid due to their design nature, while extending the guarantees for **Ph. 1** would require additional assumptions and tools (e.g., empirical Neural Tangent Kernel analysis)—see our **A1** to Reviewer j2oD,  **App. E** and **App. G.4** for detailed discussion.
> 2. **Our work follows the standard role and methodology of theoretical research.**
>    As clarified in  **A1** our aim is to **explain widely-reported** observed phenomena (e.g., the sharpening effect on math-proving datasets reported in [1]) within a  **tractable and widely accepted theoretical framework** This is a common and well-established practice: for example, **Foster et al. (COLT 2025)** and **Kim et al. (ICML 2025)** analyze realizable linear MDPs and metastable Markov chains to conceptually mirror LLM behavior—both  **without experiments**
>    In this sense, theoretical work is **complementary** to empirical studies like [1]: one observes, while the other explains.
>
> That said, we are running on additional experiments on math-proving tasks. Despite current experiments have already demonstrated our theoretical correctness, we plan to put the additional experiments in the App.D to enrich our content.
>
> ---
>
> ***Q1.*** *Entropy **does not** necessarily decrease when running on difficult problems. How would you address this class of fine-tuning?*
>
> **A3:**
> Thank you for this comment. We would like to emphasize that our theoretical analysis focuses on a **specific, controlled setting** designed to explain the **targeted** phenomena, rather than to account for all possible empirical behaviors.
>
> That said, we appreciate the interesting case you raised. Although the specific data distribution, model architecture, or finetuning protocol was not specified, we agree that entropy may fail to decrease—particularly when the training dynamics become unstable. In the context of **RLVR finetuning** using only *outcome rewards* (without SFT supervision):
>
> * If the base model’s capability is too weak for the target task (i.e., *pass rate* is too low), the **gradient variance** can become excessively large, leading to **chaotic updates** and potential *entropy increase* (Li & Ng, 2025).
> * Likewise, if the **reward oracle** is **noisy or unable to verify intermediate reasoning steps** for difficult problems (as discussed in  **Sec. E** ), the supervision signals become inconsistent, again possibly causing entropy to increase.
>
> These situations lie  **outside our theoretical assumptions** since our framework requires the low-probability transition edge to remain above a constant (c), and does not model oracle noise (see Sec. E). As suggested by Li & Ng (2025) and Wen et al. (2025), incorporating **teacher-forced SFT** to improve the base model’s competence, and enhancing  **reward oracle fidelity** —for example, by verifying intermediate steps using tools such as *Lean4* in theorem-proving—can stabilize such finetuning processes and mitigate this phenomenon.
>
> ---

---

> ### Author Response · Authors · 2025-11-21
>
> ***Q2.*** *Are there any training strategies outside of common existing ones that theory sheds light upon?*
>
> **A4:** We appreciate this stimulating question. Indeed, our theoretical framework may shed light on several *non-standard* post-training strategies beyond conventional RL or KL-regularized finetuning. For example:
>
> * **Evolution Strategy (ES) Finetuning** (Qiu et al., 2025): updates the parameter matrix $\theta_t$ at iteration (t) via
>
>   $$
>   \theta\_{t+1} = \theta\_t + \frac{\alpha}{N}\sum_{n=1}^{N} R(\theta_t + \sigma\epsilon_n)\epsilon_n,
>   $$
>
>   where $\alpha, \sigma$ are hyperparameters, $R(·)$ is reward model, and $\epsilon_n \sim \mathcal{N}(0, I)$ are **isotropic** Gaussian perturbations.
>   In our framework, such **isotropic** exploration (i.e., equal reward-access chance of $\theta_{o_{l+1}^{\text{hard}},o_{l}}$ and $\theta_{o_{l+1}^{\text{easy}},o_{l}}$ at $l$)—compared to RL finetuning, which depends on the model’s own CoT generation that prefers easy CoTs ($\hat{p}\_{\theta}(o\_{l+1}^{\text{easy}}|o\_{l})>\hat{p}\_{\theta}(o\_{l+1}^{\text{hard}}|o\_{l})$)—does **not** intensively reinforce frequent “easy’’ parameter directions. Consequently, ES naturally **mitigates the squeezing effect** by avoiding over-reinforcement of easy CoT steps.
> * **Representation-based Exploration Finetuning** (Tuyls et al., 2025): encourages **large covariance** and diversity in latent representations. Under our theoretical lens, this corresponds to maintaining diverse parameter directions (i.e., prevent $\theta_{o_{l+1}^{\text{hard}},o_{l}}$ from being squeezed), thereby increasing coverage over low-probability transitions—exactly the mechanism our theory identifies as beneficial for preventing the squeezing effect.
>
> These examples illustrate that our theoretical framework not only consolidates existing intuitions but also has the potential to shed light on  **new or less conventional exploration-based post-training strategies**.
>
> ---
>
> **Summary**
>
> Once again, while we stand by our position that theoretical work has its own merits and serves as a valuable complement to methodological or empirical studies (e.g., [1–3]), we sincerely thank the reviewer for the constructive feedback and for recognizing the rigor and clarity of our theoretical contributions. Should the reviewer have further suggestions or wish to discuss any specific points in more detail, we would be very happy to continue this productive exchange. We greatly appreciate your time and insightful comments.
>
> ---
>
> ### **References**
>
> Foster et al (2025) Is a Good Foundation Necessary for Efficient Reinforcement Learning? The Computational Role of the Base Model in Exploration.
>
> Kim et al. (ICML 2025). Metastable Dynamics of Chain-of-Thought Reasoning: Provable Benefits of Search, RL, and Distillation.
>
> Yue et al. (NeurIPS 2025 Oral). Does Reinforcement Learning Really Incentivize Reasoning Capacity in LLMs Beyond the Base Model?
>
> Xu et al. (2025). Reward models identify consistency, not causality.
>
> Tong et al. (NeurIPS 2025). DART-Math: Difficulty-Aware Rejection Tuning for Mathematical Problem-Solving.
>
> Yu et al. (2025). DAPO: An Open-Source LLM Reinforcement Learning System at Scale.
>
> Xiong et al. (2025). A Minimalist Approach to LLM Reasoning: From Rejection Sampling to Reinforce.
>
> Li & Ng (NeurIPS 2025). Reasoning Models Hallucinate More: Factuality-Aware Reinforcement Learning for Large Reasoning Models.
>
> Wen et al. (2025). Reinforcement Learning with Verifiable Rewards Implicitly Incentivizes Correct Reasoning in Base LLMs.
>
> Qiu et al. (2025). Evolution Strategies at Scale: LLM Fine-Tuning Beyond Reinforcement Learning.
>
> Tuyls et al. (2025). Representation-Based Exploration for Language Models: From Test-Time to Post-Training.

---

### Official Review · Reviewer_y8Az · 2025-10-31

**Soundness:** 3
**Presentation:** 3
**Contribution:** 3
**Rating:** 8
**Confidence:** 2

**Summary:**

This paper's core contribution is clearly identifying and modeling a key paradox: why exploration remains crucial even when the set of reasoning paths (the tree structure) is fixed. Through the TMC model, the paper establishes a direct link between hard/easy problems and the probability of rare/common CoT paths—a concise and powerful theoretical abstraction.

On this foundation, the paper discusses the failure modes of two primary post-training strategies:

1. Training-Time (RLVR):Section 3 (Thm 2) details the Squeeze Effect of RLVR. The paper argues the root cause is the inherent gap in the advantage function (prop 1) between easy-to-reason and hard-to-reason CoTs. This gap causes gradient updates to systematically suppress the probability of hard-to-reason CoTs, i.e. the ratio of the probability of hard-to-reason CoTs and easy-to-reason CoTs strictly decreases, leading to them being forgotten.
2. Inference-Time (Inference-Scaling): Section 4 (Thm 3) explores a separate, independent failure mode. **Even if hard-to-reason CoTs still exist), ORM/PRM reward models are inherently biased. Because they reward the Population Reward—the average correctness across all instances (Prop 2)—they systematically favor easy-to-reason CoTs (which have a higher average score). Consequently, when using BoN/BS on a hard instance, these reward models will incorrectly assign a high score to the easy-to-reason CoTs, causing the model to select the wrong answer and fail. This explains why, under such biased guidance, merely increasing the number of samples (like in BoN) cannot solve these hard problems.

**Strengths:**

1. The paper formalizes the intuitive paradox between fixed reasoning structures and the benefits of exploration. The TMC abstraction provides a clean mathematical model linking exploration, entropy, and rare CoTs.

2. It systematically explains three well-observed empirical trends — RLVR entropy collapse, ORM/PRM consistency bias, and the importance of rare CoTs — under one coherent framework.

3. Technical Novelty: The Doob's h-Transform-induced Process Reward Model in Sec 4 is quite technically novel. It provides a formal, constructive method to derive the exact step-wise process reward  that is mathematically guaranteed to generate samples from the optimal target Gibbs distribution. This connection between DPRM and heuristically designed PRMs is very insightful.

**Weaknesses:**

1. The toy TMC simulation is conceptually clear but far from real LLM behavior. The gap between theory and practice (e.g., DeepSeek-R1, RLVR on GSM8K) remains large.

2. The multi-task TMC assumes disjoint task trees and fixed edge probabilities, which may not capture shared latent reasoning states or adaptive token dynamics in large models.

**Questions:**

no more

---

> ### Author Response · Authors · 2025-11-21
>
> Thank you for the reviewer’s thoughtful evaluation and for highlighting several strengths of our work—namely the clarity of our conceptual framework, the unified theoretical explanation it provides for multiple empirical phenomena, and the novelty and insightfulness of our DPRM formulation. We sincerely appreciate these positive remarks and the recognition of our contribution.
>
> ---
>
> ***W1&2.** The toy TMC simulation is conceptually clear but far from real LLM behavior. The gap between theory and practice (e.g., DeepSeek-R1, RLVR on GSM8K) remains large. The multi-task TMC assumes disjoint task trees and fixed edge probabilities, which may not capture shared latent reasoning states or adaptive token dynamics in large models.*
>
> **A1:** We thank the reviewer for acknowledging that our TMC simulation is clear. As is standard in theoretical work, our goal is *not* to match every detail of large-scale LLMs, but to explain observed phenomena by analyzing a hypothesized model that captures **partial but crucial** rationales. This is precisely the methodological structure used in prior theory papers: for example, Foster et al. (COLT 2025) analyze realizable linear MDPs, and Kim et al. (ICML 2025) study metastable Markov chains to conceptually mirror LLM reasoning—both entirely theoretical and without experiments. In the same spirit, our TMC is designed as a concise and powerful theoretical abstraction, exactly as the reviewer described.
>
> We agree that real models exhibit far richer behaviors, including non-disjoint trees, adaptive token dynamics, and non-Markovian dependence. These limitations are explicitly examined in several parts of the paper: the *Humble Remark* clarifies that our results capture only partial rationales and should not be overclaimed; **App. E** systematically enumerates unmodeled complexities (e.g., variable reasoning depth, nonlinear representations, structural priors, robustness issues, reward hacking, and non-Markovianity); and **App. G.4** discusses how considering more complex neural learner introduces representational coupling through eNTK geometry, outlining how the squeezing effect may change in this challenging regimes. We also outline multiple avenues for future extensions in these sections.
>
> Thus, while our abstraction is necessarily simplified, we have taken care to clearly document its limitations, justify its relevance, and situate it within the established practice of theoretical analyses that prioritize conceptual clarity over full fidelity.
>
> ---
>
> **Summary**
>
> Thank you once again for your encouraging feedback and for finding our unified theoretical framework valuable and novel. We are sincerely grateful for your thoughtful comments and the time you devoted to our work.
>
> ---
>
> ### **References**
>
> Foster et al (2025) Is a Good Foundation Necessary for Efficient Reinforcement Learning? The Computational Role of the Base Model in Exploration.
>
> Kim et al. (ICML 2025). Metastable Dynamics of Chain-of-Thought Reasoning: Provable Benefits of Search, RL, and Distillation.

---

### Official Review · Reviewer_j2oD · 2025-11-02

**Soundness:** 3
**Presentation:** 3
**Contribution:** 3
**Rating:** 6
**Confidence:** 3

**Summary:**

This paper introduces an analytically tractable theoretical framework termed Multi-task Tree-structured Markov Chains (TMC) to address a fundamental paradox in the post-training reasoning of foundation models. Although post-training strategies such as reinforcement learning from verifiable rewards and reasoning scaling methods rely on maintaining exploration or entropy stability to enhance performance, they often merely reinforce existing reasoning trajectories rather than truly expand the reasoning space. Through the proposed TMC framework, this work provides a formal and rigorous analysis of this phenomenon.

**Strengths:**

1.The paper provides solid theoretical justification for exploration strategies such as “rejecting easy instances” and KL regularization.

2.The concepts are clearly defined, and the writing is logically structured and easy to follow.
3.The proposed multi-task TMC framework, which models pre-training as graph discovery and post-training as CoT reweighting, is an elegant attempt to formalize CoT reasoning. It effectively links the difficulty of reasoning steps with transition probabilities in a Markov chain, achieving theoretical tractability.

**Weaknesses:**

1.The theoretical analysis relies on a simplified linear Softmax predictor and a constrained TMC topology. As acknowledged in the Humble Remark, there remains a gap between the proposed model and large-scale systems like GPT, with many unmodeled complexities. While tractability is valuable, such simplifications may limit the framework’s ability to capture the dynamics of real Transformer architectures and their vast, unstructured reasoning spaces.
2.Could you further justify how the mechanisms underlying the observed “squeezing effect” and “consistency bias” in your model, which are based on a simple Softmax and advantage function, would still hold in nonlinear, high-dimensional, multihead Transformer architectures?
3.The proposed Doob h transform based PRM (DPRM) is claimed to better preserve the capabilities of foundation models. Please briefly discuss its computational challenges and overhead, particularly in comparison with standard BoN or PRM approaches.

**Questions:**

see weaknesses.

---

> ### Author Response · Authors · 2025-11-21
>
> Thank you for your recognition of our work as solid, well-written and insightfulness in modeling. We sincerely appreciate the reviewer’s thoughtful comments and the opportunity to clarify your questions as below.
>
> ---
>
> ***W1.*** *The theoretical analysis relies on a simplified linear Softmax predictor and a constrained TMC topology. As acknowledged in the Humble Remark, there remains a gap between the proposed model and large-scale systems like GPT, with many unmodeled complexities. While tractability is valuable, such simplifications may limit the framework’s ability to capture the dynamics of real Transformer architectures and their vast, unstructured reasoning spaces. Could you further justify how the mechanisms underlying the observed “squeezing effect” and “consistency bias” in your model, which are based on a simple Softmax and advantage function, would still hold in nonlinear, high-dimensional, multihead Transformer architectures?*
>
> **A1:** Thank you for the thoughtful question. The short answer is  **yes** as discussed in **App. E** and  **App. G.4**. While it is common for theoretical analyses of large-scale LMs to use **idealized surrogate models** to distill and prove generalizable principles (e.g., Foster et al., COLT 2025; Kim et al., ICML 2025), we appreciate the opportunity to further clarify how our theory connects to nonlinear Transformer setting:
>
> **1. Linear surrogates sufficiently capture tabular latent-state transitions.**
> Although Transformers compute logits through deep nonlinear transformations, the final prediction is still a softmax over a  **tabular latent space** Our TMC formalizes  *latent reasoning states* following prior work (Xu et al., 2019; Sanford et al., 2024; Abbe et al., 2024). Such tabular latent transitions are **classically representable by linear models** with enough overparameterization (Sutton & Barto, 2014), and are widely adopted in recent theory (e.g., Kim et al., 2025). Hence the linear Softmax serves as an analytically transparent surrogate without loss of conceptual generality.
>
> **2. Extension to the general (nonlinear Transformer) case.**
> Our multi-task TMC recovers **three empirically observed** phenomena (Phenomenon 1-3 in the introduction) that also hold in nonlinear multihead Transformers:
>
> * **Ph. 3(architecture-agnostic): Hard instances rely on rare CoTs.**
>   This is defined by *base-model pass rate* (Tong et al., 2025) and is independent of the underlying architecture.
> * **Ph. 2(architecture-agnostic): Consistency bias of neural verifiers.**
>   As shown in  **Prop. 2** likelihood-based population objectives intrinsically upweight frequent patterns and downweight rare CoTs—this bias arises from the  **objective** itself, not the network class. This explains the observed phenomenon in real-world LLMs (Xu et al., 2025).
> * **Ph. 1(architecture-aware): RL-induced squeezing.**
>   *Squeezing (sharpening) under RL post-training* : rare-but-correct CoTs are forgotten, a behavior **widely reported** in math/coding systems (e.g., He et al., 2025; Yu et al., 2025; Xiong et al., 2025) but previously lacking theoretical explanation. In our framework, it emerges when gradients **over-reinforce easy CoTs** driven by their higher advantage, also stemming from the decoupled neural representation of different states. **App. G.4** discusses how this mechanism extends to multi-layer non-linear case. A tractable route is to model post-training dynamics via the  **empirical Neural Tangent Kernel** supported by evidence that deep models often behave in a lazy/kernel regime during post-training (Malladi et al., 2023). The main empirical task is then to characterize **nonlinear correlations between latent reasoning states** (e.g., semantic proximity of hard vs. easy steps) via probing and to relax our independence assumptions accordingly. Our current abstraction **decouples** these interactions for  **analytic clarity** but the architecture-aware refinements are a natural and promising future direction.
>
> ---

---

> ### Author Response · Authors · 2025-11-21
>
> ***W2.*** *The proposed Doob h-transform–based PRM (DPRM) is claimed to better preserve the capabilities of foundation models. Please briefly discuss its computational challenges and overhead, particularly in comparison with standard BoN or PRM approaches.*
>
> **A2:**  Thank you for this valuable question. We are happy to point out that the Doob h-transform PRM (DPRM) provides theoretical interpretability *without introducing heavy computational burden* compared to existing PRM or BoN approaches. Specifically:
>
> 1. **Computational Cost.**
>    As discussed in  **Remark 4** DPRM incurs **no asymptotic overhead** compared with the heuristic likelihood-based PRM in Eq. (11). Both methods compute  *state-conditioned expectations* ; DPRM simply applies a different weighting (i.e., $\log\mathbb{E}[\exp(\cdot)\mid\cdot]$ instead of $\mathbb{E}[\cdot\mid\cdot]$). Although this involves extra evaluations of $\exp$ and $\log$, the additional cost is negligible in practice and does not change the computational budget.
> 2. **Relation to Standard BoN/Beam Search with PRM. Corollary 4** shows that combining BoN or Beam Search with DPRM is **theoretically equivalent** to using them with conventional PRMs, since DPRM yield identical **rankings** over CoTs as likelihood PRM -The reward values differ by Jensen’s inequality, but rankings are identical. It is worth noting that in practical systems, pretrained PRMs (e.g.,  *Qwen2.5-Math-PRM-72B* ) already provide stable heuristic step-wise rewards, so **large-scale DPRM retraining is unnecessary** when such PRMs are available.
> 3. **Sampling Efficiency**.
>    The **DPRM-based Approximate Sampling (DPRM-AS)** in Eq. (14) corresponds to a *soft-BoN* sampling scheme using DPRM as an oracle. Following Verdun et al. (2025), its approximation error decreases at the rate $O(1/N)$. Thus, its computational overhead relative to standard BoN arises solely from the number of samples needed to approximate the Gibbs distribution in Eq. (12): achieving precision $\varepsilon$ requires roughly $1/\varepsilon$ samples. No further computational costs are introduced.
>
> ---
>
> **Summary**
>
> We’re encouraged that the reviewer value our work as solid, logically structured and easy to follow. We highly value the reviewer’s valuable feedback. Should the reviewer wish to discuss any point further, we would be more than delighted to continue our productive exchange. Once again, we deeply appreciate the reviewer’s time and valuable comments.
>
> ---
>
> ### **References**
>
> Foster et al (COLT 2025) Is a Good Foundation Necessary for Efficient Reinforcement Learning? The Computational Role of the Base Model in Exploration.
>
> Kim et al. (ICML 2025). Metastable Dynamics of Chain-of-Thought Reasoning: Provable Benefits of Search, RL, and Distillation.
>
> Xu et al. (ICLR 2020) What can neural networks reason about?
>
> Sanford et al. (NeurIPS 2024) Understanding transformer reasoning capabilities via graph algorithms
>
> Abbe et al. (NeurIPS 2024) How Far Can Transformers Reason? The Globality Barrier and Inductive Scratchpad
>
> Tong et al. (NeurIPS 2025). DART-Math: Difficulty-Aware Rejection Tuning for Mathematical Problem-Solving
>
> He et al. (2025). Rewarding the Unlikely: Lifting GRPO Beyond Distribution Sharpening
>
> Xu et al. (2025). Reward models identify consistency, not causality.
>
> Yu et al. (2025). DAPO: An Open-Source LLM Reinforcement Learning System at Scale.
>
> Xiong et al. (2025). A Minimalist Approach to LLM Reasoning: From Rejection Sampling to Reinforce.
>
> Malladi et al (2023) A kernel-based view of language model fine-tuning
>
> Verdun et al. (2025) Soft best-of-n sampling for model alignment

---

### Author Response · Authors · 2025-12-03
**Summary of Reviewer Feedback and Note on a Misaligned Evaluation Criterion**

Dear PCs, SACs. ACs, and Reviewers,

Thank you for your valuable time and efforts in reviewing this paper. We are encouraged that all reviewers have acknowledged our work as solid and well-written. We are also pleased that the main concerns raised have been thoroughly addressed in our rebuttals. In particular:

- **Reviewer j2oD** asked about extending our framework to more realistic settup (**W1**). We addressed this comprehensively in **A1**—justifying our modeling choices in **A1.1** and discussing how the three phenomena can be shown in more complex scenarios in **A1.2** consistent with our original Appendix G.4. The reviewer also inquired whether the DPRM framework introduces substantial computational overhead (**W2**). As clarified in **A2**, the additional burden is mild in computational cost, relationship with standard method, and sampling efficiency.

- **Reviewer y8Az** noted that our current theoretical model may not fully capture real-world complexity (**W1-2**). We provided detailed justification in **A1**, together with references to Appendix E and Appendix G.4, which outline these limitations and possible extensions.

- **Reviewer vABH** expressed critique that appear to stem from evaluating a theoretical work through the lens of an empirical/methodological papers (e.g., demanding new SOTA algorithms). We have clarified that our paper is a **theoretical** contribution aimed at *unifying and explaining* several *empirically observed* but previously disconnected phenomena rather than propose ad-hoc training methods (**A1**). We believe our theoretical contribution complements the empirical works cited by the reviewer, rather than competing with them. We respectfully urge the AC to evaluate our work based on its theoretical rigor and explanatory power, which all the reviewers have acknowledged.

Once again, we sincerely thank all reviewers for recognizing the **rigor and clarity** of our theoretical contribution, which unifies existing observations and resolves the underlying paradox within a coherent analytical framework capable of informing broader contexts and future extensions. We believe our work provides valuable theoretical **complementarity** to methodological and empirical research, and we appreciate your consideration of its acceptance.

Sincerely,

Authors of Submission 15942

---

### Note · Program_Chairs · 2026-01-17
**Submission Desk Rejected by Program Chairs**

The following references in this submission do not refer to real documents and/or have major errors in bibliographic information:

 Weijia Yuan, Shixiang Shane Gu, Pieter Abbeel, and Yiming Yu. Scaling reward learning for language modeling. arXiv preprint arXiv:2309.06878, 2023.